# SIMSHIFT: A BENCHMARK FOR ADAPTING NEURAL SURROGATES TO DISTRIBUTION SHIFTS

## ABSTRACT

Neural surrogates for Partial Differential Equations (PDEs) often suffer significant performance degradation when evaluated on unseen problem configurations, such as new initial conditions or structural dimensions. Meanwhile, Domain Adaptation (DA) techniques have been widely used in vision and language processing to generalize from limited information about unseen configurations. In this work, we address this gap through two focused contributions. First, we introduce SIMSHIFT, a novel benchmark dataset and evaluation suite composed of four industrial simulation tasks spanning diverse processes and physics: *hot rolling*, *sheet metal forming*, *electric motor design* and *heatsink design*. Second, we extend established DA methods to state-of-the-art neural surrogates and systematically evaluate them. These approaches use parametric descriptions and ground truth simulations from multiple source configurations, together with only parametric descriptions from target configurations. The goal is to accurately predict target simulations without access to ground truth simulation data. Extensive experiments on SIMSHIFT highlight the challenges of out-of-distribution neural surrogate modeling, demonstrate the potential of DA in simulation, and reveal open problems in achieving robust neural surrogates under distribution shifts in industrially relevant scenarios.

## 1 INTRODUCTION

PDE simulations are essential tools for understanding and predicting physical phenomena in engineering and science (Evans, 2010). Over recent years, machine learning has emerged as a novel and promising modeling option for complex systems (Brunton & Kutz, 2020), significantly accelerating and augmenting simulation workflows across diverse applications, including weather and climate forecasting (Pathak et al., 2022; Bodnar et al., 2025), material design (Merchant et al., 2023; Zeni et al., 2025) and protein folding (Abramson et al., 2024) to name a few.

In practice, however, models are often deployed outside of their training distribution. This *distribution shift* (Quionero-Candela et al., 2009; Wang et al., 2023) often leads to a significant performance degradation (Bonnet et al., 2022; Herde et al., 2024). A well known analogue is clinical microscopy: models trained with data collected at a few hospitals often fail when deployed at others because microscopes, staining protocols, and lighting conditions differ (Tellez et al., 2019; Koh et al., 2020). For neural surrogates an analogous "instrument shift" arises from new initial conditions, such as material parameters or mesh geometries not encountered during training. Robustness to distribution shifts is crucial for industrial adoption and deployment also because it is becoming a compliance requirement, as stated by Article 15 of the EU AI Act (European Union, 2024).

While methods for increasing out-of-distribution performance have been at the center of research for a long time (Ben-David et al., 2006; Shimodaira, 2000; Sugiyama et al., 2007b), to the best of our knowledge, no benchmark systematically investigates such methods on simulation tasks. Addressing this gap is particularly relevant in scientific and industrial settings, where generating ground truth simulation data is costly, limiting the diversity of training configurations. In contrast, parametric descriptions, such as material types or structural dimensions, are often readily available or easy to generate. This problem is known as *Unsupervised Domain Adaptation (UDA)* (Ben-David et al., 2010), where parametric (input) descriptions and full simulation outputs are available for each *source* configuration, while only input descriptions are provided for *target* configurations, without corresponding outputs.

Figure 1: Schematic overview of the SIMSHIFT framework. In training, the model has access to inputs (e.g., parameters and meshes), corresponding outputs $(x, y)$ from the source domain (left, blue), and only inputs $x'$ from the target domain (right, yellow) are available. The neural operator $g$ and the conditioning network $\phi$ are shared across domains and jointly optimized. Two loss terms are used: $\mathcal{L}_{\text{recon}}$, computed on source labels, and $\mathcal{L}_{\text{DA}}$, which aligns source and target $\phi$ features. After training, unsupervised model selection strategies choose $\theta_{k1}$, which is expected to perform best on target domain.

To investigate the potential of UDA for neural surrogate modeling, we provide simulation data across a range of realistic tasks from industrial engineering design. We introduce a comprehensive benchmark that evaluates established UDA methods and neural surrogates. An overview of the framework is shown in Figure 1. Our contributions can be summarized as follows:

- We propose four practical datasets with flexible distribution shifts in *hot rolling*, *sheet metal forming*, *electric motor*, and *heatsink* design, based on realistic simulation setups.

- We present, to the best of our knowledge, the first joint study of established neural surrogate architectures and UDA on engineering simulations with unstructured meshes.

- We introduce *SIMSHIFT*, a modular benchmarking suite that complements our datasets with baseline models and algorithms. It allows easy integration of new simulations, machine learning methods, domain adaptation techniques, and model selection strategies.

## 2 RELATED WORK

**Unsupervised Domain Adaptation.** UDA research covers a wide spectrum of results from theoretical foundations (Ben-David et al., 2010; Zellinger et al., 2021a) to modern deep learning methods (Liu & Xue, 2021; Zellinger et al., 2019; Zhu et al., 2021; Long et al., 2018). A prominent class of methods, dubbed as *representation learning*, aims to map the data to a feature space, where source and target representations appear similar, while maintaining enough information for accurate prediction. To enforce feature similarity between domains, algorithms often employ statistical (Sun & Saenko, 2016; Gretton et al., 2006; Zhang et al., 2019; Shalit et al., 2017) or adversarial (Ganin et al., 2015; Tzeng et al., 2017) discrepancy measures. One crucial yet frequently overlooked factor in the success of UDA methods is model selection. Numerous studies underline the critical impact of hyperparameter choices on UDA algorithm performance, often overshadowing the adaptation method itself (Musgrave et al., 2021; Zellinger et al., 2021b; Dinu et al., 2023; Yang et al., 2024). Even more, since labeled data is unavailable in the target domain, standard validation approaches become infeasible. Thus, it is essential to jointly evaluate adaptation algorithms alongside their associated unsupervised model selection strategies. In this work, we focus on importance weighting strategies (Sugiyama et al., 2007a; You et al., 2019), which stand out by their general applicability, theoretical guarantees and high empirical performance.

**Benchmarks for Unsupervised Domain Adaptation.** Numerous benchmark datasets and evaluation protocols have been established for UDA methods across various machine learning domains, including computer vision (Venkateswara et al., 2017; Peng et al., 2018; Arjovsky et al., 2019), natural language processing (Blitzer et al., 2007), timeseries data (Ragab et al., 2022) and tabular data (Gardner et al.,

2023). However, to the best of our knowledge, systematic UDA benchmarking for neural surrogates remains unexplored.

**Benchmarks for Neural Surrogates.** Recent years have seen a surge of surrogates belonging to the group of neural operators (see Appendix A), and benchmarks have grown alongside them. However, designing a robust and fair benchmark in the realm of PDEs is difficult and the current literature is not without shortcomings (Brandstetter, 2025). Many focus on solving PDEs on structured, regular grids (Gupta & Brandstetter, 2022; Takamoto et al., 2022; Ohana et al., 2024), which serve as valuable platforms for developing and testing new algorithms. However, these overlook the irregular meshes commonly used in large scale industrial simulations. In that direction, other benchmarks extend to Computational Fluid Dynamics (CFD) on irregular static meshes for airfoil simulations (Bonnet et al., 2022), aerodynamics for automotive (Elrefaie et al., 2024a;b), more academic fluid problems (Luo et al., 2023), and even particle based Smoothed Particle Hydrodynamics simulations (Toshev et al., 2023; 2024). Finally, and most closely related to our work, recent efforts have explored the application of Active Learning techniques (Cohn et al., 1996; Ren et al., 2021) to neural surrogates, introducing a benchmark specifically designed for scenarios where data is scarce (Musekamp et al., 2025). Despite these contributions, all current benchmarks often fall short when addressing a critical issue: the significant performance drop models exhibit under distribution shifts, i.e., when encountering simulation configurations beyond their training setting (Quionero-Candela et al., 2009).

## 3 DATASET PRESENTATION

Our datasets follow three design principles. (i) **Industry relevance:** They reflect practical, real-world simulation use-cases. The benchmark covers a diverse set of problems, including 2D as well as 3D cases. (ii) **Parametrized conditions:** The behavior of all simulations depends on the set of initial parameters only. (iii) **Steady-state scenarios:** We constrain them to time independent problems, being the standard use case in industry. Take for example design optimization tasks: most rely on either steady-state or time-averaged solutions rather than detailed transient dynamics. This is not just a modeling convenience, but reflects how simulation is integrated into design pipelines: numerical simulations are used to assess candidates by computing scalar objective values. This practice is well documented established various application areas, including thermal systems (Majumdar, 2021), aerodynamic shape optimization for aircrafts (Martins, 2022), wind turbine design (Martins, 2022), and car aerodynamics (Dumas, 2007). Additionally with this constraint we avoid additional complexities such as autoregressive error accumulation in neural surrogates (Lippe et al., 2023).

The datasets were generated using the commercial Finite Element Method (FEM) software *Abaqus*, the open-source simulation software *HOTINT* and the open-source CFD package *OpenFoam 9*. [1] An overview of each dataset together with its most important parameters and a custom metric, motivated by engineering practice, is presented in Sections 3.1 to 3.4. Additionally, we provide detailed descriptions of the respective numerical simulations in Appendix G. Since the behavior of each simulation task is entirely determined by its input parameters, we predefine source and target domains by partitioning the parameter space into distinct, non-overlapping regions. A detailed explanation of the domain splitting strategy is provided in Section 3.5. Table 1 summarizes key characteristics of each dataset, including physical dimensionality, mesh resolution, number of conditioning parameters, and total dataset size. All datasets are publicly hosted on Hugging Face[2].

Table 1: Overview of the benchmark datasets. Heatsink meshes are subsampled to a fourth of their original size. Detailed descriptions of the parameter sampling ranges can be found in Appendix G.

| Dataset | Origin | Samples | Output channels | Avg. # nodes | Varied simulation parameters | Dim | Size (GB) |
|---------|--------|---------|-----------------|--------------|------------------------------|-----|-----------|
| Rolling | Metallurgy | 4,750 | 10 | 576 | 4 | 2D | 0.5 |
| Forming | Manufacturing | 3,315 | 10 | 6,417 | 4 | 2D | 4.1 |
| Motor | Machinery | 3,196 | 26 | 9,052 | 15 | 2D | 13.4 |
| Heatsink | Electronics | 460 | 5 | 1,385,594 | 4 | 3D | 40.8 |

---

[1] Abaqus; HOTINT; OpenFoam 9.

[2] https://huggingface.co/datasets/simshift/SIMSHIFT_data

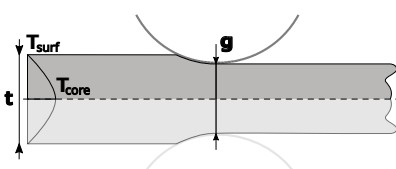 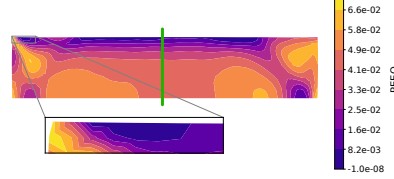

(a) Illustration of the simulation setup. The parameters correspond to those in Table 14. We use symmetry constraints and only simulate half of the slab.

(b) Metal slab after the process, showing PEEQ as a contour plot. The green line indicates the center cord, along which we measure the custom metric.

Figure 2: Overview of the *hot rolling* simulation scenario.

### 3.1 HOT ROLLING

**Problem Description.** The *hot rolling* process plastically deforms a metal slab into a sheet metal product, as visualized in Figure 2. This complex thermo-mechanical operation involves coupled elasto-plastic deformation and heat transfer phenomena (Gupta, 2021; Galantucci & Tricarico, 1999; Jo et al., 2023). The Finite Element (FE) simulation models the progressive thickness reduction and thermal evolution of the material as it passes through a rolling gap, incorporating temperature-dependent material properties and contact between the slab and the rolls. Among the output fields, the key quantity is Equivalent Plastic Strain (PEEQ), representing the material's plastic deformation, visualized in Figure 2b. The custom metric measures the relative error of the PEEQ profile along the slab's vertical center cord (green line in Figure 2b).

**Input parameters** are the initial slab thickness $t$, temperature characteristics $T_{\text{core}}$ and $T_{\text{surf}}$ of the slab, as well as the geometry of the roll gap. To vary the slab deformation we define the thickness reduction as a percentage of the initial thickness: reduction $= \frac{t-g}{t}$, where $g$ is the rolling gap distance. Table 14 in Appendix G.1 shows a detailed overview of the parameter values together with their sampling ranges used to generate the dataset.

### 3.2 SHEET METAL FORMING

**Problem Description.** The *sheet metal forming* process is a critical manufacturing operation widely used across industries such as automotive and aerospace. FEM simulations are commonly employed to estimate critical quantities such as thinning, local plastic deformation and residual stress distribution (Tekkaya, 2000; Ablat & Qattawi, 2017; Folle et al., 2024). The simulation setup consists of a symmetrical workpiece supported at the ends and center, a holder and a punch that deforms the sheet by applying a displacement ($U$ in Figure 3a). The 2D simulation predicts the sheet's elasto-plastic deformation, providing quantities such as stress, elastic and plastic strain distributions (shown in Figure 3b). An essential engineering metric used in practice is the transverse stress (xx-component) distribution along the vertical center cord (green line in Figure 3).

**Input parameters** include the deformed sheet length $l$, the sheet thickness $t$, friction coefficient $\mu$ and the radii of the holder, punch, and supports $r$. Table 15 in Appendix G.2 provides the sampling ranges for data generation.

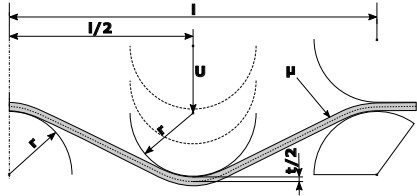 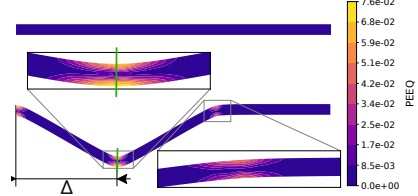

(a) Illustration of the simulation setup. The parameters correspond to those listed in Table 15.

(b) Material before (top) and after (bottom) the process, shown as PEEQ contours. $\Delta = l/2$

Figure 3: Overview of the *sheet metal forming* simulation scenario.

### 3.3 ELECTRIC MOTOR DESIGN

**Problem Description.** The *electric motor design* dataset encompasses a structural FEM simulation of a rotor in electric machinery, subjected to mechanical loading at burst speed. It is motivated by the conflicting design objectives in rotor development: while magnetic performance favors certain rotor topologies to optimize flux paths and torque generation, structural integrity requires designs capable of withstanding centrifugal loads without plastic deformation (Gerlach et al., 2021; Dorninger et al., 2021). The 2D simulation predicts stress and deformation responses due to assembly pressing forces and centrifugal loads, accounting for the rotor's topology, material properties, and rotation speed. The custom metric measures the relative error in Mises stress along the cord shown in green Figure 4.

**Input Parameters** together with their variations and a detailed technical drawing are omitted from the main body since this case is more complex than the preceding datasets. They are provided in Figure 31 and Table 17, both in Appendix G.3.

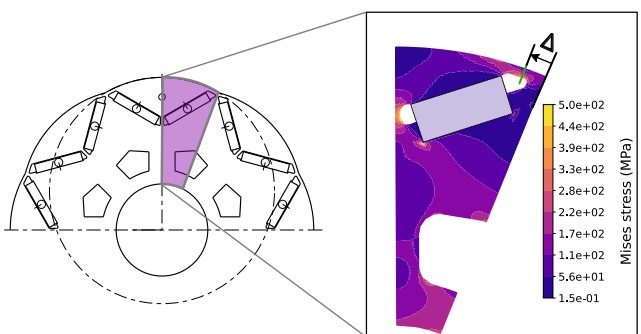

Figure 4: The *electric motor design* simulation scenario, with a schematic sketch of the motor (left) and zoomed-in detail from the simulated radial portion (right). Mises stress field contour plot is shown. The custom error metric is measured along the green line at $\Delta = \frac{t_{rsb1}}{2} + 1.1 * r_{r2}$.

### 3.4 HEATSINK DESIGN

**Problem Description.** The *heatsink design* dataset represents a CFD simulation focused on the thermal performance of heat sinks, commonly used in electronic cooling applications (Arularasan & Velraj, 2010; Rahman et al., 2024). It models the convective heat transfer from a heated base through an array of fins to the surrounding air. The simulation captures how geometric fin characteristics, specifically, the number, height, and thickness of fins, affect the overall heat dissipation, along with the temperature of the heat sink. Outputs include steady state temperature, velocity and pressure fields, enabling the assessment of design efficiency and thermal resistance under varying configurations. The main engineering metric measures the relative error in the temperature distribution along the dashed green line in Figure 5.

**Input Parameters** and their variations as well as an overview of the setup are provided in Appendix G.4.

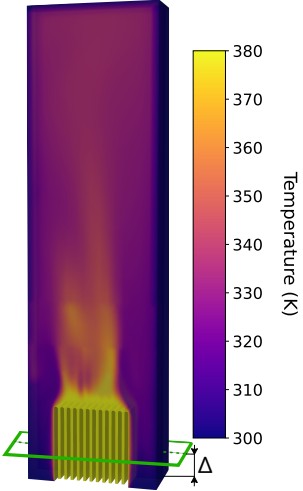

Figure 5: Slice of the *heatsink* 3D temperature field. Custom metric along dashed cord at $\Delta = 0.0025$.

### 3.5 DISTRIBUTION SHIFTS

SIMSHIFT's functionality allows for generating arbitrary n-dimensional parametric shifts for each problem, ensuring flexibility and extensibility. For benchmarking, each dataset includes three predefined distribution shifts: *easy*, *medium* and *hard*, which reflect increasing distributional distance in the respective input spaces (see Table 2 for parameter ranges). The source and target domains are constructed by shifting along the dominant input parameter of each simulation scenario, as suggested by domain experts.

To validate the design of our domain shifts we perform two analyses: (i) **Latent space inspection:** We train models across the full parameter ranges and perform a cluster analysis of their latent representations as the input conditions are varied. The resulting clusters consistently align with the parameters proposed by the experts, indicating that the chosen parameters dominate latent space variation (see visualizations in Figures 25 to 28, Appendix C). (ii) **Transfer difficulty validation:** Scalar parameter differences alone can be misleading regarding the actual shift difficulty experienced by models. We therefore provide the Proxy $\mathcal{A}$-Distance (PAD), which serves as an upper bound on the model's maximum transfer error. It works by bounding the $\mathcal{H}$-divergence, which in turn is an upper bound the maximum transfer error itself (for details see Bouvier et al. (2020), Johansson et al. (2019) and Zellinger et al. (2021b)). We estimate the PAD in the output spaces (ground truth simulation fields) using a PointNet Qi et al. (2017) mesh classifier. The resulting PAD values for each difficulty together with the domain defining parameter splits for all datasets are reported in Table 2.

The PAD values indicate a clear output-space distribution shift across all datasets. To illustrate this more concretely, consider the *hot rolling* dataset: in the *medium* difficulty setting, the range of PEEQ values in the source domain is $[0, 0.19]$, while in the target domain it extends to $[0, 0.28]$. Therefore part of the target solution field lies outside the support of the source field, demonstrating a genuine output-space shift in addition to the parametric input shift.

Beyond the predefined one-dimensional splits, we explore higher-dimensional distribution shifts. In Appendix H.2, we demonstrate that models, adaptation algorithms and model selection strategies exhibit consistent behavior under a two-dimensional shift in the *electric motor design* dataset.

Table 2: SIMSHIFT's predefined distribution shifts. We show the domain defining parameter and its respective ranges for all difficulty levels together with the corresponding PAD.

| Dataset | Parameter | Difficulty | Source range | Target range | PAD |
|---|---|---|---|---|---|
| Rolling | Reduction $r$ $(-)$ | easy | $[0.01, 0.13)$ | $[0.13, 0.15]$ | 1.063 |
| | | medium | $[0.01, 0.115)$ | $[0.115, 0.15]$ | 1.159 |
| | | hard | $[0.01, 0.10)$ | $[0.10, 0.15]$ | 1.210 |
| Forming | Thickness $t$ (mm) | easy | $[2, 4.8)$ | $[4.8, 5]$ | 0.860 |
| | | medium | $[2, 4.3)$ | $[4.3, 5]$ | 0.938 |
| | | hard | $[2, 4.1)$ | $[4.1, 5]$ | 1.030 |
| Electric Motor | Rotor slot diameter 3 $d_{r3}$ (mm) | easy | $[100, 122)$ | $[122, 126]$ | 0.762 |
| | | medium | $[99, 120)$ | $[120, 126]$ | 0.932 |
| | | hard | $[99, 118)$ | $[118, 126]$ | 0.955 |
| Heatsink | # fins | easy | $[5, 13)$ | $[13, 14]$ | 1.446 |
| | | medium | $[5, 12)$ | $[12, 15]$ | 1.683 |
| | | hard | $[5, 11)$ | $[11, 15]$ | 1.861 |

## 4 BENCHMARK SETUP

This section outlines the learning problem (Section 4.1), the UDA algorithms considered (Section 4.2), the unsupervised model selection strategies (Section 4.3), and the baseline models used (Section 4.4). Finally, we describe the experimental setup and evaluation metrics in Section 4.5.

### 4.1 LEARNING PROBLEM

Let $\mathcal{X}$ be an input space containing geometries and conditioning parameters (e.g., thickness and temperatures in Figure 2a) and $\mathcal{Y}$ be an output space containing ground truth solution fields, obtained from a numerical solver (e.g., PEEQ field in Figure 2b). Following (Ben-David et al., 2010), a *domain* is represented by a probability density function $p$ on $\mathcal{X} \times \mathcal{Y}$ (e.g., describing the probability of observing an input-output pair corresponding to the parameter range $r \in [0.01, 0.115)$ in Table 2). UDA has been formulated as follows: Given a source dataset $(x_1, y_1), ..., (x_n, y_n)$ drawn from a source domain $p_S$ together with an *unlabeled* target dataset $x'_1, ..., x'_m$ drawn from the ($\mathcal{X}$-marginal)

of a target domain $p_T$, the problem is to find a model $f : \mathcal{X} \to \mathcal{Y}$ that has small expected risk on the target domain:

$$\mathbb{E}_{(x,y)\sim p_T}[\ell(f(x), y)], \tag{1}$$

with $\ell : \mathcal{Y} \times \mathcal{Y} \to \mathbb{R}$ being some loss function. For example, consider the square loss $\ell(f(x), y) = (f(x) - y)^2$. In our setup $f(x) = g(x, \phi(x))$ is composed of a conditioning network $\phi$ and a surrogate $g$ (see Figure 1).

## 4.2 Unsupervised Domain Adaptation Algorithms

Our UDA baseline algorithms are from the class of *domain-invariant representation learning* methods. These methods are strong baselines, in the sense that their performance typically lies within the standard deviation of the winning algorithms in large scale empirical evaluations (i.e., no significant outperformance is observed), see CMD, Deep CORAL and DANN in (Dinu et al., 2023, Tables 12–14), M3SDA in (Peng et al., 2019), MMDA and HoMM in (Ragab et al., 2022).

Following Johansson et al. (2019) and Zellinger et al. (2021b), we express the objective of domain-invariant learning using two learning models: a *representation* mapping $\phi \in \Phi \subset \{\phi : \mathcal{X} \to \mathcal{R}\}$, which in our case corresponds to the conditioning network that maps simulation parameters into some representation space $\mathcal{R} \subset \mathbb{R}^k$ and a *regressor* $g \in \mathcal{G} \subset \{g : \mathcal{X} \times \mathcal{R} \to \mathcal{Y}\}$, which is realized by a neural surrogate. The goal is to find a mapping $\phi$ under which the source representations $\phi(\mathbf{x}) := (\phi(x_1), \dots, \phi(x_n))$ and the target representations $\phi(\mathbf{x}') := (\phi(x_1'), \dots, \phi(x_m'))$ appear similar, and, at the same time, enough information is preserved for prediction by $g$, see (Quionero-Candela et al., 2009). This is realized by estimating objectives of the form

$$\min_{g \in \mathcal{G}, \, \phi \in \Phi} \underbrace{\mathbb{E}_{(x,y)\sim p_S}\left[\ell\big(g(x, \phi(x)), y\big)\right]}_{\mathcal{L}_{\text{recon}}} + \lambda \cdot \underbrace{d\big(\phi(\mathbf{x}), \phi(\mathbf{x}')\big)}_{\mathcal{L}_{\text{DA}}}. \tag{2}$$

The training objective therefore consists of minimizing both terms: the supervised reconstruction loss $\mathcal{L}_{\text{recon}}$ and the domain adaptation loss $\mathcal{L}_{\text{DA}}$ as shown in Figure 1. A variety of UDA algorithms correspond to different implementations of the distance $d$. Good choices for $d$ in Equation (2) have been found to be the Wasserstein distance (Courty et al., 2017), the Maximum Mean Discrepancy (Baktashmotlagh et al., 2013), moment distances (Sun & Saenko, 2016; Zellinger et al., 2019), adversarially learned distances (Ganin et al., 2015) and other divergence measures (Johansson et al., 2019; Zhang et al., 2019). We outline the distance measures of all included algorithms in Appendix D. Furthermore, appropriately choosing the regularization parameter $\lambda$ is crucial for performance (Musgrave et al., 2021; Dinu et al., 2023; Yang et al., 2024), making model selection necessary.

## 4.3 Unsupervised Model Selection Strategies

Among all algorithm design choices in UDA, model selection has been repeatedly recognized as one of the most crucial (Musgrave et al., 2021; Yang et al., 2024), with sub-optimal choices potentially leading to *negative transfer* (Pan & Yang, 2010). However, classical approaches (e.g., validation set, cross-validation, information criterion) cannot be used due to missing labels and distribution shifts. It is therefore a natural benchmark requirement for UDA to provide also unified model selection strategies in addition to UDA algorithms.

In this work, we rely on Importance Weighted Validation (IWV) (Sugiyama et al., 2007a) and Deep Embedded Validation (DEV) (You et al., 2019) to overcome the two challenges: (i) distribution shift and (ii) missing target labels. These methods rely on the Radon-Nikodým derivative and the covariate shift assumption $p_S(y|x) = p_T(y|x)$ to obtain

$$\mathbb{E}_{(x,y)\sim p_T}[\ell(f(x), y)] = \mathbb{E}_{(x,y)\sim p_S}\left[\frac{p_T(x)\cancel{p_T(y|x)}}{p_S(x)\cancel{p_S(y|x)}}\ell(f(x), y)\right] = \mathbb{E}_{(x,y)\sim p_S}[\beta(x)\ell(f(x), y)]. \tag{3}$$

Equation (3) motivates to estimate the target error by a two step procedure: First, approaching challenge (i) by estimating the density ratio $\beta(x) = \frac{p_T(x)}{p_S(x)}$ from the input data only, and, approaching challenge (ii) by estimating target error by the weighted source error using *labeled* source data.

## 4.4 BASELINE MODELS

We provide a comprehensive range of machine learning methods, adapted to our conditioned simulation task, organized by their capacity to model interactions across different spatial scales:

*Global context models* such as PointNet (Qi et al., 2017) incorporate global information into local Multi-Layer Perceptrons (MLPs) by summarizing features of all input points by aggregation into a global representation, which is then shared among nodes. Recognizing the necessity of *local information* when dealing with complex meshes and structures, we include GraphSAGE (Hamilton et al., 2017), a proven Graph Neural Network (GNN) architecture (Scarselli et al., 2009; Battaglia et al., 2018) already used in other mesh based tasks (Pfaff et al., 2020; Bonnet et al., 2022). However, large scale applications of GNNs are challenging due to computational expense (Alkin et al., 2024a) and issues like oversmoothing (Rusch et al., 2023). Finally, to overcome these limitations, we employ *attention based models* (Vaswani et al., 2017). These models typically scale better with the number of points, and integrate both global and local information enabling stronger long-range interactions and greater expressivity. We include Transolver (Wu et al., 2024), a modern neural operator Transformer.

As an alternative categorization, baselines can also be classified by input-output pairings into *point-to-point* and *latent* approaches. The former explicitly encodes nodes, while the latter represents the underlying fields in a latent space and requires queries to retrieve nodes. While all previously mentioned models are *point-to-point*, we also include Universal Physics Transformer (UPT) (Alkin et al., 2024a; Fürst et al., 2025) and Geometry-Informed Neural Operator (GINO) Li et al. (2023b), as examples of latent field methods. Both methods are designed for large problems and offer favorable scaling on big meshes through latent field modeling. The main difference is that GINO latent space is constrained to a regular grid, where it operates in the *frequency* domain. UPT, in contrast, learns in a standard unconstrained latent domain. Both UPT and GINO are designed for large scale meshes, and therefore we benchmark them on the *heatsink design* dataset.

We provide detailed explanations of all implemented architectures in Appendix E. Our framework explicitly conditions neural operators on configuration parameters. We first embed them using a sinusoidal (sin–cos) encoding and a shallow MLP $\phi$ to produce a latent representation and then condition the neural operator $g$ by using either concatenation of the latent conditioning vector, FiLM (Perez et al., 2018) or DiT conditioning layers (Peebles & Xie, 2023). As an alternative, we also evaluate replacing $\phi$ with a geometric mesh encoder that derives the latent representation directly from the input geometry. On the *electric motor design* dataset, this variant performs worse (see Appendix H.1), supporting our design choice.

## 4.5 EXPERIMENTS AND EVALUATION

**Experimental Setup.** We benchmark four prominent UDA algorithms (Deep Coral (Sun & Saenko, 2016), CMD (Zellinger et al., 2019), DANN (Ganin et al., 2015) and DARE-GRAM (Nejjar et al., 2023)) in combination with the following four unsupervised model selection strategies: IWV (Sugiyama et al., 2007a), DEV (You et al., 2019), Source Best (SB) (selecting models based on source domain validation performance) and Target Best (TB) (selecting models based on target simulation data, which is not available in UDA but serves as a lower bound for perfect model selection).

For the baseline neural surrogate models, we evaluate PointNet, GraphSAGE, and Transolver on the *hot rolling*, *sheet metal forming*, and *electric motor design* datasets. Due to memory and runtime constraints on the large scale *heatsink design* dataset, we omit GraphSAGE and instead benchmark UPT and GINO alongside PointNet and Transolver.

**Experimental Scale.** We perform an extensive sweep over the critical UDA parameter $\lambda$ and average across four seeds, resulting in a total of **1,664** training runs (see Table 12). Details on architectures, hyperparameters, training setup and normalization, as well as a breakdown of training times are included in Appendices E and F.

**Evaluation Metrics.** For each dataset, we report the Normalized Root Mean Squared Error (NRMSE) averaged over all output fields, as well as the per field Root Mean Squared Error (RMSE) values

(computed on denormalized data), the Euclidean error for deformation predictions and the custom error metrics described in Sections 3.1 to 3.4. Additionally we provide physics-based evaluation metrics for all datasets. These metrics are tailored to the underlying PDEs. Detailed metric definitions are provided in Appendix F.2.

# 5 BENCHMARKING RESULTS

Table 3 overviews our benchmarking results, showing the best UDA and selection combination per model. Across datasets and architectures, UDA applied together with unsupervised model selection generally leads to a target error reduction, measured by NRMSE averaged across all fields. However, when examining the dataset-specific custom metrics introduced in Sections 3.1 to 3.4, the individual fields, and the physics-based metrics, gains are not uniform, and some methods improve the global loss, while performance on particular metrics. This pattern suggests that standard methods are a good starting point, but specialized algorithms tailored to high-dimensional regression tasks are needed. Furthermore, the gap between the best UDA + selection configurations and the TB oracle (lower bound on error) indicates that current unsupervised model selection strategies also leave room for improvement. Despite the clear benefits of UDA, no single UDA algorithm or unsupervised selection strategy dominates across all datasets. In addition to this summary, we report full source and target metrics across architectures, algorithms, and selection strategies in Tables 4 to 10.

Finally, since the presented tables only report performance on the *medium* difficulty setting, we additionally visualize model behavior of the best performing combination (model + UDA algorithm + selection strategy) across all difficulty levels of the *hot rolling* dataset in Figure 6. It illustrates the increase in prediction error as the domain gap widens and highlights the consistent improvements achieved by applying UDA algorithms combined with unsupervised model selection strategies on the *easy* and *medium* settings.

For the *hard* setting, however, the shown unsupervised model selection algorithm fails to identify suitable models, as the mean error matches that of the unregularized baselines with the standard deviation even increasing. Nonetheless, the theoretical lower bound (TB) remains substantially

Table 3: Best performing combination of UDA algorithm and unsupervised model selection for each dataset (*medium* difficulty) and architecture. We also report an oracle with target best (TB) selection, which provides a lower bound on the selection error. Entries show the target domain (N)RMSE or physics-based metric. Promotion on the unregularized baseline are shown in parentheses, with improvement indicated as negative values and asterisks marking unstable unregularized baselines. For each dataset, the best configuration (green) is chosen by the lowest NRMSE across all fields (bold).

| Dataset | Model | Best UDA Method + Model Selection | All Fields Normalized Avg (-) | Mises Stress (MPa) | Rel Custom Error (-) | VM Consistency (-) |
|---------|-------|-----------------------------------|-------------------------------|--------------------|----------------------|--------------------|
| Rolling | GraphSAGE | DARE-GRAM + IWV | **0.192 (-0.172)** | 12.384 (-7.406) | **0.142 (-0.092)** | **0.049 (+0.003)** |
| | PointNet | CMD + SB | 0.387 (-0.082) | 27.922 (+0.311) | 0.261 (-0.009) | 0.055 (-0.001) |
| | Transolver | CMD + SB | 0.781 (⋆) | 71.526 (⋆) | 0.507 (⋆) | 0.086 (⋆) |
| | Oracle (GraphSAGE) | DARE-GRAM + TB | 0.192 (-0.172) | 12.384 (-7.406) | 0.142 (-0.092) | 0.049 (+0.003) |
| Motor | GraphSAGE | DARE-GRAM + SB | 0.342 (-0.033) | 29.088 (-0.370) | 0.349 (-0.078) | 0.031 (-0.000) |
| | PointNet | Deep Coral + SB | 0.313 (-0.084) | 26.229 (-4.425) | 0.197 (-0.147) | 0.043 (-0.003) |
| | Transolver | Deep Coral + SB | **0.098 (-0.018)** | **7.269 (-0.729)** | **0.089 (-0.022)** | **0.016 (-0.003)** |
| | Oracle (Transolver) | Deep Coral + TB | 0.098 (-0.018) | 7.266 (-0.732) | 0.089 (-0.022) | 0.016 (-0.002) |

| Dataset | Model | Best UDA Method + Model Selection | All Fields Normalized Avg (-) | Mises Stress (MPa) | Rel Custom Error (-) | Plastic Residual (-) |
|---------|-------|-----------------------------------|-------------------------------|--------------------|----------------------|----------------------|
| Forming | GraphSAGE | DANN + IWV | 0.334 (-0.042) | 52.917 (+6.821) | 5.384 (+2.888) | 0.509 (+0.028) |
| | PointNet | Deep Coral + SB | 0.182 (-0.044) | 31.345 (-0.090) | 1.154 (+0.273) | **0.451 (-0.029)** |
| | Transolver | Deep Coral + DEV | **0.154 (-0.014)** | **24.427 (+1.457)** | **0.806 (+0.199)** | 0.581 (+0.098) |
| | Oracle (Transolver) | CMD + TB | 0.131 (-0.037) | 20.275 (-2.695) | 0.796 (+0.189) | 0.506 (+0.022) |

| Dataset | Model | Best UDA Method + Model Selection | All Fields Normalized Avg (-) | Temperature (K) | Rel Custom Error (-) | BC Violation Velocity (m/s) |
|---------|-------|-----------------------------------|-------------------------------|-----------------|----------------------|------------------------------|
| Heatsink | PointNet | DARE-GRAM + SB | 0.371 (-0.197) | 12.343 (-8.783) | 0.015 (-0.035) | 0.121 (+0.011) |
| | Transolver | Deep Coral + DEV | **0.318 (-0.128)** | **9.081 (-0.639)** | **0.009 (-0.001)** | 0.117 (+0.039) |
| | UPT | Deep Coral + SB | 0.325 (-0.116) | 12.414 (-0.619) | 0.013 (-0.000) | 0.107 (+0.036) |
| | GINO | Deep CORAL + SB | 0.356 (-0.128) | 14.031 (+0.136) | 0.017 (+0.000) | **0.107 (+0.024)** |
| | Oracle (Transolver) | Deep Coral + TB | 0.310 (-0.135) | 8.718 (-1.002) | 0.009 (-0.001) | 0.117 (+0.039) |

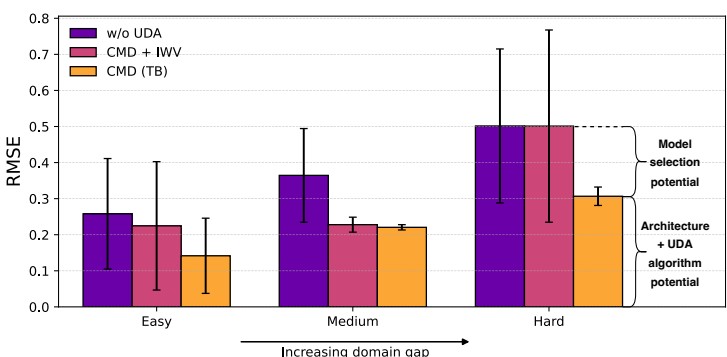

Figure 6: Target error scaling with increasing domain gap. We show the averaged RMSE across all (normalized) fields for the *easy*, *medium*, and *hard* gaps on the *hot rolling* task. We compare models without UDA, the best performing UDA method with unsupervised model selection (CMD + IWV), and the theoretical lower bound (TB). Error bars indicate the standard deviation across four seeds. Furthermore, we highlight potentials of architecture, algorithm and selection improvements on the *hard* task.

below the unregularized error. Figure 6 again highlights the two promising directions for further improvement of the presented baselines: (i) enhancement of neural surrogate architectures and UDA algorithms, and (ii) especially, improvement of unsupervised model selection strategies.

## 6 DISCUSSION

We presented SIMSHIFT, a collection of industry relevant datasets paired with a benchmarking library for comparing UDA algorithms, unsupervised model selection strategies and neural surrogates in real world scenarios. We adapted available techniques, applied them on physical simulation data and performed extensive experiments to evaluate their performance on the presented datasets. Our findings suggest that standard UDA training methods can improve performance of models in unseen parameter ranges in physical simulations, with improvement margins in line with those seen in UDA literature (Dinu et al., 2023; Ragab et al., 2022). Additionally, we find correct unsupervised model selection to be extremely important in downstream model performance on target domains, with it arguably having as much impact as the UDA training itself, which is also in agreement with other DA works (Musgrave et al., 2021).

**Limitations.** We acknowledge that our datasets are limited under two main aspects: (i) They only cover *steady-state* problems, which represent a large portion of industrial simulation tasks. However, an extension with *time-dependent* datasets could be valuable for certain application areas. (ii) They cover a wide range of mesh sizes, ranging from roughly $\mathcal{O}(10^2)$ up to $\mathcal{O}(10^6)$ nodes. Nevertheless, many industrial scenarios require substantially larger meshes. These limitations reflect design choices aimed at benchmarking clarity and computational feasibility and leave room for future extensions.

**Future Directions.** Motivated by our results, we identify several promising research directions: (i) Although we include a diverse and competitive set of UDA algorithms and unsupervised model selection techniques, a wide range of methods remain unexplored in the context of scientific ML. Examples include ensembling based adaptation (Cha et al., 2021), adversarial information bottleneck approaches (Luo et al., 2019; Song et al., 2020) or diffusion based methods (Peng et al., 2024; Liao et al., 2025). In addition, test-time adaptation methods (Wang et al., 2021; Adachi et al., 2025) could be designed and tested using our benchmark. (ii) SIMSHIFT currently evaluates standard UDA algorithms and does not integrate physics constraints (Karniadakis et al., 2021) into training. Our framework and datasets allows to include physics constraints, and we find the direction of a specific physics-inspired UDA method a very interesting and potentially fruitful gap in the current research.

REPRODUCIBILITY STATEMENT

The first step towards reproducibility are the datasets. We provide a download link for all dataset with our predefined domain shifts and a high–level description of each dataset and the splitting strategy in Section 3 and Appendix C. To take transparency a step further, we present detailed descriptions of the respective numerical simulations together with the employed solvers, initial/boundary conditions and the configuration details in Appendix G. Concerning the machine learning parts of the benchmark, we provide detailed descriptions of the architectures, their configurations and training hyperparameters in Appendix F.1. Additionally, we provide an anonymized codebase with pinned dependencies, fixable seeds and a comprehensive `README.md` along with all necessary configuration files used in our benchmarking pipeline.

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

# LLM USAGE DISCLOSURE

In general, LLM tools were used to refine writing in parts of the paper. DeepSeek-R1 and GPT-5 were additionally used to make visualizations prettier, speed up the development of plotting functions, and dump experimental results neatly into latex tabled tables. Beyond that, they were not used to a significant degree in other parts of the code, as neither Copilot nor Cursor are used by the main author. AI assistants were strictly editors and decorators, i.e. they were not involved in ideation, reordering ideas, or at any higher or lower conceptual level.

## A ON NEURAL OPERATORS

One prominent approach in neural surrogate modeling for PDEs is operator learning (Kovachki et al., 2021; Li et al., 2020a; Lu et al., 2021; Alkin et al., 2024a; Li et al., 2020b). In this setting, an operator maps input functions, such as boundary or initial conditions, to the corresponding solution of the PDE. During training, neural operators typically learn from input-output pairs of discretized functions (Kovachki et al., 2021; Li et al., 2020a; Lu et al., 2021; Alkin et al., 2024a). While some methods expect regular, grid based inputs (Li et al., 2020a), others can be applied to any kind of data structure (Alkin et al., 2024a; Li et al., 2020b; 2023b). One notable property is *discretization invariance*, which, along with the ability to handle irregular data, enables generalization across different resolutions and mesh geometries. This is a highly desirable property for industrial simulations (Pfaff et al., 2020; Alkin et al., 2024a; Fürst et al., 2025; Li et al., 2023a; Franco et al., 2022), where non-uniform meshes are the standard due to the computational and modeling advantages. In this work, we focus on domain adaptation rather than benchmarking discretization invariance, and include neural surrogates that may not satisfy this property, such as (Hamilton et al., 2017). Such models have been leveraged in several large scale industrial contexts, including CFD for automotive (Bleeker et al., 2025) or Discrete Element Method (DEM) simulations for industrial processes (Alkin et al., 2024b).

## B DETAILED RESULTS

Complementing the summary in Table 3 of the main paper, the following sections present detailed results for each dataset. For every dataset, we present a complete empirical evaluation of our benchmark that compares the performance for all combinations of models, UDA algorithms and model selection strategies across all output fields.

While these quantitative metrics offer a high level summary of model performance, industry practitioners often need a more fine grained picture to assess the neural surrogate's capabilities under distribution shifts. To address this, we include additional analyses and visualizations alongside the quantitative results. First, we provide error distribution histograms to better illustrate the difficulty of the domain shift occurring in each dataset. Additionally, we present fringe and scatter plots comparing model predictions with the respective ground truth numerical solutions.

### B.1 HOT ROLLING

Table 4 presents the complete benchmarking results for the *hot rolling* dataset.

To gain more insights, we conduct additional analyses on the best performing model, selected based on having the lowest average normalized target domain error across all fields. Figure 7 shows the error distribution of this model and clearly highlights the substantial distribution shift between the source and target domain of the *hot rolling* dataset. Errors in the target domain are noticeably larger, almost up to an order of magnitude higher than those observed in the source domain.

To further illustrate the model's performance, we analyze two representative samples, one from the source and one from the target domain. Since the most critical field for downstream applications is PEEQ, we restrict the following analysis on this scalar field only.

Table 5 presents a summary of the absolute PEEQ prediction errors for the selected source and target samples. Additionally, Figure 8 and Figure 9 visualize the ground truth, predictions, and absolute errors for these samples using fringe plots.

Table 4: RMSE (mean ± std over 4 seeds) on the *hot rolling* dataset at *medium* difficulty. Values are target domain errors (lower is better). Bold marks the overall best model + UDA algorithm + model selection combination. For each architecture, the unregularized baseline row is shaded beige, whereas the best UDA + selection within that architecture is underlined and shaded green. Asterisks denote unstable runs ($\geq 10\times$ the column median).

| Model | DA Algorithm | Model Selection | All Fields Normalized Avg (-) SRC | TGT | Deformation (mm) SRC | TGT | Logarithmic Strain (×10⁻²) SRC | TGT | Equivalent Plastic Strain (×10⁻²) SRC | TGT | Mises Stress (MPa) SRC | TGT | Relative Custom Error (-) SRC | TGT | VM Consistency (-) SRC | TGT |
|---|---|---|---|---|---|---|---|---|---|---|---|---|---|---|---|---|
| | - | - | 0.016(±0.000) | 0.365(±0.130) | 0.525(±0.023) | 5.715(±1.567) | 0.018(±0.000) | 0.997(±0.377) | 0.033(±0.000) | 2.113(±0.789) | 1.972(±0.024) | 19.790(±7.186) | 0.041(±0.000) | 0.234(±0.081) | 0.059(±0.000) | 0.046(±0.002) |
| | DANN | DEV | 0.014(±0.000) | 1.171(±0.058) | 0.599(±0.051) | 17.369(±0.802) | 0.019(±0.001) | 3.451(±0.176) | 0.035(±0.001) | 7.296(±0.405) | 2.076(±0.055) | 110.648(±8.295) | 0.017(±0.003) | 0.730(±0.052) | 0.058(±0.001) | 0.080(±0.023) |
| | DANN | IWV | 0.014(±0.000) | 0.289(±0.147) | 0.561(±0.032) | 5.359(±1.848) | 0.018(±0.000) | 0.792(±0.186) | 0.033(±0.001) | 1.622(±0.306) | 1.992(±0.037) | 24.471(±22.423) | 0.042(±0.002) | 0.183(±0.045) | 0.059(±0.001) | 0.068(±0.042) |
| | DANN | SB | 0.014(±0.000) | 0.692(±0.511) | 0.573(±0.043) | 11.090(±7.161) | 0.018(±0.000) | 2.120(±1.506) | 0.034(±0.001) | 4.510(±3.201) | 1.991(±0.045) | 60.352(±51.358) | 0.043(±0.004) | 0.466(±0.318) | 0.058(±0.001) | 0.055(±0.009) |
| | DANN | TB | 0.014(±0.000) | 0.230(±0.041) | 0.604(±0.010) | 4.640(±0.593) | 0.018(±0.001) | 0.740(±0.134) | 0.034(±0.001) | 1.549(±0.275) | 2.017(±0.047) | 1.740(±3.085) | 0.042(±0.001) | 0.170(±0.026) | 0.059(±0.001) | 0.044(±0.002) |
| | CMD | DEV | 0.015(±0.000) | 1.328(±0.056) | 0.613(±0.045) | 17.690(±1.402) | 0.019(±0.002) | 3.584(±0.244) | 0.036(±0.004) | 7.358(±0.555) | 2.143(±0.175) | 130.494(±23.592) | 0.045(±0.005) | 0.779(±0.049) | 0.059(±0.001) | 0.241(±0.172) |
| | CMD | IWV | 0.014(±0.000) | 0.270(±0.082) | 0.581(±0.026) | 5.227(±1.179) | 0.018(±0.000) | 0.843(±0.195) | 0.033(±0.001) | 1.759(±0.381) | 1.979(±0.038) | 18.785(±8.449) | 0.043(±0.003) | 0.194(±0.043) | 0.058(±0.000) | 0.057(±0.020) |
| GraphSAGE | CMD | SB | 0.014(±0.000) | 0.766(±0.535) | 0.571(±0.040) | 12.160(±7.174) | 0.018(±0.000) | 2.403(±1.541) | 0.033(±0.000) | 5.068(±3.248) | 1.974(±0.034) | 68.509(±55.017) | 0.041(±0.001) | 0.526(±0.327) | 0.058(±0.000) | 0.079(±0.045) |
| | CMD | TB | 0.014(±0.000) | 0.221(±0.007) | 0.583(±0.033) | 4.607(±0.261) | 0.018(±0.000) | 0.711(±0.014) | 0.033(±0.000) | 1.507(±0.045) | 1.992(±0.021) | 14.288(±1.040) | 0.044(±0.001) | 0.165(±0.003) | 0.058(±0.000) | 0.048(±0.001) |
| | DARE-GRAM | DEV | 0.014(±0.000) | 1.022(±0.168) | 0.538(±0.018) | 15.744(±2.302) | 0.019(±0.001) | 3.109(±0.500) | 0.035(±0.001) | 6.573(±1.071) | 2.065(±0.035) | 86.703(±21.972) | 0.045(±0.002) | 0.673(±0.116) | 0.059(±0.001) | 0.156(±0.051) |
| | **DARE-GRAM** | **IWV** | **0.014(±0.000)** | **0.192(±0.029)** | **0.555(±0.026)** | **4.082(±0.597)** | **0.018(±0.000)** | **0.616(±0.103)** | **0.033(±0.000)** | **1.297(±0.221)** | **2.005(±0.018)** | **12.384(±1.900)** | **0.042(±0.000)** | **0.142(±0.022)** | **0.058(±0.001)** | **0.049(±0.001)** |
| | DARE-GRAM | SB | 0.014(±0.000) | 0.237(±0.044) | 0.477(±0.090) | 4.535(±0.809) | 0.018(±0.000) | 0.773(±0.139) | 0.033(±0.000) | 1.624(±0.307) | 1.985(±0.018) | 14.838(±3.624) | 0.041(±0.002) | 0.179(±0.030) | 0.059(±0.000) | 0.047(±0.002) |
| | DARE-GRAM | TB | 0.014(±0.000) | 0.192(±0.029) | 0.555(±0.026) | 4.082(±0.597) | 0.018(±0.000) | 0.616(±0.103) | 0.033(±0.000) | 1.297(±0.221) | 2.005(±0.018) | 12.384(±1.900) | 0.042(±0.000) | 0.142(±0.022) | 0.058(±0.001) | 0.049(±0.001) |
| | Deep Coral | DEV | 0.014(±0.000) | 0.723(±0.302) | 0.560(±0.058) | 11.013(±4.292) | 0.018(±0.000) | 2.202(±0.860) | 0.034(±0.000) | 4.601(±1.848) | 2.002(±0.016) | 61.162(±33.535) | 0.042(±0.002) | 0.490(±0.182) | 0.058(±0.001) | 0.073(±0.019) |
| | Deep Coral | IWV | 0.014(±0.000) | 0.273(±0.068) | 0.583(±0.054) | 5.137(±0.652) | 0.018(±0.000) | 0.841(±0.153) | 0.033(±0.000) | 1.774(±0.304) | 1.980(±0.014) | 19.965(±8.986) | 0.040(±0.001) | 0.203(±0.043) | 0.058(±0.001) | 0.052(±0.010) |
| | Deep Coral | SB | 0.014(±0.000) | 0.511(±0.420) | 0.548(±0.031) | 8.679(±5.518) | 0.018(±0.000) | 1.597(±1.222) | 0.033(±0.000) | 3.385(±2.597) | 1.970(±0.010) | 41.196(±3.492) | 0.041(±0.000) | 0.358(±0.260) | 0.058(±0.000) | 0.056(±0.017) |
| | Deep Coral | TB | 0.014(±0.000) | 0.212(±0.012) | 0.590(±0.045) | 4.547(±0.361) | 0.018(±0.000) | 0.679(±0.050) | 0.033(±0.001) | 1.427(±0.095) | 1.992(±0.028) | 13.829(±0.609) | 0.040(±0.002) | 0.158(±0.014) | 0.059(±0.001) | 0.049(±0.002) |
| | - | - | 0.023(±0.001) | 0.409(±0.055) | 2.240(±0.001) | 11.474(±0.290) | 0.026(±0.001) | 1.225(±0.165) | 0.051(±0.002) | 2.519(±0.385) | 2.860(±0.138) | 27.611(±5.693) | 0.057(±0.002) | 0.270(±0.046) | 0.063(±0.001) | 0.056(±0.011) |
| | DANN | DEV | 0.020(±0.001) | 1.126(±0.047) | 2.237(±0.002) | 18.207(±0.312) | 0.027(±0.001) | 3.377(±0.105) | 0.053(±0.004) | 7.181(±0.218) | 2.955(±0.180) | 106.747(±6.049) | 0.060(±0.003) | 0.727(±0.025) | 0.064(±0.002) | 0.088(±0.020) |
| | DANN | IWV | 0.020(±0.001) | 0.984(±0.424) | 2.241(±0.009) | 17.007(±3.626) | 0.026(±0.002) | 2.959(±1.236) | 0.054(±0.004) | 6.270(±2.634) | 2.967(±0.193) | 91.530(±44.149) | 0.064(±0.005) | 0.631(±0.265) | 0.063(±0.001) | 0.099(±0.034) |
| | DANN | SB | 0.019(±0.001) | 0.951(±0.347) | 2.239(±0.004) | 16.497(±3.351) | 0.027(±0.001) | 2.906(±1.015) | 0.052(±0.003) | 6.165(±2.173) | 2.886(±0.135) | 85.396(±36.953) | 0.063(±0.002) | 0.633(±0.213) | 0.063(±0.000) | 0.105(±0.049) |
| | DANN | TB | 0.020(±0.001) | 0.336(±0.054) | 2.239(±0.002) | 11.137(±0.329) | 0.027(±0.001) | 1.092(±0.206) | 0.052(±0.002) | 2.312(±0.421) | 2.988(±0.138) | 22.461(±4.074) | 0.060(±0.004) | 0.240(±0.047) | 0.064(±0.001) | 0.053(±0.005) |
| | CMD | DEV | 0.020(±0.001) | 1.164(±0.107) | 2.240(±0.002) | 18.411(±1.132) | 0.028(±0.001) | 3.591(±0.305) | 0.054(±0.003) | 7.643(±0.591) | 2.996(±0.178) | 102.004(±14.284) | 0.068(±0.009) | 0.790(±0.066) | 0.061(±0.001) | 0.059(±0.019) |
| | CMD | IWV | 0.020(±0.001) | 1.232(±0.036) | 2.240(±0.002) | 19.099(±0.332) | 0.026(±0.002) | 3.775(±0.107) | 0.055(±0.002) | 8.015(±0.237) | 3.053(±0.149) | 110.674(±4.420) | 0.072(±0.007) | 0.836(±0.039) | 0.061(±0.001) | 0.077(±0.021) |
| PointNet | CMD | SB | 0.019(±0.001) | 0.387(±0.059) | 2.241(±0.002) | 11.327(±0.574) | 0.027(±0.001) | 1.201(±0.297) | 0.051(±0.001) | 2.511(±0.699) | 2.852(±0.079) | 27.922(±6.676) | 0.060(±0.004) | 0.261(±0.084) | 0.061(±0.002) | 0.055(±0.013) |
| | CMD | TB | 0.019(±0.001) | 0.353(±0.078) | 2.240(±0.002) | 11.231(±0.508) | 0.026(±0.000) | 1.147(±0.284) | 0.051(±0.001) | 2.402(±0.634) | 2.843(±0.083) | 23.881(±4.994) | 0.058(±0.003) | 0.249(±0.077) | 0.061(±0.001) | 0.049(±0.005) |
| | DARE-GRAM | DEV | 0.021(±0.001) | 1.217(±0.034) | 2.240(±0.001) | 19.001(±0.305) | 0.029(±0.002) | 3.706(±0.123) | 0.058(±0.004) | 7.872(±0.254) | 3.205(±0.202) | 111.781(±4.997) | 0.076(±0.012) | 0.820(±0.035) | 0.062(±0.002) | 0.084(±0.007) |
| | DARE-GRAM | IWV | 0.019(±0.001) | 0.881(±0.440) | 2.240(±0.001) | 14.978(±4.056) | 0.027(±0.001) | 2.067(±0.830) | 0.051(±0.002) | 4.147(±1.607) | 2.878(±0.116) | 98.547(±75.723) | 0.059(±0.006) | 0.482(±0.249) | 0.062(±0.001) | 0.204(±0.083) |
| | DARE-GRAM | SB | 0.019(±0.001) | 0.773(±0.482) | 2.241(±0.001) | 14.019(±4.369) | 0.026(±0.001) | 1.933(±0.592) | 0.051(±0.001) | 3.948(±0.859) | 2.837(±0.073) | 84.934(±85.428) | 0.058(±0.003) | 0.479(±0.301) | 0.062(±0.001) | 0.117(±0.124) |
| | DARE-GRAM | TB | 0.019(±0.001) | 0.318(±0.068) | 2.238(±0.001) | 11.167(±0.357) | 0.026(±0.001) | 1.002(±0.267) | 0.050(±0.002) | 2.085(±0.571) | 2.870(±0.094) | 22.504(±3.875) | 0.060(±0.003) | 0.225(±0.061) | 0.062(±0.001) | 0.057(±0.008) |
| | Deep Coral | DEV | 0.020(±0.001) | 1.029(±0.106) | 2.239(±0.003) | 17.061(±1.272) | 0.026(±0.001) | 3.064(±0.375) | 0.056(±0.003) | 6.479(±0.889) | 3.027(±0.119) | 95.926(±11.663) | 0.068(±0.010) | 0.678(±0.087) | 0.061(±0.001) | 0.094(±0.008) |
| | Deep Coral | IWV | 0.020(±0.001) | 1.048(±0.167) | 2.238(±0.003) | 17.395(±1.880) | 0.028(±0.001) | 3.077(±0.508) | 0.055(±0.002) | 6.461(±1.120) | 2.984(±0.108) | 100.276(±20.956) | 0.067(±0.004) | 0.661(±0.117) | 0.061(±0.001) | 0.108(±0.010) |
| | Deep Coral | SB | 0.019(±0.000) | 0.977(±0.158) | 2.243(±0.002) | 16.764(±1.497) | 0.027(±0.001) | 2.947(±0.409) | 0.052(±0.001) | 6.257(±0.856) | 2.866(±0.081) | 88.933(±21.502) | 0.060(±0.004) | 0.651(±0.079) | 0.060(±0.001) | 0.094(±0.005) |
| | Deep Coral | TB | 0.019(±0.000) | 0.346(±0.078) | 2.239(±0.003) | 11.099(±0.287) | 0.027(±0.001) | 1.100(±0.270) | 0.051(±0.001) | 2.304(±0.618) | 2.857(±0.089) | 24.024(±6.005) | 0.058(±0.003) | 0.236(±0.061) | 0.063(±0.001) | 0.050(±0.005) |
| | - | - | 0.028(±0.001) | * | 0.580(±0.035) | * | 0.035(±0.001) | * | 0.070(±0.002) | * | 3.534(±0.088) | * | 0.077(±0.003) | * | 0.060(±0.005) | 0.650(±0.562) |
| | DANN | DEV | 0.024(±0.001) | 1.316(±0.047) | 0.566(±0.029) | 19.257(±0.628) | 0.037(±0.002) | 3.826(±0.120) | 0.074(±0.005) | 8.116(±0.248) | 3.534(±0.134) | 135.380(±7.612) | 0.084(±0.011) | 0.819(±0.030) | 0.062(±0.002) | 0.081(±0.007) |
| | DANN | IWV | 0.023(±0.001) | * | 0.563(±0.032) | * | 0.036(±0.002) | * | 0.072(±0.004) | * | 3.505(±0.111) | * | 0.080(±0.006) | * | 0.060(±0.004) | * |
| | DANN | SB | 0.023(±0.000) | * | 0.557(±0.026) | * | 0.035(±0.000) | * | 0.068(±0.001) | * | 3.413(±0.037) | * | 0.078(±0.003) | * | 0.065(±0.002) | * |
| | DANN | TB | 0.023(±0.001) | 1.248(±0.044) | 0.581(±0.052) | 18.346(±0.511) | 0.036(±0.001) | 3.634(±0.123) | 0.072(±0.004) | 7.702(±0.270) | 3.477(±0.130) | 126.738(±6.268) | 0.081(±0.012) | 0.772(±0.028) | 0.062(±0.005) | 0.083(±0.008) |
| | CMD | DEV | 0.024(±0.001) | 1.122(±0.311) | 0.615(±0.033) | 16.889(±3.999) | 0.037(±0.001) | 3.344(±0.794) | 0.073(±0.002) | 7.127(±1.619) | 3.573(±0.097) | 107.289(±44.373) | 0.085(±0.008) | 0.716(±0.151) | 0.059(±0.004) | 0.119(±0.041) |
| | CMD | IWV | 0.024(±0.001) | 2.631(±3.517) | 0.598(±0.040) | 78.602(±130.758) | 0.037(±0.002) | 4.517(±4.473) | 0.073(±0.003) | 12.726(±14.421) | 3.598(±0.107) | 351.111(±537.157) | 0.083(±0.007) | 1.370(±1.618) | 0.062(±0.004) | 0.502(±0.846) |
| Transolver | CMD | SB | 0.023(±0.000) | 0.781(±0.350) | 0.587(±0.024) | 11.988(±4.742) | 0.036(±0.001) | 2.352(±0.967) | 0.071(±0.002) | 4.980(±2.072) | 3.556(±0.090) | 71.526(±40.616) | 0.081(±0.006) | 0.507(±0.201) | 0.062(±0.004) | 0.086(±0.013) |
| | CMD | TB | 0.024(±0.000) | 1.567(±3.137) | 0.615(±0.046) | 9.350(±2.012) | 0.037(±0.002) | 1.798(±0.445) | 0.074(±0.003) | 3.834(±1.010) | 3.593(±0.154) | 41.957(±11.589) | 0.082(±0.004) | 0.405(±0.100) | 0.058(±0.004) | 0.071(±0.023) |
| | DARE-GRAM | DEV | 0.023(±0.001) | 3.153(±2.545) | 0.558(±0.028) | 51.238(±44.025) | 0.036(±0.001) | 8.082(±6.894) | 0.071(±0.003) | 15.250(±12.194) | 3.462(±0.093) | 360.475(±324.276) | 0.078(±0.006) | 1.193(±0.663) | 0.063(±0.002) | 0.322(±0.185) |
| | DARE-GRAM | IWV | 0.023(±0.001) | * | 0.554(±0.020) | * | 0.036(±0.002) | * | 0.070(±0.004) | * | 3.431(±0.071) | * | 0.078(±0.005) | * | 0.059(±0.005) | 0.666(±0.660) |
| | DARE-GRAM | SB | 0.022(±0.000) | * | 0.545(±0.020) | * | 0.034(±0.002) | 22.287(±33.097) | 0.067(±0.002) | 46.255(±69.242) | 3.385(±0.049) | * | 0.073(±0.005) | 3.331(±4.667) | 0.060(±0.005) | * |
| | DARE-GRAM | TB | 0.023(±0.001) | 0.827(±0.112) | 0.556(±0.010) | 12.936(±1.844) | 0.035(±0.002) | 2.619(±0.376) | 0.070(±0.004) | 5.681(±0.772) | 3.415(±0.082) | 62.083(±21.908) | 0.077(±0.004) | 0.570(±0.069) | 0.063(±0.002) | 0.095(±0.020) |
| | Deep Coral | DEV | 0.023(±0.001) | 3.242(±4.897) | 0.596(±0.014) | 92.226(±160.351) | 0.035(±0.002) | 9.242(±13.641) | 0.070(±0.002) | 22.735(±34.969) | 3.475(±0.171) | 254.700(±375.199) | 0.079(±0.005) | 1.587(±2.089) | 0.060(±0.002) | 0.376(±0.598) |
| | Deep Coral | IWV | 0.023(±0.001) | * | 0.580(±0.023) | * | 0.035(±0.002) | * | 0.069(±0.005) | * | 3.439(±0.114) | * | 0.080(±0.005) | 4.597(±4.748) | 0.062(±0.003) | 0.603(±0.492) |
| | Deep Coral | SB | 0.023(±0.001) | 3.611(±4.679) | 0.582(±0.017) | 95.080(±158.462) | 0.034(±0.002) | 10.329(±12.991) | 0.068(±0.004) | 25.707(±33.261) | 3.402(±0.079) | 270.011(±365.767) | 0.076(±0.005) | 1.931(±1.940) | 0.064(±0.003) | 0.813(±0.768) |
| | Deep Coral | TB | 0.023(±0.000) | 0.656(±0.188) | 0.589(±0.021) | 10.202(±2.895) | 0.037(±0.001) | 1.985(±0.589) | 0.072(±0.003) | 4.247(±1.285) | 3.518(±0.053) | 53.782(±18.391) | 0.077(±0.005) | 0.427(±0.119) | 0.061(±0.004) | 0.078(±0.014) |

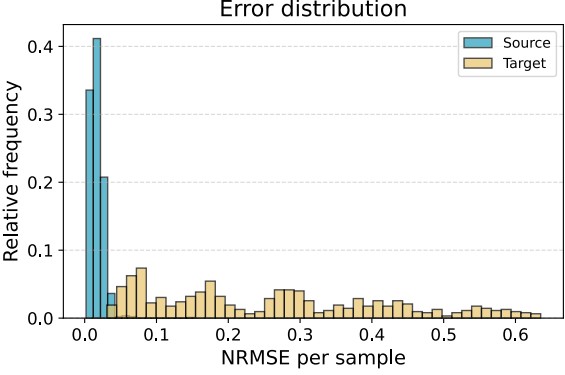

Figure 7: Distribution of NRMSE (averaged across all fields) for the test sets of the source (blue) and target (yellow) domains in the *hot rolling* dataset. Bar height indicates the relative frequency of samples within each bin.

Table 5: Absolute error of PEEQ predictions for representative samples from the source and target domain of the *hot rolling* dataset. Lowest value per metric is bold.

| Metric | Source | Target |
|---|---|---|
| Mean | **2.07e-04** | 1.46e-02 |
| Std | **1.87e-04** | 2.73e-03 |
| Median | **1.66e-04** | 1.49e-02 |
| $Q_{01}$ | **7.45e-09** | 6.82e-03 |
| $Q_{25}$ | **6.38e-05** | 1.37e-02 |
| $Q_{75}$ | **2.99e-04** | 1.58e-02 |
| $Q_{99}$ | **7.61e-04** | 2.14e-02 |

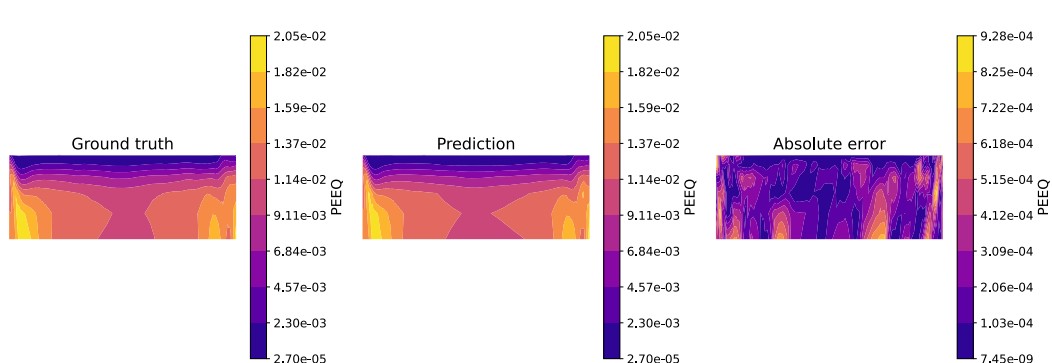

Figure 8: Fringe plot of the *hot rolling* dataset (representative source sample). Shown is the ground truth (left) and predicted (middle) PEEQ, as well as the absolute error (right).

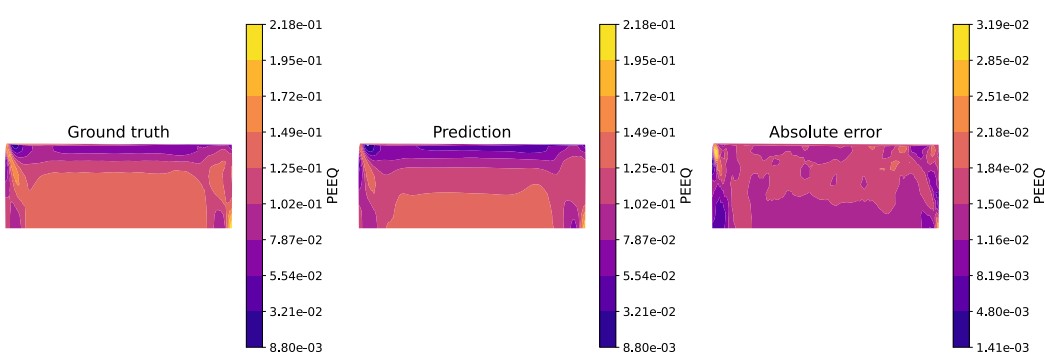

Figure 9: Fringe plot of the *hot rolling* dataset (representative target sample). Shown is the ground truth (left) and predicted (middle) PEEQ, as well as the absolute error (right).

## B.2 SHEET METAL FORMING

In contrast to the substantial shift observed in the hot rolling dataset, the distribution shift in the *sheet metal forming* dataset is moderate. Table 6 presents the detailed performance across all models, algorithms, and selections for this dataset.

Table 6: RMSE (mean ± std over 4 seeds) on the *sheet metal forming* dataset at *medium* difficulty. Values are target domain errors (lower is better). Bold marks the overall best model + UDA algorithm + model selection combination. For each architecture, the unregularized baseline row is shaded beige, whereas the best UDA + selection within that architecture is underlined and shaded green.

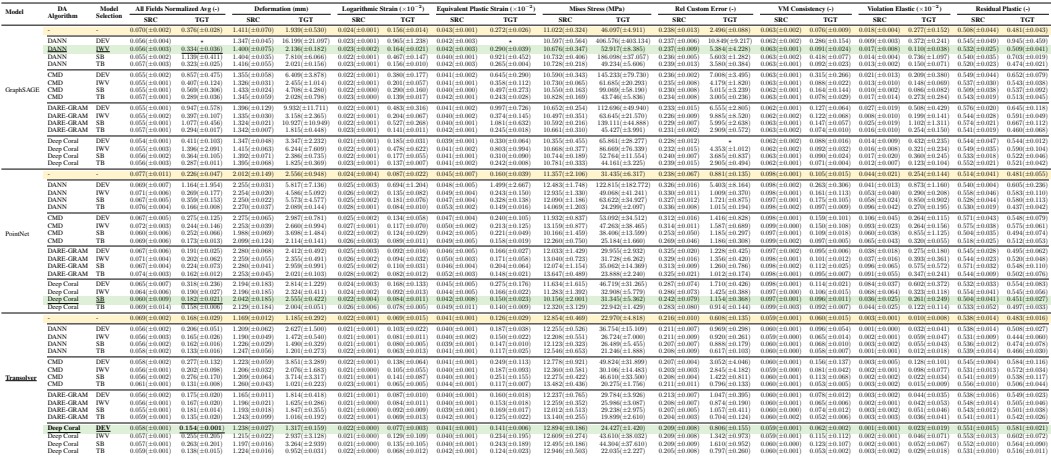

To further illustrate model behavior under distribution shift, we examine the best performing model, selected by lowest normalized average target domain error. The error distribution (Figure 10) shows a moderate distribution shift between the source and target domain with some outliers in the target domain.

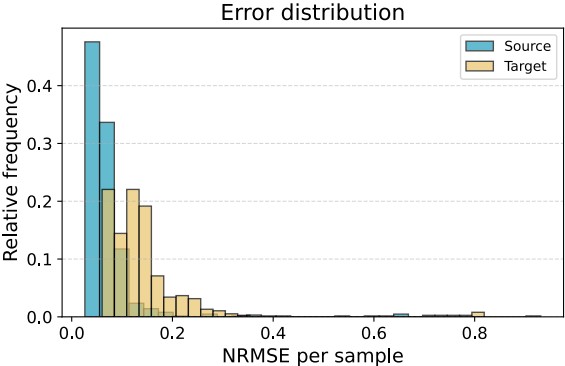

Figure 10: Distribution of NRMSE (averaged across all fields) for the test sets of the source (blue) and target (yellow) domains in the *sheet metal forming* dataset. Bar height indicates the relative frequency of samples within each bin.

To better understand the model's predictive behavior in this setting, we analyze best- and worst-case examples in each domain, again focusing on the critical PEEQ field. Table 7 provides a statistical summary of the absolute PEEQ prediction errors across the selected cases. Fringe plots in Figures 11 to 14 provide a visual understanding of model accuracy. These visualizations emphasize that while the best prediction in the target domain remains reasonably accurate, others (e.g., the worst case sample) exhibit notable discrepancies in the localized regions around the bends that we are most interested in.

Table 7: Absolute error of PEEQ predictions for the best and worst samples from the source and target domain of the *sheet metal forming* dataset. Lowest value per metric is bold.

| Metric | Source | | Target | |
|---|---|---|---|---|
| | **Best** | **Worst** | **Best** | **Worst** |
| Mean | **5.47e-05** | 1.68e-04 | 1.72e-04 | 1.86e-03 |
| Std | **1.35e-04** | 3.80e-04 | 4.52e-04 | 5.97e-03 |
| Median | **1.96e-05** | 4.35e-05 | 5.58e-05 | 2.85e-04 |
| $Q_{01}$ | **2.43e-07** | 6.52e-07 | 1.03e-06 | 6.66e-06 |
| $Q_{25}$ | **7.49e-06** | 1.84e-05 | 2.60e-05 | 1.53e-04 |
| $Q_{75}$ | **4.05e-05** | 1.05e-04 | 9.10e-05 | 4.67e-04 |
| $Q_{99}$ | **7.37e-04** | 1.96e-03 | 2.56e-03 | 3.40e-02 |

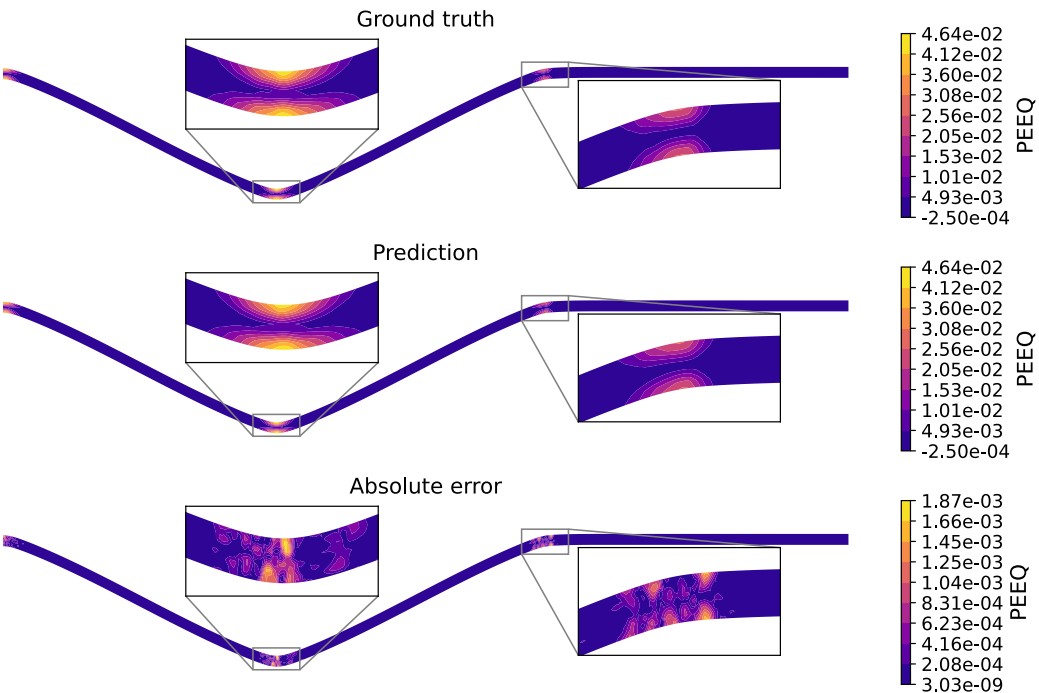

Figure 11: Fringe plot of the *sheet metal forming* dataset (best source sample). Shown is the ground truth (top) and predicted (middle) PEEQ, aswell as the absolute error (bottom).

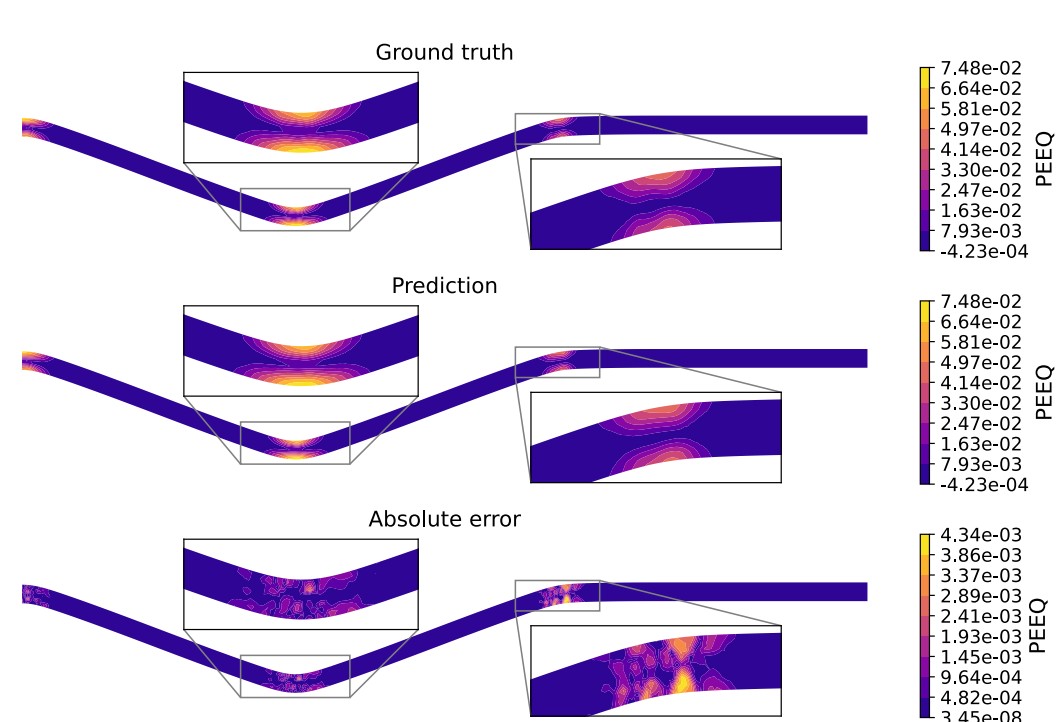

Figure 12: Fringe plot of the *sheet metal forming* dataset (worst source sample). Shown is the ground truth (top) and predicted (middle) PEEQ, as well as the absolute error (bottom).

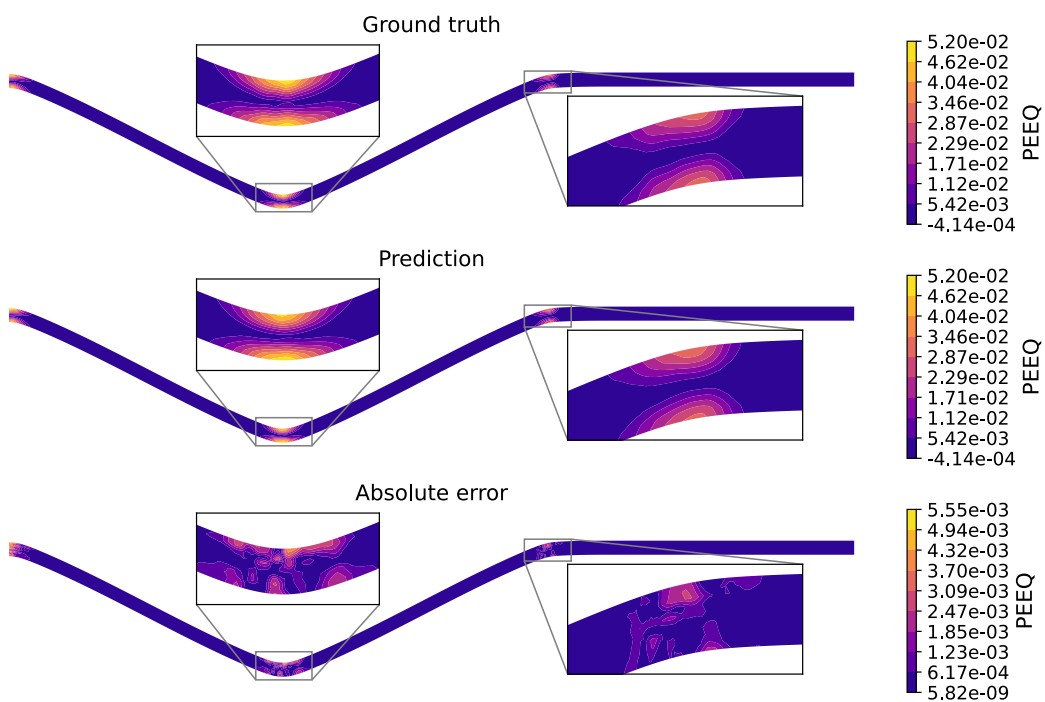

Figure 13: Fringe plot of the *sheet metal forming* dataset (best target sample). Shown is the ground truth (top) and predicted (middle) PEEQ, as well as the absolute error (bottom).

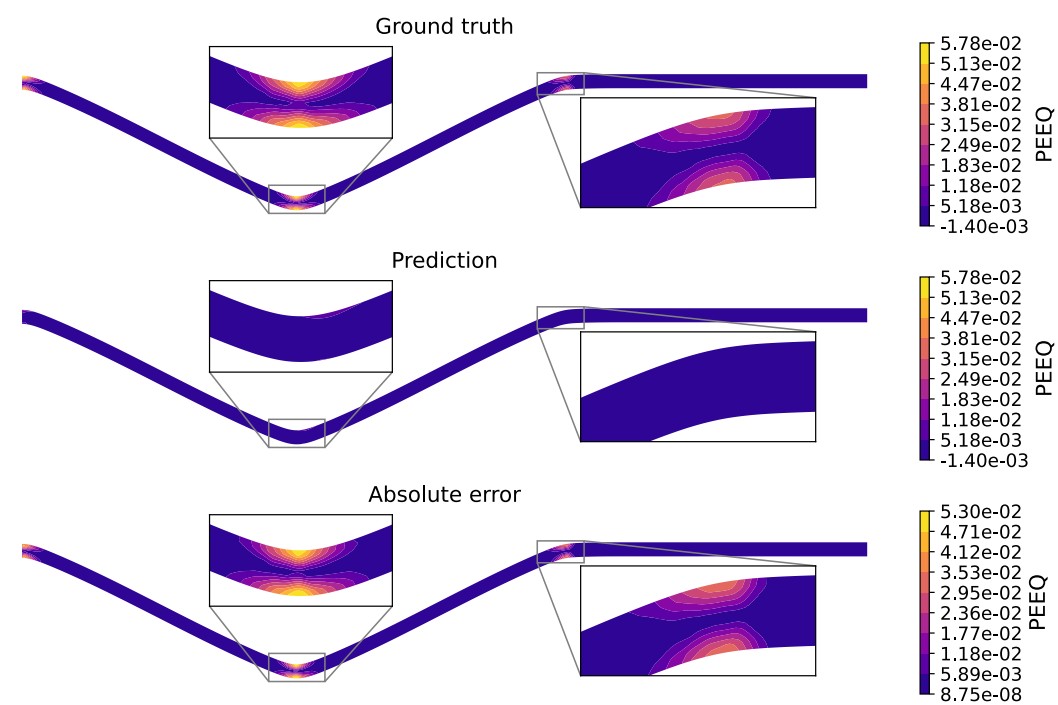

Figure 14: Fringe plot of the *sheet metal forming* dataset (worst target sample). Shown is the ground truth (top) and predicted (middle) PEEQ, as well as the absolute error (bottom).

### B.3 ELECTRIC MOTOR DESIGN

Table 8 presents the complete benchmarking results for the *electric motor design* dataset. For this dataset the relative degradation in model performance in the target domain is in general smaller than in the previous two presented above.

Table 8: RMSE (mean ± std over 4 seeds) on the *electric motor design* dataset at *medium* difficulty. Values are target domain errors (lower is better). Bold marks the overall best model + UDA algorithm + model selection combination. For each architecture, the unregularized baseline row is shaded beige, whereas the best UDA + selection within that architecture is underlined and shaded green.

| Model | DA Algorithm | Model Selection | All Fields Normalized Avg (-) | | Deformation (m) | | Logarithmic Strain (×10⁻²) | | Principal Strain (×10⁻²) | | Cauchy Stress (MPa) | | Mises Stress (MPa) | | Total Strain (×10⁻²) | | Rel Custom Error (-) | | VM Consistency (-) | |
|---|---|---|---|---|---|---|---|---|---|---|---|---|---|---|---|---|---|---|---|---|
| | | | SRC | TGT | SRC | TGT | SRC | TGT | SRC | TGT | SRC | TGT | SRC | TGT | SRC | TGT | SRC | TGT | SRC | TGT |

To assess the effect of the domain shift on prediction accuracy in the *electric motor design* dataset further, Figure 15 shows the distribution of NRMSEs for the best performing model, selected by lowest average error in the target domain, in the source and target domain.

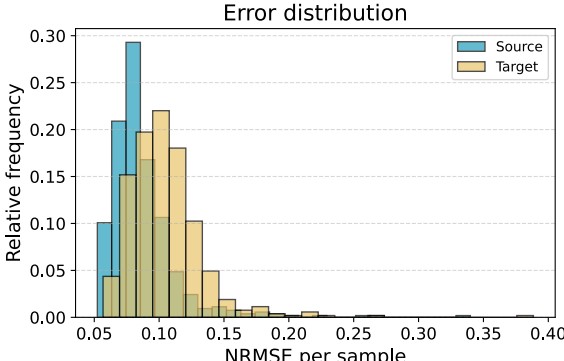

Figure 15: Distribution of NRMSE (averaged across all fields) for the test sets of the source (blue) and target (yellow) domains in the *electric motor design* dataset. Bar height indicates the relative frequency of samples within each bin.

In this task, the Mises stress is used as a scalar summary of the multi-axial stress state and is particularly interesting for downstream analysis and optimization. We therefore focus our closer inspection on this field.

Table 9 presents a comparison of absolute Mises stress errors for the best and worst samples from both the source and target test sets. The corresponding fringe plots are shown in Figures 16 to 19, comparing the ground truth and predicted fields alongside their absolute errors. They show that the best samples are predicted very well, whereas the worst sample of the source domain visually appears slightly worse than the one of the target domain. On average, however, it is still predicted more accurately than the worst sample of the target domain, as shown in Table 9.

Table 9: Absolute error (MPa) of Mises stress predictions for the best and worst samples from the source and target domain of the *electric motor design* dataset. Lowest value per metric is bold.

| Metric | Source | | Target | |
|---|---|---|---|---|
| | **Best** | **Worst** | **Best** | **Worst** |
| Mean | **2.00** | 20.50 | 2.67 | 23.21 |
| Std | **2.73** | 39.50 | 3.13 | 23.32 |
| Median | **1.26** | 13.63 | 1.68 | 13.09 |
| $Q_{01}$ | **0.02** | 0.18 | 0.03 | 0.11 |
| $Q_{25}$ | **0.60** | 5.08 | 0.75 | 2.87 |
| $Q_{75}$ | **2.24** | 24.05 | 3.50 | 41.97 |
| $Q_{99}$ | **13.43** | 140.17 | 15.05 | 78.28 |

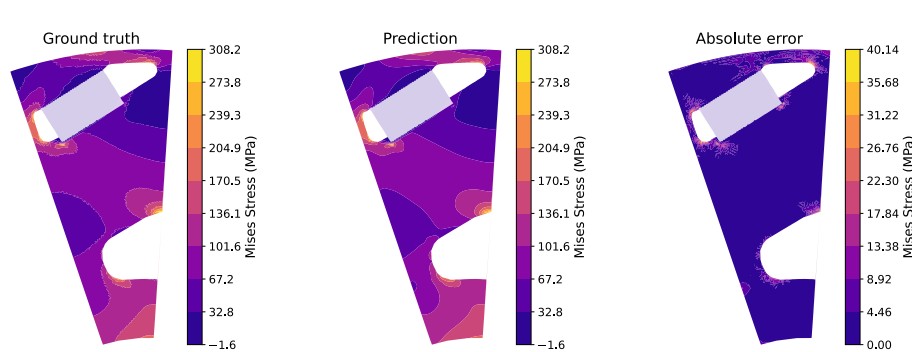

Figure 16: Fringe plot of the *electric motor design* dataset (best source sample). Shown is the ground truth (left) and predicted (middle) Mises stress, as well as the absolute error (right).

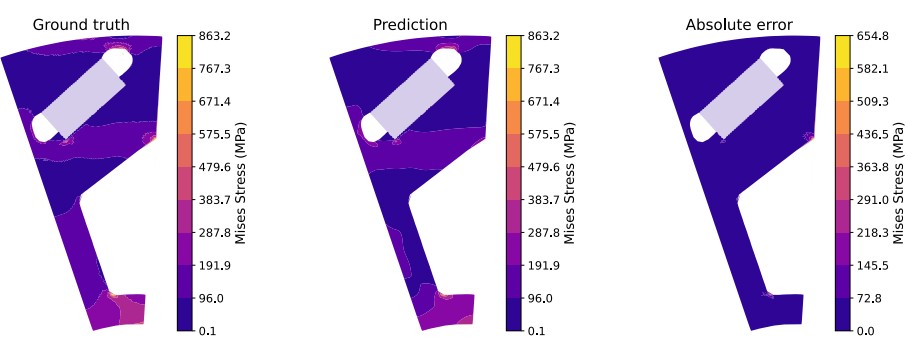

Figure 17: Fringe plot of the *electric motor design* dataset (worst source sample). Shown is the ground truth (left) and predicted (middle) Mises stress, as well as the absolute error (right).

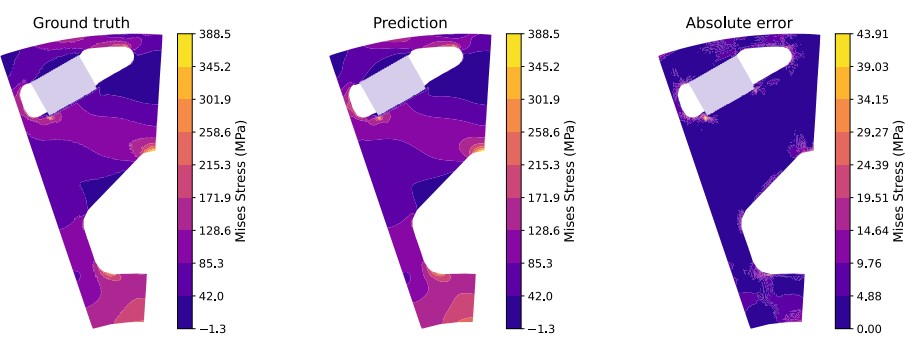

Figure 18: Fringe plot of the *electric motor design* dataset (best target sample). Shown is the ground truth (left) and predicted (middle) Mises stress, as well as the absolute error (right).

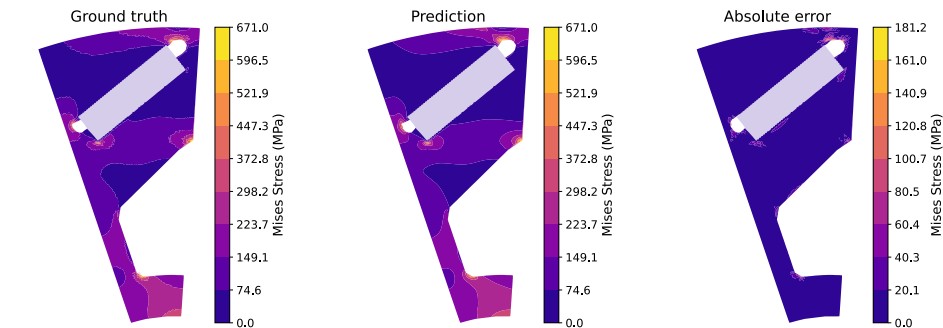

Figure 19: Fringe plot of the *electric motor design* dataset (worst target sample). Shown is the ground truth (left) and predicted (middle) Mises stress, as well as the absolute error (right).

## B.4 HEATSINK DESIGN

Table 10 presents the complete benchmarking results for the *heatsink design* dataset.

Table 10: RMSE (mean $\pm$ std over 4 seeds) on the *heatsink* dataset at *medium* difficulty. Values are target domain errors (lower is better). Bold marks the overall best model + UDA algorithm + model selection combination. For each architecture, the unregularized baseline row is shaded beige, whereas the best UDA + selection within that architecture is underlined and shaded green.

| Model | DA Algorithm | Model Selection | All Fields Normalized Avg (-) SRC | TGT | Temperature (K) SRC | TGT | Velocity (m/s) SRC | TGT | Pressure (kPa) SRC | TGT | Rel Custom Error (-) SRC | TGT | BC Violation Temperature (-) SRC | TGT | BC Violation Velocity (m/s) SRC | TGT |
|---|---|---|---|---|---|---|---|---|---|---|---|---|---|---|---|---|
| PointNet | - | - | 0.525(±0.026) | 0.568(±0.030) | 15.581(±1.535) | 21.126(±2.365) | 0.054(±0.002) | 0.044(±0.000) | 385.602(±34.377) | 1879.390(±239.203) | 0.036(±0.006) | 0.050(±0.009) | 0.257(±0.026) | 0.265(±0.020) | 0.105(±0.007) | 0.110(±0.008) |
| | DANN | DEV | 0.339(±0.104) | 0.442(±0.050) | 12.078(±4.555) | 19.408(±3.391) | 0.043(±0.009) | 0.047(±0.007) | 814.988(±1032.338) | 1997.691(±359.872) | 0.026(±0.014) | 0.041(±0.012) | 0.176(±0.085) | 0.261(±0.042) | 0.088(±0.031) | 0.107(±0.015) |
| | DANN | IWV | 0.289(±0.056) | 0.429(±0.052) | 10.167(±2.894) | 18.172(±3.222) | 0.040(±0.008) | 0.047(±0.007) | 283.202(±70.570) | 1805.938(±144.882) | 0.019(±0.008) | 0.036(±0.009) | 0.145(±0.064) | 0.248(±0.042) | 0.079(±0.026) | 0.114(±0.012) |
| | DANN | SB | 0.298(±0.016) | 0.494(±0.028) | 6.668(±1.013) | 20.129(±2.380) | 0.031(±0.002) | 0.055(±0.002) | 207.063(±13.650) | 2102.712(±614.979) | 0.010(±0.002) | 0.041(±0.008) | 0.063(±0.018) | 0.286(±0.023) | 0.047(±0.010) | 0.128(±0.007) |
| | DANN | TB | 0.304(±0.036) | 0.397(±0.019) | 10.964(±1.411) | 15.719(±1.387) | 0.041(±0.005) | 0.043(±0.002) | 331.207(±140.774) | 1907.682(±232.159) | 0.021(±0.004) | 0.029(±0.006) | 0.159(±0.027) | 0.212(±0.021) | 0.086(±0.011) | 0.114(±0.010) |
| | CMD | DEV | 0.424(±0.003) | 0.443(±0.003) | 16.292(±0.119) | 20.676(±0.141) | 0.042(±0.001) | 0.042(±0.001) | 2388.245(±17.831) | 2464.906(±43.671) | 0.042(±0.000) | 0.048(±0.001) | 0.272(±0.002) | 0.286(±0.009) | 0.080(±0.002) | 0.109(±0.003) |
| | CMD | IWV | 0.239(±0.008) | 0.480(±0.020) | 7.577(±0.479) | 18.524(±1.213) | 0.033(±0.001) | 0.051(±0.002) | 192.803(±4.658) | 2455.172(±118.483) | 0.012(±0.001) | 0.035(±0.003) | 0.082(±0.009) | 0.246(±0.012) | 0.055(±0.004) | 0.130(±0.002) |
| | CMD | SB | 0.238(±0.007) | 0.477(±0.023) | 7.483(±0.406) | 18.666(±1.057) | 0.033(±0.001) | 0.051(±0.002) | 196.144(±5.022) | 2375.674(±157.350) | 0.012(±0.001) | 0.035(±0.003) | 0.080(±0.008) | 0.246(±0.012) | 0.054(±0.003) | 0.129(±0.002) |
| | CMD | TB | 0.302(±0.096) | 0.442(±0.018) | 10.801(±4.087) | 17.800(±2.256) | 0.037(±0.004) | 0.046(±0.004) | 756.939(±1076.860) | 2288.916(±108.201) | 0.022(±0.014) | 0.036(±0.009) | 0.154(±0.087) | 0.234(±0.033) | 0.069(±0.014) | 0.121(±0.010) |
| | DARE-GRAM | DEV | 0.220(±0.009) | 0.393(±0.003) | 6.286(±0.618) | 14.444(±0.800) | 0.030(±0.001) | 0.045(±0.001) | 202.806(±6.781) | 1556.506(±260.868) | 0.009(±0.001) | 0.020(±0.004) | 0.057(±0.015) | 0.215(±0.013) | 0.043(±0.008) | 0.125(±0.002) |
| | DARE-GRAM | IWV | 0.219(±0.009) | 0.378(±0.027) | 6.196(±0.632) | 13.346(±1.525) | 0.030(±0.001) | 0.044(±0.003) | 201.887(±6.192) | 1473.618(±311.372) | 0.009(±0.001) | 0.018(±0.003) | 0.055(±0.015) | 0.210(±0.012) | 0.043(±0.008) | 0.122(±0.007) |
| | **DARE-GRAM** | **SB** | 0.214(±0.003) | 0.371(±0.031) | 5.887(±0.161) | 12.343(±1.062) | 0.029(±0.000) | 0.043(±0.004) | 200.738(±7.351) | 1401.808(±235.957) | 0.009(±0.000) | 0.015(±0.002) | 0.048(±0.003) | 0.215(±0.016) | 0.038(±0.002) | 0.121(±0.007) |
| | DARE-GRAM | TB | 0.218(±0.009) | 0.368(±0.027) | 6.155(±0.662) | 12.800(±1.537) | 0.030(±0.001) | 0.042(±0.003) | 202.412(±6.875) | 1398.169(±230.128) | 0.009(±0.001) | 0.018(±0.003) | 0.055(±0.015) | 0.208(±0.013) | 0.042(±0.009) | 0.121(±0.007) |
| | Deep Coral | DEV | 0.275(±0.070) | 0.394(±0.048) | 9.313(±3.544) | 17.959(±2.281) | 0.038(±0.010) | 0.044(±0.006) | 238.834(±83.989) | 1001.401(±504.884) | 0.017(±0.010) | 0.033(±0.012) | 0.125(±0.081) | 0.282(±0.015) | 0.069(±0.028) | 0.109(±0.014) |
| | Deep Coral | IWV | 0.270(±0.061) | 0.394(±0.048) | 9.071(±3.069) | 17.428(±1.939) | 0.037(±0.009) | 0.044(±0.006) | 224.078(±54.512) | 1037.166(±573.817) | 0.016(±0.008) | 0.031(±0.010) | 0.117(±0.067) | 0.276(±0.024) | 0.070(±0.029) | 0.112(±0.010) |
| | Deep Coral | SB | 0.270(±0.061) | 0.394(±0.048) | 9.071(±3.069) | 17.428(±1.939) | 0.037(±0.009) | 0.044(±0.006) | 224.078(±54.512) | 1037.166(±573.817) | 0.016(±0.008) | 0.031(±0.010) | 0.117(±0.067) | 0.276(±0.024) | 0.070(±0.029) | 0.112(±0.010) |
| | Deep Coral | TB | 0.343(±0.063) | 0.384(±0.042) | 12.763(±3.067) | 18.517(±2.502) | 0.047(±0.009) | 0.042(±0.004) | 324.404(±102.861) | 1438.782(±427.051) | 0.027(±0.009) | 0.040(±0.012) | 0.205(±0.075) | 0.276(±0.021) | 0.098(±0.024) | 0.095(±0.011) |
| Transolver | - | - | 0.245(±0.004) | 0.446(±0.018) | 4.327(±0.059) | 9.720(±0.272) | 0.024(±0.000) | 0.038(±0.001) | 307.197(±32.557) | 1686.316(±263.085) | 0.007(±0.000) | 0.010(±0.001) | 0.035(±0.002) | 0.083(±0.007) | 0.021(±0.002) | 0.078(±0.004) |
| | DANN | DEV | 0.183(±0.001) | 0.436(±0.011) | 4.318(±0.081) | 15.272(±0.927) | 0.024(±0.000) | 0.049(±0.001) | 302.659(±26.598) | 2032.574(±47.931) | 0.007(±0.000) | 0.023(±0.002) | 0.035(±0.004) | 0.211(±0.008) | 0.021(±0.001) | 0.131(±0.002) |
| | DANN | IWV | 0.213(±0.056) | 0.425(±0.023) | 5.965(±2.508) | 14.712(±1.183) | 0.028(±0.007) | 0.047(±0.004) | 374.773(±149.925) | 2164.583(±190.783) | 0.011(±0.006) | 0.023(±0.002) | 0.067(±0.060) | 0.204(±0.007) | 0.036(±0.027) | 0.128(±0.005) |
| | DANN | SB | 0.183(±0.002) | 0.422(±0.015) | 4.335(±0.077) | 13.936(±1.133) | 0.024(±0.000) | 0.047(±0.002) | 294.823(±24.069) | 2100.219(±65.406) | 0.007(±0.000) | 0.021(±0.002) | 0.035(±0.005) | 0.193(±0.010) | 0.021(±0.001) | 0.131(±0.002) |
| | DANN | TB | 0.242(±0.068) | 0.406(±0.024) | 7.477(±3.603) | 14.114(±0.341) | 0.032(±0.009) | 0.045(±0.003) | 436.249(±137.337) | 2067.819(±155.414) | 0.014(±0.008) | 0.023(±0.003) | 0.093(±0.066) | 0.207(±0.006) | 0.051(±0.035) | 0.125(±0.004) |
| | CMD | DEV | 0.312(±0.088) | 0.437(±0.041) | 11.189(±4.591) | 16.986(±3.498) | 0.029(±0.004) | 0.043(±0.002) | 1886.108(±1078.918) | 2700.699(±448.359) | 0.030(±0.015) | 0.038(±0.014) | 0.233(±0.136) | 0.232(±0.040) | 0.036(±0.014) | 0.116(±0.006) |
| | CMD | IWV | 0.181(±0.001) | 0.375(±0.032) | 4.266(±0.120) | 11.923(±2.496) | 0.024(±0.000) | 0.042(±0.004) | 274.272(±9.637) | 1816.616(±31.411) | 0.007(±0.000) | 0.015(±0.006) | 0.031(±0.004) | 0.181(±0.012) | 0.021(±0.002) | 0.124(±0.007) |
| | CMD | SB | 0.180(±0.001) | 0.413(±0.044) | 4.281(±0.109) | 13.836(±2.825) | 0.024(±0.000) | 0.047(±0.005) | 300.372(±28.729) | 1960.714(±199.356) | 0.007(±0.000) | 0.021(±0.008) | 0.034(±0.003) | 0.189(±0.015) | 0.021(±0.002) | 0.126(±0.005) |
| | CMD | TB | 0.180(±0.002) | 0.367(±0.025) | 4.232(±0.135) | 11.217(±2.086) | 0.024(±0.000) | 0.041(±0.003) | 260.544(±21.474) | 1894.720(±185.168) | 0.007(±0.000) | 0.014(±0.005) | 0.030(±0.003) | 0.176(±0.008) | 0.021(±0.002) | 0.124(±0.007) |
| | DARE-GRAM | DEV | 0.181(±0.003) | 0.346(±0.038) | 4.255(±0.051) | 10.142(±1.074) | 0.024(±0.000) | 0.039(±0.003) | 292.275(±21.590) | 1725.831(±344.902) | 0.007(±0.000) | 0.010(±0.001) | 0.034(±0.002) | 0.173(±0.013) | 0.021(±0.001) | 0.119(±0.002) |
| | DARE-GRAM | IWV | 0.181(±0.003) | 0.333(±0.030) | 4.290(±0.086) | 9.856(±1.025) | 0.024(±0.000) | 0.038(±0.003) | 280.252(±13.840) | 1627.061(±277.975) | 0.007(±0.000) | 0.011(±0.001) | 0.035(±0.002) | 0.174(±0.016) | 0.023(±0.002) | 0.118(±0.003) |
| | DARE-GRAM | SB | 0.180(±0.003) | 0.354(±0.026) | 4.268(±0.127) | 10.753(±1.187) | 0.024(±0.000) | 0.040(±0.003) | 285.610(±15.083) | 1768.578(±187.647) | 0.007(±0.000) | 0.012(±0.003) | 0.033(±0.003) | 0.176(±0.002) | 0.022(±0.002) | 0.120(±0.006) |
| | DARE-GRAM | TB | 0.180(±0.003) | 0.332(±0.028) | 4.345(±0.064) | 9.840(±0.983) | 0.024(±0.000) | 0.038(±0.003) | 282.034(±17.742) | 1649.841(±294.017) | 0.007(±0.000) | 0.010(±0.001) | 0.034(±0.002) | 0.171(±0.011) | 0.023(±0.001) | 0.119(±0.005) |
| | **Deep Coral** | **DEV** | 0.181(±0.001) | 0.318(±0.027) | 4.251(±0.073) | 9.081(±0.980) | 0.024(±0.000) | 0.037(±0.002) | 284.773(±13.081) | 1425.648(±323.586) | 0.007(±0.000) | 0.009(±0.002) | 0.031(±0.003) | 0.170(±0.010) | 0.022(±0.001) | 0.117(±0.004) |
| | Deep Coral | IWV | 0.182(±0.001) | 0.337(±0.010) | 4.309(±0.037) | 9.993(±0.208) | 0.024(±0.000) | 0.038(±0.000) | 287.654(±13.792) | 1663.737(±219.883) | 0.007(±0.000) | 0.010(±0.001) | 0.034(±0.002) | 0.183(±0.002) | 0.022(±0.001) | 0.122(±0.003) |
| | Deep Coral | SB | 0.180(±0.002) | 0.341(±0.021) | 4.269(±0.039) | 9.982(±0.946) | 0.024(±0.000) | 0.039(±0.002) | 284.878(±11.282) | 1712.116(±272.961) | 0.007(±0.000) | 0.010(±0.002) | 0.032(±0.001) | 0.181(±0.011) | 0.022(±0.000) | 0.119(±0.003) |
| | Deep Coral | TB | 0.181(±0.001) | 0.310(±0.020) | 4.253(±0.073) | 8.718(±0.833) | 0.024(±0.000) | 0.037(±0.002) | 290.683(±22.141) | 1334.387(±174.740) | 0.007(±0.000) | 0.009(±0.002) | 0.033(±0.005) | 0.166(±0.002) | 0.021(±0.001) | 0.117(±0.004) |
| UPT | - | - | 0.244(±0.002) | 0.441(±0.024) | 4.316(±0.028) | 13.033(±1.059) | 0.025(±0.000) | 0.040(±0.002) | 231.716(±14.316) | 816.181(±48.995) | 0.006(±0.000) | 0.013(±0.004) | 0.028(±0.002) | 0.127(±0.025) | 0.026(±0.002) | 0.071(±0.006) |
| | DANN | DEV | 0.188(±0.011) | 0.446(±0.026) | 4.651(±0.781) | 15.580(±0.609) | 0.026(±0.002) | 0.050(±0.003) | 222.968(±13.096) | 2164.527(±302.095) | 0.007(±0.002) | 0.024(±0.003) | 0.034(±0.018) | 0.215(±0.013) | 0.032(±0.014) | 0.122(±0.003) |
| | DANN | IWV | 0.194(±0.012) | 0.468(±0.038) | 5.039(±0.887) | 15.283(±1.155) | 0.026(±0.002) | 0.050(±0.002) | 241.463(±18.499) | 2693.651(±570.114) | 0.008(±0.002) | 0.023(±0.007) | 0.044(±0.019) | 0.222(±0.017) | 0.039(±0.014) | 0.122(±0.002) |
| | DANN | SB | 0.184(±0.002) | 0.480(±0.018) | 4.285(±0.072) | 15.689(±0.806) | 0.025(±0.000) | 0.051(±0.001) | 243.566(±23.766) | 2729.453(±516.783) | 0.006(±0.000) | 0.023(±0.003) | 0.025(±0.003) | 0.225(±0.002) | 0.025(±0.002) | 0.125(±0.003) |
| | DANN | TB | 0.273(±0.092) | 0.398(±0.038) | 9.411(±4.841) | 15.644(±4.334) | 0.037(±0.012) | 0.043(±0.004) | 285.494(±72.700) | 1872.493(±365.646) | 0.019(±0.013) | 0.027(±0.011) | 0.136(±0.096) | 0.222(±0.040) | 0.074(±0.032) | 0.108(±0.018) |
| | CMD | DEV | 0.210(±0.055) | 0.406(±0.046) | 5.994(±3.353) | 14.299(±2.054) | 0.028(±0.007) | 0.046(±0.005) | 236.012(±21.869) | 1873.691(±394.369) | 0.010(±0.008) | 0.021(±0.006) | 0.068(±0.061) | 0.202(±0.019) | 0.039(±0.025) | 0.123(±0.005) |
| | CMD | IWV | 0.182(±0.000) | 0.363(±0.015) | 4.297(±0.038) | 12.908(±0.487) | 0.025(±0.000) | 0.043(±0.001) | 221.371(±9.451) | 1365.372(±256.867) | 0.006(±0.000) | 0.017(±0.001) | 0.028(±0.003) | 0.204(±0.020) | 0.026(±0.001) | 0.118(±0.001) |
| | CMD | SB | 0.179(±0.001) | 0.444(±0.010) | 4.135(±0.026) | 16.130(±0.427) | 0.024(±0.000) | 0.050(±0.001) | 230.955(±7.948) | 1914.677(±52.455) | 0.006(±0.000) | 0.021(±0.001) | 0.020(±0.001) | 0.220(±0.006) | 0.022(±0.000) | 0.126(±0.001) |
| | CMD | TB | 0.182(±0.000) | 0.363(±0.015) | 4.297(±0.038) | 12.908(±0.487) | 0.025(±0.000) | 0.043(±0.001) | 221.371(±9.451) | 1365.372(±256.867) | 0.006(±0.000) | 0.017(±0.001) | 0.028(±0.003) | 0.204(±0.020) | 0.026(±0.001) | 0.118(±0.001) |
| | DARE-GRAM | DEV | 0.181(±0.001) | 0.335(±0.006) | 4.215(±0.052) | 11.291(±0.774) | 0.024(±0.000) | 0.039(±0.001) | 236.908(±22.992) | 1079.205(±251.854) | 0.006(±0.000) | 0.008(±0.002) | 0.025(±0.003) | 0.227(±0.017) | 0.025(±0.002) | 0.115(±0.003) |
| | DARE-GRAM | IWV | 0.182(±0.001) | 0.349(±0.006) | 4.271(±0.033) | 13.077(±0.930) | 0.025(±0.000) | 0.041(±0.001) | 225.229(±2.040) | 1007.086(±192.883) | 0.006(±0.000) | 0.011(±0.003) | 0.026(±0.003) | 0.250(±0.017) | 0.025(±0.001) | 0.113(±0.004) |
| | DARE-GRAM | SB | 0.182(±0.001) | 0.331(±0.013) | 4.310(±0.113) | 12.065(±1.734) | 0.025(±0.000) | 0.039(±0.002) | 242.949(±17.732) | 1050.490(±240.188) | 0.006(±0.000) | 0.011(±0.003) | 0.028(±0.003) | 0.231(±0.021) | 0.026(±0.002) | 0.113(±0.004) |
| | DARE-GRAM | TB | 0.181(±0.001) | 0.326(±0.009) | 4.261(±0.117) | 12.270(±0.971) | 0.025(±0.000) | 0.039(±0.002) | 229.463(±15.792) | 867.115(±236.837) | 0.006(±0.000) | 0.010(±0.003) | 0.025(±0.004) | 0.241(±0.006) | 0.025(±0.002) | 0.110(±0.005) |
| | Deep Coral | DEV | 0.183(±0.001) | 0.345(±0.013) | 4.318(±0.067) | 13.290(±0.655) | 0.025(±0.000) | 0.041(±0.001) | 221.346(±8.107) | 809.557(±99.406) | 0.006(±0.000) | 0.010(±0.002) | 0.028(±0.002) | 0.244(±0.010) | 0.026(±0.001) | 0.109(±0.003) |
| | Deep Coral | IWV | 0.183(±0.001) | 0.329(±0.020) | 4.344(±0.055) | 12.437(±1.027) | 0.025(±0.000) | 0.041(±0.002) | 223.375(±7.211) | 778.306(±64.952) | 0.006(±0.000) | 0.009(±0.002) | 0.028(±0.002) | 0.244(±0.009) | 0.026(±0.002) | 0.104(±0.002) |
| | **Deep Coral** | **SB** | 0.182(±0.000) | 0.325(±0.008) | 4.307(±0.042) | 12.414(±1.209) | 0.025(±0.000) | 0.040(±0.002) | 214.055(±7.216) | 840.409(±183.867) | 0.006(±0.000) | 0.013(±0.007) | 0.027(±0.004) | 0.236(±0.012) | 0.027(±0.001) | 0.107(±0.007) |
| | Deep Coral | TB | 0.182(±0.000) | 0.321(±0.008) | 4.347(±0.039) | 12.637(±0.949) | 0.025(±0.000) | 0.039(±0.001) | 218.167(±11.586) | 791.889(±121.634) | 0.006(±0.000) | 0.013(±0.007) | 0.028(±0.004) | 0.235(±0.015) | 0.027(±0.001) | 0.107(±0.007) |
| GINO | - | - | 0.336(±0.011) | 0.484(±0.023) | 8.539(±0.477) | 13.895(±0.204) | 0.043(±0.002) | 0.042(±0.002) | 278.161(±12.622) | 1079.753(±152.799) | 0.014(±0.001) | 0.016(±0.002) | 0.092(±0.009) | 0.137(±0.007) | 0.059(±0.005) | 0.083(±0.004) |
| | DANN | DEV | 0.253(±0.010) | 0.486(±0.026) | 8.465(±0.532) | 18.087(±2.490) | 0.034(±0.001) | 0.052(±0.002) | 292.040(±18.691) | 2431.868(±734.757) | 0.014(±0.001) | 0.034(±0.010) | 0.094(±0.006) | 0.261(±0.018) | 0.060(±0.003) | 0.123(±0.004) |
| | DANN | IWV | 0.253(±0.010) | 0.457(±0.023) | 8.461(±0.542) | 18.827(±2.718) | 0.034(±0.001) | 0.053(±0.003) | 287.635(±16.154) | 2247.152(±687.350) | 0.014(±0.001) | 0.025(±0.001) | 0.091(±0.008) | 0.271(±0.020) | 0.061(±0.004) | 0.127(±0.005) |
| | DANN | SB | 0.250(±0.009) | 0.490(±0.025) | 8.303(±0.502) | 19.127(±1.534) | 0.034(±0.001) | 0.053(±0.001) | 283.814(±11.682) | 2253.860(±469.838) | 0.014(±0.001) | 0.036(±0.008) | 0.091(±0.008) | 0.273(±0.013) | 0.059(±0.004) | 0.129(±0.003) |
| | DANN | TB | 0.265(±0.019) | 0.452(±0.025) | 9.122(±1.029) | 17.060(±1.705) | 0.035(±0.002) | 0.049(±0.001) | 335.654(±60.476) | 2141.697(±662.675) | 0.015(±0.002) | 0.031(±0.009) | 0.105(±0.018) | 0.249(±0.012) | 0.066(±0.009) | 0.123(±0.004) |
| | CMD | DEV | 0.493(±0.009) | 0.523(±0.015) | 18.407(±0.472) | 25.595(±1.117) | 0.051(±0.001) | 0.055(±0.001) | 2529.365(±25.475) | 2437.075(±36.475) | 0.045(±0.002) | 0.068(±0.005) | 0.328(±0.017) | 0.354(±0.015) | 0.104(±0.008) | 0.113(±0.007) |
| | CMD | IWV | 0.249(±0.006) | 0.494(±0.007) | 8.101(±0.230) | 19.163(±0.806) | 0.033(±0.001) | 0.055(±0.001) | 287.728(±434.038) | 2154.704(±15.594) | 0.014(±0.000) | 0.037(±0.002) | 0.088(±0.005) | 0.262(±0.005) | 0.057(±0.001) | 0.126(±0.003) |
| | CMD | SB | 0.248(±0.004) | 0.485(±0.010) | 8.227(±0.279) | 19.200(±0.984) | 0.033(±0.001) | 0.054(±0.001) | 284.873(±15.767) | 2040.150(±163.039) | 0.014(±0.001) | 0.035(±0.003) | 0.088(±0.005) | 0.260(±0.005) | 0.058(±0.001) | 0.125(±0.003) |
| | CMD | TB | 0.252(±0.003) | 0.484(±0.013) | 8.392(±0.066) | 18.718(±1.336) | 0.034(±0.001) | 0.053(±0.001) | 299.252(±9.470) | 2055.212(±179.452) | 0.014(±0.000) | 0.037(±0.003) | 0.090(±0.003) | 0.251(±0.020) | 0.059(±0.001) | 0.124(±0.003) |
| | DARE-GRAM | DEV | 0.249(±0.003) | 0.393(±0.040) | 8.204(±0.122) | 14.737(±1.785) | 0.033(±0.001) | 0.044(±0.004) | 279.833(±11.947) | 1399.506(±454.664) | 0.013(±0.000) | 0.019(±0.005) | 0.087(±0.002) | 0.250(±0.020) | 0.058(±0.002) | 0.114(±0.006) |
| | DARE-GRAM | IWV | 0.249(±0.003) | 0.393(±0.040) | 8.204(±0.122) | 14.737(±1.785) | 0.033(±0.001) | 0.044(±0.004) | 279.833(±11.947) | 1399.506(±454.664) | 0.013(±0.000) | 0.019(±0.005) | 0.087(±0.002) | 0.250(±0.020) | 0.058(±0.002) | 0.114(±0.006) |
| | DARE-GRAM | SB | 0.249(±0.004) | 0.413(±0.071) | 8.136(±0.143) | 14.686(±2.786) | 0.033(±0.001) | 0.046(±0.007) | 283.028(±14.455) | 1729.784(±747.392) | 0.013(±0.000) | 0.022(±0.011) | 0.088(±0.002) | 0.242(±0.023) | 0.058(±0.002) | 0.116(±0.010) |
| | DARE-GRAM | TB | 0.250(±0.008) | 0.372(±0.029) | 8.312(±0.464) | 14.587(±1.570) | 0.034(±0.001) | 0.043(±0.002) | 290.095(±9.324) | 1144.219(±455.032) | 0.014(±0.001) | 0.016(±0.006) | 0.090(±0.007) | 0.242(±0.023) | 0.058(±0.004) | 0.111(±0.004) |
| | Deep Coral | DEV | 0.251(±0.005) | 0.382(±0.034) | 8.467(±0.283) | 15.271(±2.039) | 0.034(±0.001) | 0.045(±0.005) | 283.264(±4.893) | 967.525(±133.924) | 0.014(±0.001) | 0.019(±0.004) | 0.091(±0.005) | 0.255(±0.016) | 0.061(±0.004) | 0.111(±0.006) |
| | Deep Coral | IWV | 0.249(±0.006) | 0.380(±0.043) | 8.324(±0.294) | 14.801(±1.201) | 0.033(±0.001) | 0.045(±0.004) | 282.224(±9.132) | 1036.671(±195.506) | 0.014(±0.001) | 0.018(±0.005) | 0.091(±0.006) | 0.243(±0.014) | 0.058(±0.004) | 0.108(±0.004) |
| | **Deep Coral** | **SB** | 0.248(±0.006) | 0.356(±0.011) | 8.234(±0.295) | 14.031(±0.738) | 0.033(±0.001) | 0.041(±0.001) | 280.474(±6.420) | 1002.131(±167.663) | 0.014(±0.001) | 0.017(±0.003) | 0.091(±0.006) | 0.245(±0.014) | 0.058(±0.004) | 0.107(±0.001) |
| | Deep Coral | TB | 0.248(±0.006) | 0.356(±0.012) | 8.230(±0.287) | 14.066(±0.740) | 0.033(±0.001) | 0.041(±0.002) | 279.836(±6.636) | 1010.928(±179.067) | 0.014(±0.001) | 0.017(±0.003) | 0.090(±0.004) | 0.245(±0.014) | 0.058(±0.004) | 0.110(±0.007) |

Again, we further investigate model performance under distribution shift by examining predictions from the best performing model, selected by lowest average error in the target domain. Figure 20 presents the respective distribution of prediction errors in the source and target domain, clearly indicating the negative effects of the distribution shift on model performance.

In this task, the temperature field is the most critical for downstream analysis and optimization, which is why we focus our detailed analysis on it.

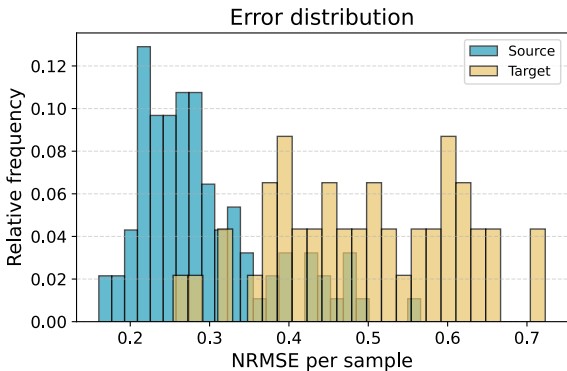

Figure 20: Distribution of NRMSE (averaged across all fields) for the test sets of the source (blue) and target (yellow) domains in the *heatsink design* dataset. Bar height indicates the relative frequency of samples within each bin.

Table 11 compares the absolute temperature prediction errors for the best and worst samples from both the source and target test sets. The corresponding scatter plots are shown in Figures 21 to 24, comparing the ground truth and predicted temperature fields, alongside their absolute errors.

While the best source domain prediction is quite accurate, with low average and percentile errors (Table 11, Figure 21), the 99th percentile of the worst source domain prediction reaches up to 29K. Given a total temperature range of 100K, this represents a relative error of nearly 30%. The worst target domain prediction is even less accurate, showing substantial visual and quantitative deviations from the ground truth (Table 11, Figure 24).

Table 11: Absolute error (K) of temperature predictions for the best and worst samples in the source and target domain of the *heatsink design* dataset. Lowest value per metric is bold.

| Metric | Source | | Target | |
|---|---|---|---|---|
| | **Best** | **Worst** | **Best** | **Worst** |
| Mean | **1.84e+00** | 5.79e+00 | 2.23e+00 | 1.42e+01 |
| Std | **1.94e+00** | 5.90e+00 | 2.85e+00 | 1.46e+01 |
| Median | **1.25e+00** | 4.06e+00 | 1.31e+00 | 8.84e+00 |
| $Q_{01}$ | **2.17e-02** | 7.51e-02 | 2.41e-02 | 1.62e-01 |
| $Q_{25}$ | **5.49e-01** | 1.92e+00 | 5.95e-01 | 4.49e+00 |
| $Q_{75}$ | **2.44e+00** | 7.48e+00 | 2.68e+00 | 1.87e+01 |
| $Q_{99}$ | **9.26e+00** | 2.88e+01 | 1.49e+01 | 6.61e+01 |

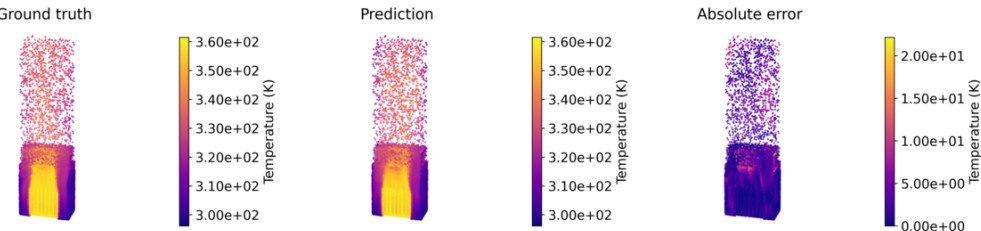

Figure 21: Sliced scatter plot of the *heatsink design* dataset (best source sample). Shown is the ground truth (left) and predicted (middle) temperature field, as well as the absolute error (right).

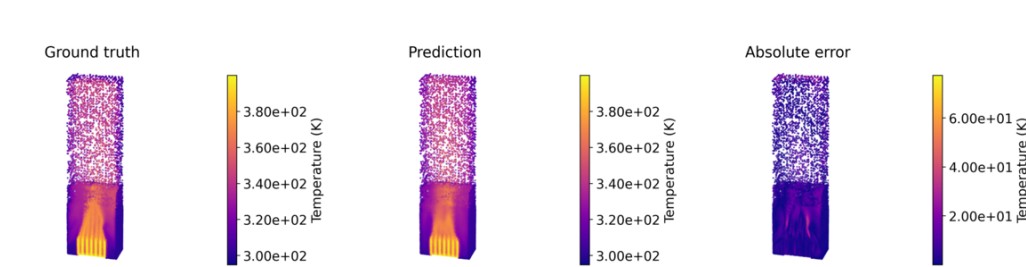

Figure 22: Sliced scatter plot of the *heatsink design* dataset (worst source sample). Shown is the ground truth (left) and predicted (middle) temperature field, as well as the absolute error (right).

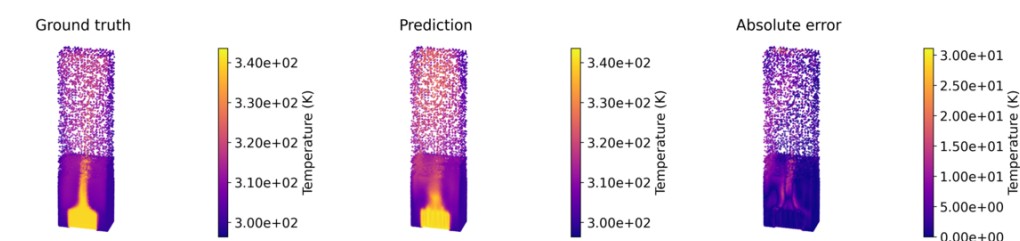

Figure 23: Sliced scatter plot of the *heatsink design* dataset (best target sample). Shown is the ground truth (left) and predicted (middle) temperature field, as well as the absolute error (right).

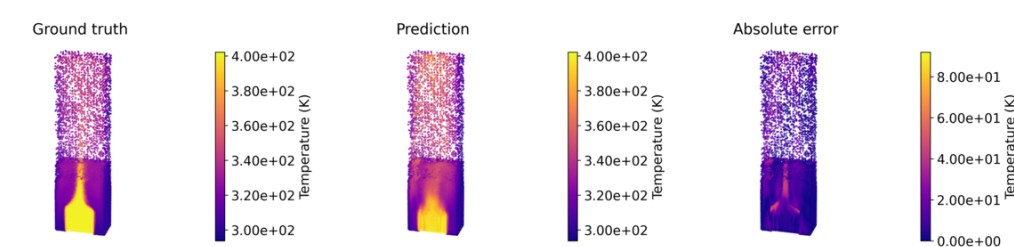

Figure 24: Sliced scatter plot of the *heatsink design* dataset (worst target sample). Shown is the ground truth (left) and predicted (middle) temperature field, as well as the absolute error (right).

## C    DISTRIBUTION SHIFTS

To gain more insights into the parameter importance besides the domain experts' opinion, we visualize the latent space of the conditioning network for all presented datasets in Figures 25 to 28.

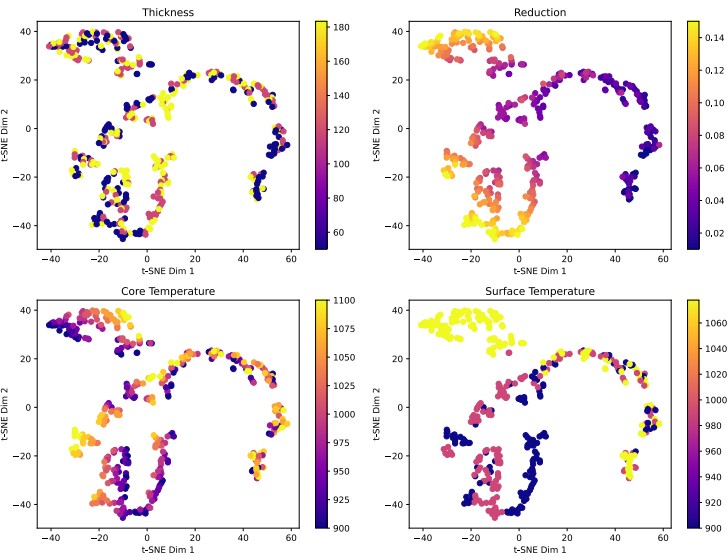

Figure 25: T-SNE visualization of the conditioning vectors for the *hot rolling* dataset. Point color indicates the magnitude of the respective parameter. While the slab thickness $t$ appears to be uniformly distributed, the remaining three exhibit distinct clustering patterns. Taking into account domain knowledge from industry experts, we defined the reduction parameter $r$ as the basis for constructing distribution shifts.

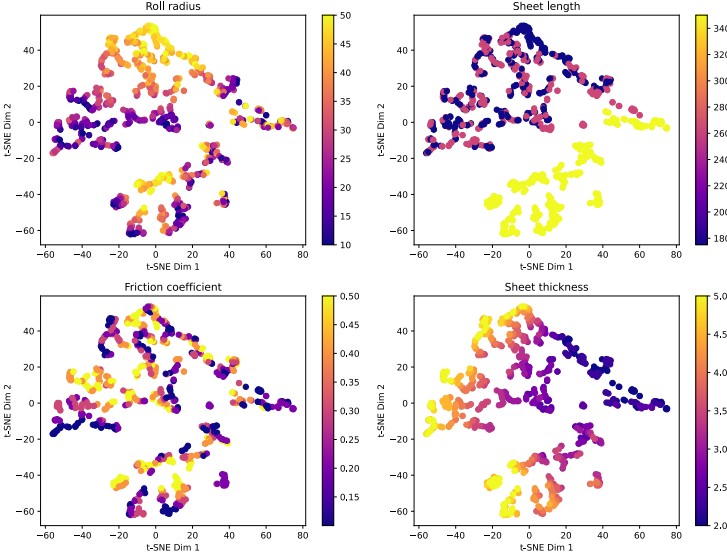

Figure 26: T-SNE visualization of the conditioning vectors for the *sheet metal forming* dataset. Point color indicates the magnitude of the respective parameter. The sheet length $l$ shows the most distinct groupings, but with only three discrete values, it is unsuitable for defining domain splits. The friction coefficient $\mu$ appears uniformly distributed across the embedding. In contrast, sheet thickness $t$ and roll radius $r$ show clustering behavior, making them more appropriate candidates for inducing distribution shifts. We choose $t$ as the domain defining parameter.

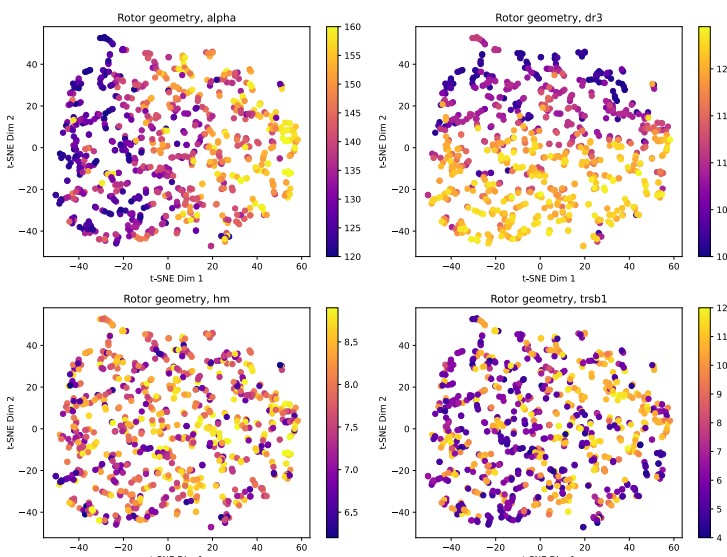

Figure 27: T-SNE visualization of the conditioning vectors for the *electric motor design* dataset. Point color indicates the magnitude of the respective parameter. For clarity, we only show selected parameters. The only parameter for which exhibits see some structure in the latent space is $d_{r3}$, we therefore choose this to be our domain defining parameter in accordance with domain experts.

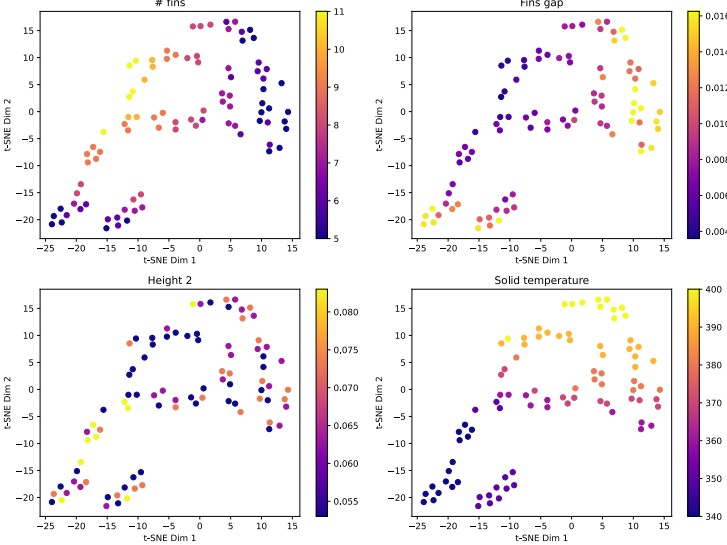

Figure 28: T-SNE visualization of the conditioning vectors for the *heatsink design* dataset. Point color indicates the magnitude of the respective parameter. Height 2 is distributed equally across the representation, but the other parameters show concrete grouping behavior. We therefore choose the number of fins as the domain defining parameter.

# D   DISTANCE MEASURES

**Deep CORAL.**   This distance measures the difference in second-order statistics (covariances) of source and target latent features and can be calculated as follows:

$$d_{\text{deep\_coral}}(\phi(\mathbf{x}), \phi(\mathbf{x}')) = \frac{1}{4k^2} \left\| \mathbf{C} - \mathbf{C}' \right\|_F^2 ,$$

where $\phi(\mathbf{x}), \phi(\mathbf{x}') \in \mathbb{R}^{n \times k}$ denote latent source and target features for a batch size $n$ and a feature dimension $k$, $\mathbf{C}$ and $\mathbf{C}'$ are the source and target feature covariances and $\|\cdot\|_F^2$ is the squared Frobenius norm.

**CMD.**   CMD measures not only the difference in first and second moments of source and target latent features, but also in higher-order central moments. Let $\phi(\mathbf{x}), \phi(\mathbf{x}') \in \mathbb{R}^{n \times k}$ denote the latent activations for a batch size $n$ and feature dimension $k$. The CMD distance up to order $P$ is defined as

$$d_{\text{cmd}}(\phi(\mathbf{x}), \phi(\mathbf{x}')) = \frac{1}{|b - a|} \left\| \boldsymbol{\mu} - \boldsymbol{\mu}' \right\|_2 + \sum_{p=2}^{P} \frac{1}{|b - a|^p} \left\| \mathbf{c}_p(\phi(\mathbf{x})) - \mathbf{c}_p(\phi(\mathbf{x}')) \right\|_2 ,$$

where $\boldsymbol{\mu}, \boldsymbol{\mu}' \in \mathbb{R}^k$ are the source and target empirical mean feature vectors, $|b - a|^p$ can be seen as a hyperparameter of the method which we set to 2 to reflect the original implementation, and $\mathbf{c}_p(\phi(\mathbf{x})), \mathbf{c}'_p(\phi(\mathbf{x}')) \in \mathbb{R}^k$ are the respective $p$-th central moments which are calculated as:

$$\mathbf{c}_p(\phi(\mathbf{x})) = \frac{1}{n} \sum_{i=1}^{n} (\phi(\mathbf{x})_i - \boldsymbol{\mu})^{\odot p}, \qquad \mathbf{c}_p(\phi(\mathbf{x}')) = \frac{1}{n} \sum_{i=1}^{n} (\phi(\mathbf{x}')_i - \boldsymbol{\mu}')^{\odot p}.$$

Above, $(\cdot)^{\odot p}$ denotes the element-wise $p$-th power. Choosing the number of higher-order moments to align is another hyperparameter of the method. For our benchmark, we choose $P = 5$.

**DANN.**   DANN is introduced to minimize an upper bound on the $\mathcal{H}$-divergence between source and target feature distributions. Since it is intractable to compute this directly, the authors use a domain classifier in the form of a small MLP trained to distinguish whether a latent feature comes from the source or the target domain. The error of this classifier is then used to compute the PAD, which, up to a constant depending on the model's VC dimension, upper-bounds the $\mathcal{H}$-divergence (Ganin et al., 2015).

Training is performed via a min–max optimization, i.e. the domain classifier is trained to maximize its classification accuracy, while the feature encoder $\phi$ is trained to *minimize* this separability by using a gradient reversal layer. This adversarial interaction encourages the latent representations of source and target samples to become indistinguishable, thereby promoting domain-invariant features.

**DARE-GRAM.**   DARE-GRAM aims to align a selected low-rank subspace of the pseudo-inverse Gram matrices of source and target features. Given feature matrices $\phi(\mathbf{x}), \phi(\mathbf{x}') \in \mathbb{R}^{n \times k}$ for a batch size $n$ and feature dimension $k$, we can compute their Gram matrices:

$$G = \phi(\mathbf{x})^\top \phi(\mathbf{x}), \qquad G' = \phi(\mathbf{x})'^\top \phi(\mathbf{x})'.$$

Each Gram matrix is then decomposed via eigendecomposition, and its truncated Moore–Penrose pseudo-inverse is formed by keeping the top $p^*$ eigenvalues that explain a fixed proportion of variance (95% for our implementation):

$$G^+ = U_{1:p^*} \, \Lambda_{1:p^*}^{-1} \, U_{1:p^*}^\top, \qquad (G')^+ = U'_{1:p^*} \, (\Lambda'_{1:p^*})^{-1} \, (U'_{1:p^*})^\top.$$

We can then define the difference in angles as

$$d_{\text{angle}}(G, G') = \left\| \mathbf{1} - \cos(\theta_{1:p^*}) \right\|_1,$$

where $\cos(\theta_i)$ is the cosine similarity between the $i$-th column of $G^+$ and $(G')^+$.

Furthermore, we can define the difference in scale as

$$d_{\text{scale}}(G, G') = \left\| \lambda_{1:p^*} - \lambda'_{1:p^*} \right\|_2.$$

The first term aligns the orientation of the dominant inverse-Gram subspaces, whereas the second term matches the principal eigenvalues of the Gram matrices to ensure that feature scale is consistent across source and target.

The total DARE-GRAM distance is defined as a weighted sum of the two:

$$d_{\text{dare\_gram}} = \alpha_{\text{angle}} \, d_{\text{angle}} + \gamma_{\text{scale}} \, d_{\text{scale}},$$

where the $\alpha$ and $\gamma$ are hyperparameters. Following the original authors, we set $\alpha_{\text{angle}} = 0.02$ and $\gamma_{\text{scale}} = 0.001$.

## E  MODEL ARCHITECTURES

This section provides explanations of all model architectures used in our benchmark. All models are implemented in PyTorch and are adapted to our conditional regression task. All models have in common, that they take node coordinates as inputs and embed them using a sinusoidal positional encoding. Additionally, all models are conditioned on the input parameters of the respective simulation sample, which are encoded through a conditioning network described below.

**Conditioning Network.**    The conditioning module used for all neural surrogate architectures embeds the simulation input parameters into a latent vector used for conditioning. The network consists of a sinusoidal encoding followed by a simple MLP. The dimension of the latent encoding is 8 throughout all experiments.

**PointNet.**    Our PointNet implementation is adapted from (Qi et al., 2017) for node-level regression. Input node coordinates are first encoded using sinusoidal embeddings and passed through an encoder MLP. The resulting representations are aggregated globally using max pooling over nodes to obtain a global feature vector. To propagate this global feature, it is concatenated back to each point's feature vector. This fused representation is then fed into a final MLP, which produces the output fields. The conditioning is performed by concatenating the conditioning vector to the global feature before propagating it to the nodes features. We use a PointNet base dimension of 16 for the small model and 32 for the larger model.

**GraphSAGE.**    We adapt GraphSAGE (Hamilton et al., 2017) to the conditional mesh regression setting. Again, input node coordinates are embedded using a sinusoidal encoding and passed through an MLP encoder. The main body of the model consists of multiple GraphSAGE message passing layers with mean aggregation. We support two conditioning modes, namely concatenating the latent conditioning vector to the node features, or applying FiLM style modulation (Perez et al., 2018) to the node features before each message passing layer. We always use FiLM modulation in the presented results. After message passing, the node representations are passed through a final MLP decoder to produce the output fields. The base dimension of the model is kept at 128 and we employ 4 GraphSAGE layers.

**Transolver.**    The Transolver model follows the originally introduced architecture (Wu et al., 2024). Similar to the other models, node coordinates first are embedded using a sinusoidal encoding and passed through an MLP encoder to produce initial features. Through learned assignement, each node then gets mapped to a slice, and inter- as well as intra-slice attention is performed. Afterwards, fields are decoded using an MLP readout. The architecture supports two conditioning modes:

concatenation, where the conditioning vector is concatenated to the input node features before projection, or modulation through DiT layers across the network. For our experiments, DiT is used. We choose a latent dimension of 128, a slice base of 32 and we apply four attention blocks for the small model. For the larger model, we scale to 256, 128 and 8 layers respectively.

**UPT.**   Our UPT implementation builds on the architecture proposed in (Alkin et al., 2024a). First, a fixed number of supernodes are uniformly sampled from the input nodes. Node coordinates are embedded using a sinusoidal encoding followed by an MLP. The supernodes aggregate features from nearby nodes using one-directional message passing and serve as tokens for subsequent transformer processing. They are then processed by stack of DiT blocks, which condition the network on the simulation input parameters. For prediction, we employ a DiT Perceiver (Jaegle et al., 2022) decoder that performs cross-attention between the latent representation and a set of query positions. This allows the model to generate field predictions at arbitrary spatial locations, which is a desirable property for inference. We sample 4096 supernodes and use a base dimension of 192. We use 8 DiT blocks for processing and 4 DiT Perceiver blocks for decoding.

**GINO.**   GINO was proposed in (Li et al., 2023b). Input coordinates are again embedded via sinusoidal encoding, after which the mesh is projected onto a regular latent grid. This is achieved via message passing with connections generated via a radius graph. On the latent grid, the conditioning is concatenated to the features at each grid point before Fourier Neural Operator (FNO) Li et al. (2020a) layers are employed. Afterwards, features are mapped back onto the output grid by querying the latent grid, again via message passing. Our implementation uses a latent grid of size $(16 \times 16 \times 16)$ with 16 latent channels and a radius of 0.1 to construct the radius graph for messing passing operations. For our implementation, we use the library of the original authors.[3]

# F   EXPERIMENTS

This section provides a detailed overview of the performed experiments for this benchmark. First, we explain the benchmarking setup used to generate the benchmarking results in detail in Appendix F.1 and the evaluation procedure in Appendix F.2. Furthermore, we provide information about training times for the presented methods in Appendix F.3.

## F.1   EXPERIMENTAL SETUP

**Dataset Splits.**   We split each dataset into source and target domains as outlined in Section 3.5 and Appendix C. Within source domains, we use a 50%/25%/25% split for training, validation, and testing, respectively. For target domains, where labels are unavailable during training in our UDA setup, we use a 50%/50% split for training and test sets. The large validation and test sets are motivated the industrial relevance of our benchmark, where reliable performance estimation on unseen data is a crucial factor.

**Training Pipeline.**   For training, we use a dataset wide per field z-score normalization strategy, with statistics computed on the source domain training set. We use a batch size of 16 and the AdamW optimizer (Loshchilov & Hutter, 2019) with a weight decay of 1e-5 and a cosine learning rate schedule, starting from 1e-3. Gradients are clipped to a maximum norm of 1. For the large scale *heatsink design* dataset, we enable Automatic Mixed Precision (AMP) to reduce memory consumption and training time. Additionally, we use Exponential Moving Average (EMA) updates with a decay factor of 0.95 to stabilize training.

Performance metrics are evaluated every 10 epochs, and we train all models for a maximum of 3000 epochs with early stopping after 500 epochs of no improvement on the source domain validation loss.

**Domain Adaptation Specifics.**   To enable UDA algorithms, we jointly sample mini batches from the source and target domains at each training step and pass them thorugh the model. Since target labels are not available, we compute supervised losses only on the source domain outputs. In addition,

---

[3]https://github.com/neuraloperator/neuraloperator

we compute DA losses on the latent representations of source and target domains in order to encourage domain invariance.

Since a crucial factor in the performance of UDA algorithms is the choice of the domain adaptation loss weight $\lambda$, we perform extensive sweeps over this hyperparameter and select models using the unsupervised model selection strategies described in Section 4.3.

For the three smaller datasets, we sweep $\lambda$ logarithmically over $\lambda \in \{10^{-1}, 10^{-2}, \ldots, 10^{-9}\}$, while for the large scale *Heatsink design* dataset, we sweep a smaller range, namely $\lambda \in \{10^{2}, 10^{-1}, \ldots, 10^{-2}\}$, motivated by the balancing principle (Zellinger et al., 2021b).

Table 12 provides an overview of the number of trained models for benchmarking performance of all models and all UDA algorithms on the *medium* difficulty domain shifts across all datasets.

Table 12: Overview of the benchmarking setup and number of trained models across all datasets.

| Dataset | Models | UDA algorithms | $\lambda$ values | # seeds | # models trained |
|---|---|---|---|---|---|
| Rolling | PointNet, GraphSAGE, Transolver | Deep Coral, CMD, DANN, DARE-GRAM | $\{10^{-1}; 10^{-9}\}$ | 4 | 432 |
| | | w/o UDA | – | 4 | 12 |
| Forming | PointNet, GraphSAGE, Transolver | Deep Coral, CMD, DANN, DARE-GRAM | $\{10^{-1}; 10^{-9}\}$ | 4 | 432 |
| | | w/o UDA | – | 4 | 12 |
| Motor | PointNet, GraphSAGE, Transolver | Deep Coral, CMD, DANN, DARE-GRAM | $\{10^{-1}; 10^{-9}\}$ | 4 | 432 |
| | | w/o UDA | – | 4 | 12 |
| Heatsink | PointNet, Transover, UPT, GINO | Deep Coral, CMD, DANN, DARE-GRAM | $\{10^{2}; 10^{-2}\}$ | 4 | 320 |
| | | w/o UDA | – | 4 | 12 |
| **Sum** | | | | | **1,664** |

**Additional Details.** For the three smaller datasets, we use smaller networks, while for the large scale *heatsink design* dataset, we train larger model configurations to accommodate the increased data complexity. An overview of model sizes along with average training times per dataset is provided in Table 13. We also refer to the accompanying code repository for a complete listing of all model hyperparameters, where we provide all baseline configuration files and detailed step by step instructions for reproducibility of our results.

Another important detail is that, during training on the *heatsink design* dataset, we randomly subsample 16,000 nodes from the mesh in each training step to ensure computational tractability. However, all reported performance metrics are computed on the full resolution of the data without any subsampling.

### F.2 EVALUATION METRICS

#### F.2.1 GENERAL METRICS

We report the RMSE for each predicted output field. For field $i$, the RMSE is defined as:

$$\text{RMSE}_i^{\text{field}} = \frac{1}{M} \sum_{m=1}^{M} \sqrt{\frac{1}{N_m} \sum_{n=1}^{N_m} \left( y_{m,n}^{(i)} - f(x)_{m,n}^{(i)} \right)^2},$$

where $M$ is the number of test samples (graphs), $N_m$ the number of nodes in graph $m$, $y_{m,n}^{(i)}$ the ground truth value of field $i$ at node $n$ of graph $m$, and $f(x)_{m,n}^{(i)}$ the respective model prediction.

For aggregated evaluation, we define the total Normalized RMSE (NRMSE) as:

$$\text{NRMSE} = \frac{1}{K} \sum_{i=1}^{K} \text{NRMSE}_i^{\text{field}},$$

where $K$ is the number of predicted fields. For this metric, all individual field errors are computed on normalized fields before aggregation.

In addition to the error on the fields, we report the mean Euclidean error of the predicted node displacement. This is computed based on the predicted coordinates $\hat{\mathbf{c}}_{m,n} \in \mathbb{R}^d$ and the ground truth coordinates $\mathbf{c}_{m,n} \in \mathbb{R}^d$, where $d \in \{2, 3\}$ is the spatial dimensionality, as follows:

$$\text{RMSE}^{\text{deformation}} = \frac{1}{M} \sum_{m=1}^{M} \sqrt{\frac{1}{N_m} \sum_{n=1}^{N_m} \|\mathbf{c}_{m,n} - \hat{\mathbf{c}}_{m,n}\|_2}.$$

### F.2.2 PHYSICS METRICS

**Von Mises stress consistency.** For all structural simulations in our benchmark, we predict both the relevant Cauchy stress tensor components and the von Mises equivalent stress. This allows for an internal consistency check using the standard von Mises definition:

$$\sigma_{vM} = \sqrt{\frac{1}{2}\left[(\sigma_{11} - \sigma_{22})^2 + (\sigma_{22} - \sigma_{33})^2 + (\sigma_{33} - \sigma_{11})^2 + 6\tau_{12}^2\right]},$$

with $\sigma_{11}, \sigma_{22}, \sigma_{33}$ denote the normal stresses and $\tau_{12}$ the in-plane shear stress.

We can recompute $\sigma_{vM}$ from the predicted tensor components and compare it to the predicted von Mises value using a normalized mean absolute error:

$$\text{Consistency}_{\text{vM}} = \frac{\sum_{i=1}^{N} |\sigma_{vM,i} - \sigma_{vM_{recalc},i}|}{\sum_{i=1}^{N} |\sigma_{vM,i}|}$$

**Constitutive law consistency.** For the *sheet metal forming* dataset, the material is modeled as elastoplastic with von Mises plasticity and linear isotropic hardening. This defines a yield surface, $\sigma_y$, which represents the material's current strength as a function of the equivalent plastic strain ($\varepsilon_p$):

$$\sigma_y(\varepsilon_p) = \sigma_{y0} + H\,\varepsilon_p,$$

where $\sigma_{y0}$ is the initial yield stress, H the hardening modulus, and $\varepsilon_p$ the equivalent plastic strain.

A physically-correct model must adhere to two conditions based on this law:

1. Elastic nodes ($\varepsilon_p = 0$) must have a stress below this surface: $\sigma_{vM} \leq \sigma_{y0}$.
2. Plastic nodes ($\varepsilon_p > 0$) must have a stress on this surface: $\sigma_{vM} = \sigma_y(\varepsilon_p)$.

Based on these two conditions, we introduce two metrics to evaluate the physical consistency of the predictions:

1. Elastic violation rate (percentage of elastic nodes that incorrectly violate the initial yield stress):

$$\text{Violation}_{\text{elastic}} = \frac{1}{N_{\text{el}}} \sum_{i \in \mathcal{E}} \mathbf{1}\left[\sigma_{vM,i} > \sigma_{y0}\right],$$

   where $\mathcal{E}$ is the set points in the elastic regime and $N_{\text{el}} = |\mathcal{E}|$ is the number of elastic nodes.

2. Plastic Law Residual (NMAE for all plastic nodes):

$$\text{Residual}_{\text{plastic}} = \frac{1}{N_{\text{pl}}} \sum_{i \in \mathcal{P}} \frac{|\sigma_{vM,i} - (\sigma_{y0} + H\,\varepsilon_{p,i})|}{\sigma_{y0}},$$

   where $\mathcal{P}$ is the set points in the plastic regime and $N_{\text{pl}} = |\mathcal{P}|$ is the number of plastic nodes.

**Boundary condition satisfaction.** The *heatsink design* simulations impose two important Dirichlet Boundary Conditions (BCs) on the fin surfaces: no slip velocity and the solid temperature of the fins. Therefore we define the two following errors to measure the violation of these BCs for our surrogates:

$$\text{BC-violation}_T = \frac{1}{N_{\text{fin}}} \sum_{i \in \mathcal{F}} \frac{|T_i - T_{\text{solid}}|}{|T_{\text{solid}} - T_{\text{env}}|},$$

and

$$\text{BC-violation}_{\mathbf{u}} = \frac{1}{N_{\text{fin}}} \sum_{i \in \mathcal{F}} \|\mathbf{u}_i\|,$$

where $\mathcal{F}$ is the set of fin nodes, $N_{\text{fin}} = |\mathcal{F}|$ is the number of fin nodes, and $T_i$ and $\mathbf{u}_i$ are the respective predictions for temperature and velocity at node $i$.

These should be interpreted as "soft" BC consistency checks. OpenFOAM enforces Dirichlet BCs on faces of boundary patches, whereas our dataset contains cell center values. Cells adjacent to the fins generally exhibit nonzero gradients, meaning their ground-truth temperatures and velocities do not exactly satisfy the BCs, also in the ground truth data.

### F.3 COMPUTATIONAL RESOURCES AND TIMINGS

While generating the results reported on the *medium* difficulty level of our benchmark, we measured average training times per dataset and model architecture. All our runs were timed on a single NVIDIA H200 144GB GPU for a fair comparison. While the total compute budget is difficult to estimate due early stopping, we provide a detailed analysis of the average training times for 2000 epochs in Table 13.

This table refers to models trained with Deep CORAL, however different UDA algorithms do not add significant computational cost. What is more impactful concerning the full pipeline (including model selection) is the number of hyperparameter variations. The total cost of one UDA algorithm & model selection pipeline can be estimated by multiplying the average training time by the number of trained models (e.g. $\times 9$ if one sweeps over 9 hyperparameters of $\lambda$), for sequential execution. Furthermore, the model selection method's runtime training is negligible compared to the training times.

Table 13: Average training times (averaged for 2000 epochs) and parameter counts for each model on the *medium* difficulty benchmark tasks. Times are measured on a H200 144GB GPU using a batch size of 16.

| Dataset | # samples | Avg. # nodes | Model | # parameters | Avg. training time (h) |
|---|---|---|---|---|---|
| Rolling | 4,750 | 576 | PointNet | 0.3M | 0.75 |
| | | | GraphSAGE | 0.2M | 1.77 |
| | | | Transolver | 0.57M | 1.77 |
| Forming | 3,315 | 6,417 | PointNet | 0.3M | 2.35 |
| | | | GraphSAGE | 0.2M | 10.82 |
| | | | Transolver | 0.57M | 3.74 |
| Motor | 3,196 | 9,052 | PointNet | 0.3M | 2.35 |
| | | | GraphSAGE | 0.2M | 12.14 |
| | | | Transolver | 0.57M | 3.60 |
| Heatsink | 460 | 1,385,594 | PointNet | 1.08M | 3.88 |
| | | | Transolver | 4.07M | 4.94 |
| | | | UPT | 5.77M | 4.73 |
| | | | GINO | 2.5M | 5.94 |

## G DATASET DETAILS

### G.1 HOT ROLLING

The *hot rolling* dataset represents a hot rolling process in which a metal slab undergoes plastic deformation to form a sheet metal product. The model considers a plane-strain representation of a heated steel slab segment with a core temperature $T_{\text{core}}$ and a surface temperature $T_{\text{Surf}}$, initially at thickness $t$, passing through a simplified roll stand with a nominal roll gap $g$ (see Figure 2a). This roll

gap effectively matches the exit thickness of the workpiece. Given the material properties, the initial temperature distribution over the slab thickness and the specified pass reduction, the model aims to capture the evolution of the thermo-mechanical state of the workpiece as it traverses the roll gap.

To reduce computational complexity, the analysis is confined to the vertical midplane along the rolling direction based on a plane-strain assumption. This is well justified by the high width-to-thickness ratio characteristic of the workpiece. Additionally, vertical symmetry is also exploited. Consequently, only the upper half of the workpiece and the upper work roll are modeled.

The workpiece is discretized using plane-strain, reduced-integration, quadrilateral elements. Mesh generation is fully automated, with the element size calibrated according to findings from a mesh convergence study. In terms of mechanical behavior, the workpiece is modeled as elasto-plastic with isotropic hardening, employing tabulated flow curves representative for a titanium alloy (Lesuer, 2000; Lu et al., 2018). The elastic modulus and flow stress are temperature dependent, with the latter also influenced by the plastic strain rate. In contrast, material density and Poisson's ratio are assumed to remain constant. The work roll with a diameter of 1000 mm is idealized as an analytically defined rigid body.

In addition to the mechanical behavior, the elements also feature a temperature degree of freedom that captures thermal phenomena, which are in turn fully coupled with the mechanical field. Heat conduction within the workpiece is governed by temperature dependent thermal conductivity and specific heat capacity. Heat transfer at the interface between the workpiece and the roll is modeled as proportional to the temperature difference between the contacting surfaces, using a heat transfer coefficient of 5 $\mathrm{mW/mm^2K}$. The model also accounts for internal heat generation due to plastic deformation, based on the standard assumption that 90% of plastic work is converted into heat. Additionally, all frictional energy is assumed to be fully transformed into heat and evenly divided between the workpiece and the roll. However, since the analysis focuses on the workpiece, only the portion of this heat entering the workpiece is considered.

The FE simulation is performed with the *Abaqus* explicit solver using a relatively high mass scaling factor of 100. This mass scaling proved to be a suitable choice for maintaining both computational efficiency and solution accuracy. The pre-processing, evaluation and post-processing of the simulations was automated in Python. A full factorial design of experiments was conducted by varying the parameters outlined in Table 14. Simulation outputs from Abaqus (.odb files) were converted to a more suitable .h5 format in post-processing, enabling seamless integration into the SIMSHIFT framework. All simulations were run on a Gigabyte Aorus 15P KD consumer laptop equipped with an Intel Core i7-11800H CPU (8 cores, 16 threads, 2.30–4.60 GHz), 16 GB DDR4 RAM at 3200 MHz and a 1 TB NVMe SSD. The single-core CPU time for one simulation was 25 seconds on average, depending on the mesh size and convergence speed.

Table 14: Input parameter ranges for the *hot rolling* simulations. Samples are generated by equally spacing each parameter within the specified range using the indicated number of steps, resulting in $5 \times 19 \times 10 \times 5 = 4750$ total samples.

| Parameter | Description | Min | Max | Steps |
|---|---|---|---|---|
| $t\ (mm)$ | Initial slab thickness. | 50.0 | 183.3 | 5 |
| reduction $(-)$ | Reduction of initial slab thickness. | 0.01 | 0.15 | 19 |
| $T_{\mathrm{core}}\ (^\circ C)$ | Core slab temperature. | 900.0 | 1000.0 | 10 |
| $T_{\mathrm{surf}}\ (^\circ C)$ | Surface slab temperature. | 900.0 | 1077.77 | 5 |

### G.2 SHEET METAL FORMING

For the *sheet metal forming* dataset, a w-shaped bending process was selected due to its complex contact interactions and the highly nonlinear progression of bending forces. For this purpose, a parameterized 2D FE model of the process was developed using the commercial FEM software *Abaqus* and its implicit solver, with the simulation pipeline implemented in Python. The initial configuration of the finite element model is shown in Figure 29 and described below.

Due to geometric and loading symmetry, only the right half of the sheet with a thickness $t$ was modeled. The die and punch were idealized as rigid circular segments with a shared radius $r$. Additionally, a

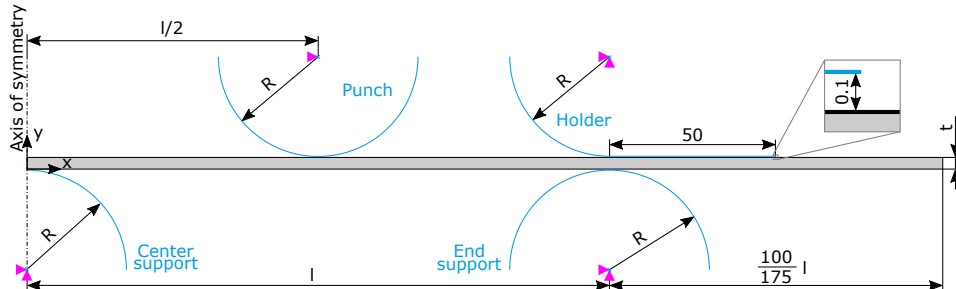

Figure 29: Bending process abstraction, initial configuration.

rigid blank holder comprising an arc and a straight segment was positioned $0.1$ mm above the sheet to maintain contact and restrain vertical motion. The required sheet length was determined by the support span $l$, enabling material flow toward the center in response to the downward motion of the punch.

The sheet was discretized using bilinear, plane-strain quadrilateral elements with reduced integration and hourglass control (Abaqus element type CPE4R). A prior mesh convergence study indicated that accurate simulation results require a minimum of 10 element rows across the sheet thickness. The element size was fixed at $0.125 \times 0.1$ mm to ensure a uniform aspect ratio, constraining the sheet thickness to $t > 1$ mm.

The sheet material was modeled as elastoplastic with von Mises plasticity and linear isotropic hardening. The following properties were assigned: Young's modulus of 210 GPa, Poisson's ratio of 0.3, yield stress of 410 MPa, and hardening modulus of 2268 MPa.

For all contact interfaces, a normal contact formulation with surface-to-surface discretization, penalty enforcement, and finite-sliding tracking was employed. Tangential contact was modeled via a Coulomb friction law with a coefficient $\mu$.

The supports and blank holder were fixed by constraining horizontal and vertical translations as well as in-plane rotations. These constraints were applied at the centroid of each arc segment, representing the reference point for the respective rigid body. The punch was similarly constrained against horizontal movement and rotation but retained vertical mobility. The deformed configuration following a vertical displacement $U$ of the punch is illustrated in Figure 30.

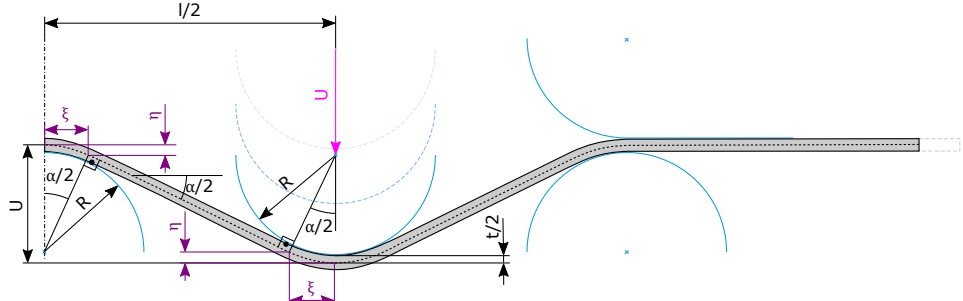

Figure 30: Bending process abstraction, deformed configuration.

A full factorial design of experiments was conducted by varying the parameters outlined in Table 15. As for the hot rolling simulations, outputs from Abaqus (.odb files) were converted to .h5 format in post-processing, to integrate them into the SIMSHIFT framework. All simulations were run on a Gigabyte Aorus 15P KD consumer laptop equipped with an Intel Core i7-11800H CPU (8 cores, 16 threads, 2.30–4.60 GHz), 16 GB DDR4 RAM at 3200 MHz and a 1 TB NVMe SSD. The single-core CPU time for one simulation run was 300 seconds on average, depending on mesh size and convergence speed.

Table 15: Input parameter ranges for the *sheet metal forming* simulations. Samples are generated by equally spacing each parameter within the specified range using the indicated number of steps, resulting in $17 \times 13 \times 3 \times 5 = 3315$ total samples.

| Parameter | Description | Min | Max | Steps |
|---|---|---|---|---|
| $r \ (mm)$ | Roll radius. | 10.0 | 50.0 | 17 |
| $t \ (mm)$ | Sheet thickness. | 2.0 | 5.0 | 13 |
| $l \ (mm)$ | Sheet length. | 175.0 | 350.0 | 3 |
| $\mu \ (-)$ | Friction coefficient between the sheet and the rolls. | 0.1 | 0.5 | 5 |

### G.3 ELECTRIC MOTOR DESIGN

The *electric motor design* dataset includes a structural FE simulation of a rotor within electric machinery, subjected to mechanical loading at burst speed. The rotor topology is modeled after the motor architecture of the 2010 Toyota Prius (Burress et al., 2011), an industry-recognized benchmark frequently used for validation and comparison in academic and industrial research. The Prius rotor topology is based on a V-shaped magnet configuration as shown in Figure 31.

Structural rotor simulations are essential in multi-physics design optimization, where motor performance is evaluated across multiple domains including electromagnetic, thermal, acoustic, and structural. Using a design optimization framework, stator and rotor design are iteratively refined to identify Pareto-optimal solutions based on objectives such as efficiency, torque, weight, and speed. In this process, the structural FE model predicts stress and deformation due to loading ensuring the rotor's structural integrity.

The set up and execution of the structural simulations for this dataset are automated and implemented in the open source design optimization framework *SyMSpace*[4]. The FE simulation of the rotor is performed using a mixed 2D plane stress and plane strain formulation with triangular elements. To enhance computational efficiency, geometric symmetry is exploited and only a 1/16 sector of the full rotor is modeled. The mechanical simulation is static and evaluates the rotor under centrifugal loading, incorporating press-fit conditions between the rotor core and shaft, as well as contact interactions between the rotor core and embedded magnets.

An elastic material behavior is employed for all components, including the rotor core, shaft, and magnets. Material properties are summarized in Table 16. Based on the parametrized CAD model of the rotor topology, the geometry is automatically meshed using *Netgen*[5]. The design optimization tool also automatically identifies nodes for boundary conditions and contact surfaces and applies the corresponding constraints and interactions required for the simulation. The implicit FE solver *HOTINT* is used to compute the quasi-static response of the system, providing local stress and strain fields across the rotor topology.

Table 16: Material parameters for the structural *electric motor design* simulations.

| | Rotor Core | Rotor Shaft | Permanent Magnet |
|---|---|---|---|
| Material | NO27-14 Y420HP | 42CrMo4 | BMN-40SH |
| Density ($kg/dm^3$) | 7.6 | 7.72 | 7.55 |
| Possions ratio (-) | 0.29 | 0.3 | 0.24 |
| Young's Modulus ($kN/mm^2$) | 185.0 | 210.0 | 175.0 |
| Tensile Strength ($kN/mm^2$) | 550.0 | 850.0 | 250.0 |

To generate the electric motor dataset, a comprehensive motor optimization study was conducted using *SyMSpace*, based on design specifications of the 2010 Toyota Prius. The optimization aimed to minimize multiple performance metrics, including motor mass, material costs, rotor torque ripple, motor losses, coil temperature, stator terminal current, and elastic rotor deformation. A genetic algorithm was employed to explore the design space and identify Pareto-optimal solutions. In the

---

[4]https://symspace.lcm.at/

[5]https://ngsolve.org/

process, 3,196 motor configurations were evaluated by varying, among other factors, the rotor's topological parameters within the bounds specified in Table 17. The outputs of the structural simulations were generated in .vtk format and then stored in .h5 files, allowing direct integration into the SIMSHIFT framework. Each structural simulation required approximately 4 to 5 minutes of single-core CPU time on a Intel Core i9-14900KS processor (24 Cores, 3200 MHz), depending on convergence speed of the contact algorithm.

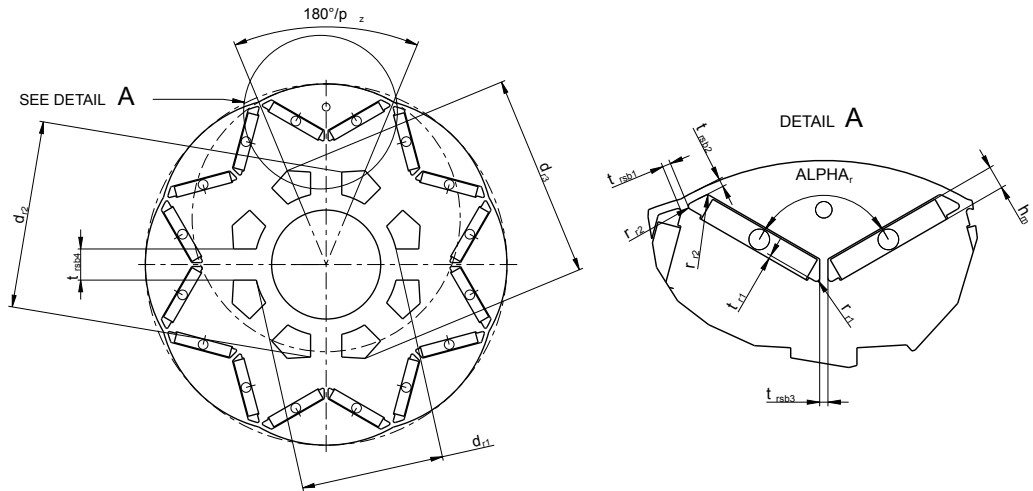

Figure 31: Technical drawing of the electrical motor. Sampling ranges for the shown parameters can be found in Table 17.

Table 17: Input parameters for the *electric motor design* simulations. Since the design space was explored by a genetic algorithm, the parameters are not uniformly sampled as in the previous simulation scenarios. In total, 3196 simulations were performed.

| Parameter | Description | Min | Max |
|---|---|---|---|
| $d_{si}$ $(mm)$ | Stator inner diameter. | 150.0 | 180.0 |
| $h_m$ $(mm)$ | Magnet height. | 6.0 | 9.0 |
| $\alpha_r$ $(°)$ | Angle between magnets. | 120.0 | 160.0 |
| $t_{r1}$ $(mm)$ | Magnet step. | 1.0 | 5.0 |
| $r_{r1}$ $(mm)$ | Rotor slot fillet radius 1. | 0.5 | 2.5 |
| $r_{r2}$ $(mm)$ | Rotor slot fillet radius 2. | 0.5 | 3.5 |
| $r_{r3}$ $(mm)$ | Rotor slot fillet radius 3. | 0.5 | 5.0 |
| $r_{r4}$ $(mm)$ | Rotor slot fillet radius 4. | 0.5 | 3.0 |
| $t_{rsb1}$ $(mm)$ | Thickness saturation bar 1. | 4.0 | 12.0 |
| $t_{rsb2}$ $(mm)$ | Thickness saturation bar 2. | 1.0 | 3.0 |
| $t_{rsb3}$ $(mm)$ | Thickness saturation bar 3. | 1.2 | 4.0 |
| $t_{rsb4}$ $(mm)$ | Thickness saturation bar 4. | 5.0 | 12.0 |
| $d_{r1}$ $(mm)$ | Rotor slot diameter 1. | 60.0 | 80.0 |
| $d_{r2}$ $(mm)$ | Rotor slot diameter 2. | 80.0 | 120.0 |
| $d_{r3}$ $(mm)$ | Rotor slot diameter 3. | 100.0 | 125.0 |

## G.4 HEATSINK DESIGN

The *heatsink design* dataset consists of heatsink geometries similar to the example shown in Figure 32, placed centrally at the bottom of a surrounding box-shaped domain filled with air. The dimensions of the surrounding enclosure are 0.14 m × 0.14 m × 0.5 m (length × width × height).

The geometric configuration of each heatsink is defined by several parameters, which were varied within specified bounds for the design study. These parameters and their corresponding value ranges are summarized in Table 18. A total of 460 simulation cases were generated, with non-uniform sampling across the parameter space.

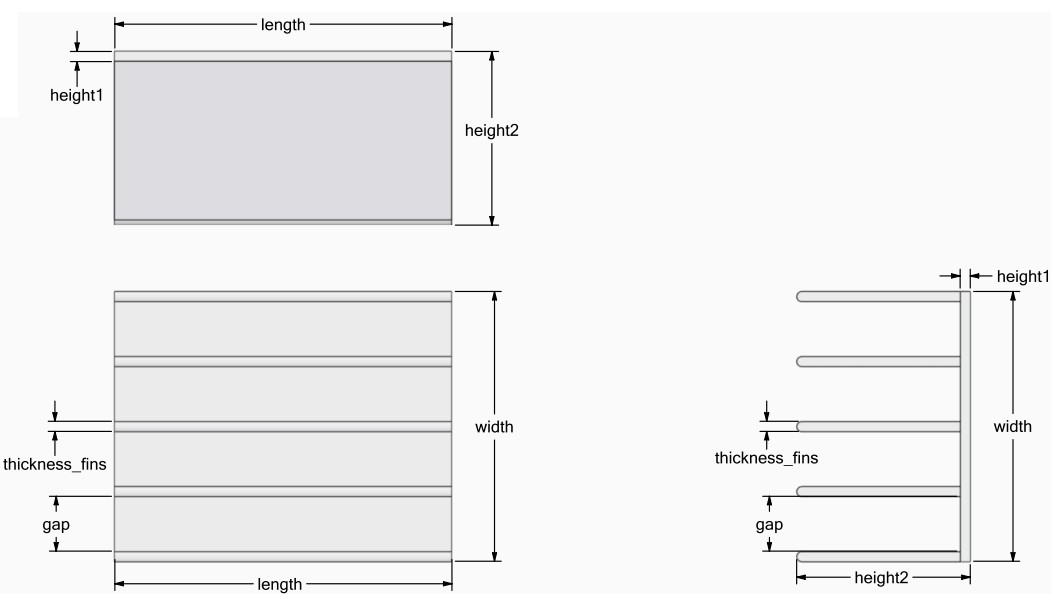

Figure 32: Technical drawing of the solid body in the *heatsink design* dataset. Some of the shown parameters are varied for data generation (see Table 18).

Table 18: Geometric and physical parameters of the *heatsink design* simulations. The variable parameters were not uniformly sampled. In total, 460 simulations were performed.

| Parameter | Description | fixed Value | Min | Max |
|---|---|---|---|---|
| length (m) | Heatsink length | 0.1 | - | - |
| width (m) | Heatsink width | 0.08 | - | - |
| height1 (m) | Baseplate height | 0.003 | - | - |
| T(amb) (K) | Ambient Temperature | 300 | - | - |
| fins (−) | Number of fins | - | 5 | 14 |
| gap (m) | Gap between fins | - | 0.0023 | 0.01625 |
| thickness_fins (m) | Thickness of fins | - | 0.003 | 0.004 |
| height2 (m) | Heatsink height | - | 0.053 | 0.083 |
| T (solid) (K) | Temperature of the solid fins | - | 340 | 400 |

The dataset was generated using CFD simulations based on the Reynolds-Averaged Navier-Stokes (RANS) equations coupled with the energy equation. All simulations were conducted in the open-source CFD suite *OpenFOAM 9*.

The computational domain was discretized using a finite volume method with second-order spatial discretization schemes. A structured hexahedral background mesh was generated with the blockMesh utility in OpenFOAM, followed by mesh refinement using snappyHexMesh to accurately resolve the heatsink structure defined in STL format.

To simulate buoyancy driven natural convection, the buoyantSimpleFoam solver was employed. This solver is designed for steady state, compressible, buoyant flows, using the SIMPLE algorithm for pressure-momentum coupling, extended with under relaxation techniques to enhance numerical stability and robust convergence.

Boundary conditions were applied as follows:

- Walls of the surrounding: no-slip velocity condition with fixed ambient temperature as defined in Table 18.

- Walls of the heatsink: no-slip velocity condition with solid temperature within the range specified for parameter T (solid) in Table 18.

Given the turbulent nature of the flow, the RANS equations were closed using the SST k–$\omega$ turbulence model (Menter et al., 2003). Near-wall regions were modeled using a $y^+$-insensitive near-wall treatment, allowing accurate resolution of boundary layers without the need for excessively fine meshes.

A mesh convergence study was conducted to ensure numerical accuracy. Depending on mesh resolution, each simulation required approximately 11 to 18 hours of single-core CPU time on an Intel Core i9-14900KS processor (24 cores, 3.2 GHz).

## H ABLATION STUDIES

In the following sections, we present ablations on the SIMSHIFT framework.

### H.1 GEOMETRIC ENCODING

The design concept of SIMSHIFT is to allow plug-in integration of any UDA algorithm and model architecture, as long as the model can be conditioned in some way (see Figure 1). However, explicitly conditioning models on scalar geometric parameters is not the only option: for instance, domain-specific information may be encoded implicitly in the mesh itself. To investigate this, we provide an ablation in which the model encodes the mesh directly and is not explicitly conditioned on the scalar parameters. Specifically, we replace the feed-forward conditioning network with a geometric PointNet based encoder to embed the input mesh into a global latent vector, on which UDA is then performed.

We report results of this setup on the *electric motor design* dataset. The setup follows the benchmarking procedure described in Section 4 and Appendix F.1: for each UDA algorithm, we train across 9 different regularizer strengths and 4 random seeds.

Table 19: RMSE (mean ± std over 4 seeds) on the *electric motor design* dataset when using a PointNet geometry encoder. Values are target domain errors (lower is better). Bold marks the overall best model + UDA algorithm + model selection combination. For each architecture, the unregularized baseline row is shaded beige, whereas the best UDA + selection within that architecture is underlined and shaded green.

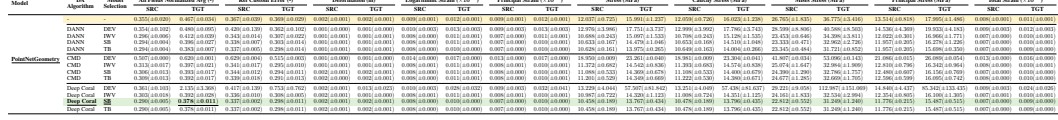

Table 19 shows that UDA algorithms can boost target performance compared to the unregularized baseline model. However compared to our chosen benchmarking design in Table 8, both the performance of the unregularized baseline as well as the one of the best performing UDA method is worse, which supports our choice of explicitly conditioning on scalar parameters in the main benchmark.

## H.2 TWO-DIMENSIONAL SHIFTS

Defining shifts based on one parameter allows for controlled experiments, also given that the parameters were picked based on preliminary experiments (see Appendix C) and consultation with domain experts. In real-world scenarios, however, distribution shifts often affect multiple parameters simultaneously rather than only a single one. It is therefore important to investigate the performance of the benchmarked UDA algorithms under multidimensional parameter shifts. As a step in this direction, we provide an ablation on the *electric motor design* dataset for a two-dimensional parameter shift.

To be concise, we jointly shift the rotor slot diameter $d_{r3}$ (parameter shift in the main benchmark) and the angle between the magnets $\alpha_r$. Table 20 shows the corresponding two-dimensional distribution shift between the source and the target domain.

Table 20: Parameter ranges for the two-dimensional distribution shift on the *electric motor design* dataset.

| Parameter | Source range | Target range |
|---|---|---|
| Rotor slot diameter 3 $d_{r3}$ ($mm$) | [100, 120) | [120, 126] |
| Angle between magnets $\alpha_r$ (°) | [119, 153) | [153, 170] |

We train all models with each UDA algorithm following the procedure in Section 4.5 and Appendix F.1, and report the results in Table 21.

Table 21: RMSE (mean $\pm$ std over 4 seeds) on the *electric motor design* dataset at with a two-dimensional distribution shift in parameter space. Values are target domain errors (lower is better). Bold marks the overall best model + UDA algorithm + model selection combination. For each architecture, the unregularized baseline row is shaded beige, whereas the best UDA + selection within that architecture is underlined and shaded green.

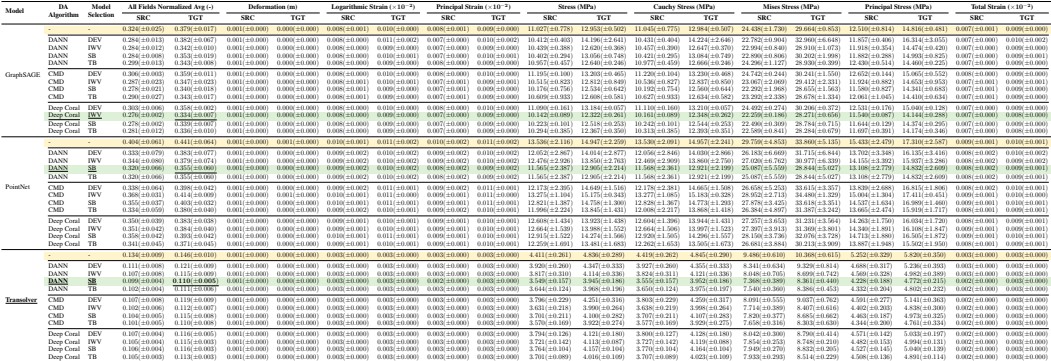

Comparing these results with the original one-dimensional shift (Table 8), two observations stand out: (i) For two out of three architectures, both the unregularized baseline and the best UDA algorithm and model selection combination exhibit higher errors in the average field NRMSE than in the one-dimensional shift setting, confirming that the two-dimensional shift is a more challenging task. (ii) The relative improvements over the unregularized baselines are larger, indicating that UDA training provides greater benefits under this more challenging distribution shift.

These findings highlight the potential of UDA to handle increasingly complex distribution shifts, underscoring its practical relevance for real-world applications.

