# OpenReview forum: "SIMSHIFT: A Benchmark for Adapting Neural Surrogates to Distribution Shifts"
_ICLR.cc/2026/Conference — Submitted to ICLR 2026_

### Official Review · Reviewer_hbC7 · 2025-10-30

**Soundness:** 2
**Presentation:** 2
**Contribution:** 2
**Rating:** 2
**Confidence:** 3

**Summary:**

The paper introduces SIMSHIFT, a benchmark and dataset suite designed to evaluate unsupervised domain adaptation (UDA) methods for neural surrogates trained on physics-based simulations. It provides four industrially relevant simulation datasets (rolling, forming, motor, and heatsink design) and evaluates classical UDA algorithms (CMD, Deep CORAL, and DANN) across several neural surrogate architectures (PointNet, GraphSAGE, Transolver, UPT). The goal is to quantify performance degradation under parametric distribution shifts and assess whether domain adaptation can recover generalization in unseen configurations.

**Strengths:**

- While neural surrogates themselves are not new, the authors are arguably the first to systematically benchmark domain adaptation methods in this context, bridging simulation modeling and UDA research.
- The proposed datasets are realistic, diverse, and physically grounded, covering a range of FEM and CFD problems from different industries.
- The benchmark is well-documented, publicly released, and accompanied by detailed simulation parameters, architectures, and hyperparameter sweeps.
- The study runs a large number of controlled experiments with multiple architectures, adaptation methods, and model-selection strategies, highlighting consistent trends and challenges in out-of-distribution generalization.

**Weaknesses:**

- The benchmark primarily relies on pre-2019 UDA methods (CMD, Deep CORAL, DANN). More modern approaches, e.g., adversarial information bottleneck, self-supervised alignment, or diffusion-based DA, could offer stronger baselines and better contextualize the results.
- All benchmarked shifts are parametric (covariate) shifts, while the introduction also mentions “instrument shifts” such as mesh geometry or solver discretization changes, which are not actually tested. Including such structural or concept-level shifts would significantly increase the benchmark’s value.
- The current metrics focus on RMSE or custom geometric error terms. The benchmark does not assess physical validity (e.g., conservation of energy, boundary-condition satisfaction, equilibrium consistency) of adapted surrogate predictions, which is critical in engineering contexts.
- The work is primarily a benchmark contribution, not a conceptual advance in neural surrogate modeling or domain adaptation algorithms.
- The distinction between “neural surrogates” and generic regression surrogates (MLPs) seems somewhat artificial; many of these architectures are used interchangeably in most industrial ML contexts, whether simulation-based or data-driven. The framing could overstate novelty relative to standard supervised regression with UDA.

**Questions:**

- Could the authors benchmark recent UDA or domain generalization methods, such as SWAD, Tent, AdaBN, or contrastive/self-supervised approaches, to contextualize performance trends?
- Can non-parametric shifts (e.g., mesh geometry changes, discretization refinements, or modified boundary conditions) be introduced to better reflect the “instrument shift” mentioned in the motivation?
- Would it be possible to integrate physics-based evaluation criteria, such as PDE residuals or constraint violation penalties, into the assessment?
- Are all datasets equally sensitive to domain gaps? A cross-dataset or per-task ablation might reveal which physics or modeling types pose the greatest adaptation challenges.
- Finally, could the authors clarify the extent to which the surrogates incorporate physics priors (operator learning, inductive biases) versus being purely data-driven regressors?

---

> ### Author Response · Authors · 2025-11-20
>
> We thank reviewer __hbC7__ for their detailed review, raising viable points. It helped us improve our work and we will discuss all brought up points below.
>
> ## [W1 + Q1] Pre-2019 UDA
>
> We agree that it is valuable for the community to benchmark a UDA method that came out more recently. Below, we build on your proposals to improve our setting w.r.t. more recent algorithmic developments.
>
> Your proposal of TENT [1] is very interesting and in fact extends our benchmark for *unsupervised domain adaptation* to the related problem of *test-time adaptation*, where access to source data is prohibited during adaptation. To investigate test-time adaptation, we added the following experiments: As TENT is not for regression (our setup) but for classification (present softmax uncertainties), we implemented the related regression method Significant-subspace Alignment (SSA) [2].
>
> The novel results are shown in the table below, where we show target NRMSE on the target domain of a Transolver model trained in an unregularized fashion (-) and with UDA compared to the unregularized model where we apply SSA at test-time (Avg over four runs).
>
> | Dataset  |    -   |  UDA  |  SSA  |
> | :------- | :----: | :---: | :---: |
> | Rolling  | 0.365  | 0.192 | 0.559 |
> | Forming  | 0.168  | 0.155 | 0.138 |
> | Motor    | 0.116  | 0.098 | 0.110 |
> | Heatsink | 0.441  | 0.328 | 0.632 |
>
> Overall the results show a decrease in target performance for some datasets but an increase for others. These preliminary results underline the fact that test-time adaptation is an interesting direction in the field and we hope our benchmark can fuel some future work.
>
> Beyond test-time adaptation, we added a recent UDA algorithm from 2023. Contrastive methods (as they are for classification) are intersting but not applicable to regression without modification. We instead implemented __DARE-GRAM__ [3] and run novel experiments on all datasets with the same number of seeds (4) and hyperparameter sweeps as for the other methods and on all mode architectures, resulting in a total of 404 additional training runs and report a summary of the results below. We show the NRMSE over all fields in the target domain and values in parentheses indicate promotion compared to unregularized baseline model. We constrain this table to the best model and model selection algorithm for DARE-GRAM per dataset.
>
> | Dataset  | Model  |  Model Selection  |  All Fields Normalized Avg TGT (promotion to baseline)  |
> | :------- | :----: | :---: | :---: |
> | Rolling  | GraphSAGE  | IWV | 0.192 (-0.173) |
> | Forming  | Transolver  | IWV | 0.167 (-0.001) |
> | Motor    | Transolver  | IWV | 0.102 (-0.014) |
> | Heatsink | UPT  | SB | 0.331 (-0.11) |
>
> The results are promising showing a __reduction in target error for all datasets__. Furthermore, when comparing with all other UDA algorithms, DARE-GRAM is the best performing one on the rolling dataset. We add results for DARE-GRAM for all models over all datasets in the main table (Table 3) and the detailed tables in the Appendix (Tables 4, 6, 8 and 10).
>
> To fuel further research in the area, we add a paragraph __"Future Directions"__ into our Discussion (Section 6), where we include your suggestions of adversarial-information bottleneck methods, diffusionn-based approaches, ensembling techniques like SWAD and others, allowing users of the benchmark to extend these settings to our high-dimensional regression problems.
>
> ---
>
> [1] Wang et al. (2021). Tent: Fully Test-Time Adaptation by Entropy Minimization. ICLR 2021
>
> [2] Adachi et al. (2025). Test-Time Adaptation for Regression by Subspace Alignment. ICLR 2025
>
> [3] Nejjar, et al. (2023). DARE-GRAM: Unsupervised Domain Adaptation Regression by Aligning Inverse Gram Matrices. CVPR 2023
>
> ## [W2 + Q2] Alternative domain shift definitions
>
> As you correctly pointed out, we mention geometry as a possible domain shift in the introduction. We want to emphasize it is exactly this setting that we test in most datasets. Effectively, in three of the four datasets, the domain shift defining parameter(s) describe the initial geometry of the samples: thickness of the plate for forming, rotor slot diameter for the electric motor and the number of fins for the heatsink (see Table 2 for the list of parameters and Figures 2, 3, 31 and 32 for technical drawings of the problems). With a domain shift defined on these parameters, it implicitly implies a shift in geometry between the domains. We even go a step further to investigate this by providing an ablation where the model is not conditioned on the scalar parameters, but a PointNet encoder is used to directly encode the mesh (which implicitly contains the geometry parameters). The results and a corresponding discussion can be found in Appendix H.1.
>
> Concerning a change in BCs, the *hot rolling* dataset can be seen as an example of this. The domain defining parameter is the reduction, wich is essentially a geometric boundary condition on our domain.

---

> ### Author Response · Authors · 2025-11-20
>
> ## [W3 + Q3] Physics-based evaluation
>
> We agree that physics-based assessments are important in engineering contexts and, in addition to our global and local RMSE-based metrics, our datasets allow for the following checks.
>
> (1) __Von Mises Consistency__ (all structural datasets). We predict both the Cauchy stress tensor components and the von Mises equivalent stress. This allows us to check the internal consistency of the stresses via the standard von Mises definition.
>
> (2) __Constitutive Law Consistency__ (forming dataset)
> For *sheet metal forming*, two possible constraints are possible:
>    - Elastic violation rate: ratio of elastic nodes where predicted stress exceeds the initial yield stress.
>    - Plastic law residual: NMAE between predicted von Mises stress and the constitutive law yield stress for plastic nodes.
>
> (3) __Boundary Conditions__ (heatsink dataset)
> For *heatsink*, also two possible constraints are possible:
>    - Temperature BC: mean deviation of predicted temperatures from the imposed solid wall temperature (at fins).
>    - Velocity BC: no-slip condition violation as mean velocity magnitude at fin nodes.
>
> For more details check Appendix F.2.2 of the revision. We re-evaluated all of our trained models including these new metrics and added results in Table 3 as well as in the detailed results tables (Tables 4, 6, 8 and 10).
>
> ## [W4] Novelty and Datasets & Benchmarks
>
> We fully agree that our work does not introduce a new method, which is why we submitted in the datasets and benchmark category. However, as you also noted in the strengths, our contribution addresses an important and previously unexplored gap between two research fields. Currently, there is no benchmark for unsupervised domain adaptation in the very-high-dimensional regression setting of neural simulation. Despite the high practical relevance of such scenarios in scientific and engineering pipelines, this space remains largely unexplored.
>
> ## [W5] Clariying neural surrogates
> This point requires some clarification: the _neural surrogates_ denomination we adopt is commonly used in scientific and industrial ML when referring to models designed or trained to emulate computationally expensive simulations.
> A significant aspect that differentiates our setting from "standard supervised regression with UDA" is the output dimensionality: as far as we know there is no other work that tackles domain adaptation on regression with very high dimensional outputs. The only similar research directions with point-based (or pixel-based) outputs are on segmentation.
> As a final note, while PointNet and GraphSAGE are generic point cloud regressors, Transolver, UPT and GINO are specifically designed for physics data.
>
> ## [Q4] Domain gap difficulty scaling
>
> Our benchmark already evaluates all combinations of model, UDA algorithms and model selection methods across all datasets, leading to a total of 1,664 training runs including the additional ones we added for this rebuttal. The differences in domain gaps between the datasets can nicely be seen in Figures 7, 10, 15 and 20 that show the error distribution in the source and target domains respectively for all datasets. When comparing them, one can clearly see that the domain gaps for the *hot rolling* and *heatsink design* datasets are significantly larger (harder for the model) than the ones for the *sheet metal forming* and *electric motor design* dataset reflected by the larger difference in error distributions.
>
> Additionally we provide three domain shift difficulty levels for each dataset. The impact of this is shown for the *hot rolling* dataset in Figure 6, where we show the increasing target error with increasing domain shift difficulty.
>
> ## [Q5] Physics priors in neural surrogates
>
> Transolver, UPT and GINO are based on a continuum, field view of the space and were originally conceived as neural operators which implicitly include characteristics such as discretization convergence, local-global interactions, spectral structure, etc.
>
> As for other priors, we do not enforce symmetries or "baked-in" conservations. However, it is an interesting aspect that could bring improvements to OOD performance, especially in data-sparse settings [1,2]. Therefore we do believe that this is an important future direction.
>
> ---
>
> [1] Brehmer et al., 2024, Does equivariance matter at scale?
>
> [2] Prantl et al., 2022, Guaranteed Conservation of Momentum for Learning Particle-based Fluid Dynamics

---

### Official Review · Reviewer_uay5 · 2025-10-31

**Soundness:** 3
**Presentation:** 2
**Contribution:** 2
**Rating:** 6
**Confidence:** 3

**Summary:**

The paper provides a benchmark dataset with 4 industry relevant simulations with the goal of solving the unsupervised domain adaptation (UDA) problem for neural surrogates. They provide evaluate error across multiple baseline models, UDA methods, and unsupevised model selction methods.

**Strengths:**

- The paper provides code and data for industry relevant applications.
- Multiple models, UDA methods, and unsupervised model selection methods are tested.
- The supplied data and experimental details are thoroughly explained and provide a nice test bed for evaluating steady state problems with UDA for other researchers.

**Weaknesses:**

- It would be interesting to see the author's hypotheses on why certain UDA mehods and unsupervised model selection methods worked well in specific datasets.
- Steady state only problems limit the scope of this benchmark, adding time dependent problems would also be useful.

**Questions:**

- For the unsupervised domain adaptation methods, which distance metric $d$ do you use?
- In Figure 1 of the paper the caption refers to two loss terms $\mathcal{L}_{recon}$ and $\mathcal{L}_{DA}$ but these terms are not defined in the main text as far as I could tell.
- There are a few typos in the paper which should be corrected:
  - Line 105: "Numerous different Numerous benchmark datasets and evaluation protocols have been established" "Numerous" appears twice.
  - Line "Detailed and descriptions of the parameter sampling ranges can be found in Appendix F."

---

> ### Author Response · Authors · 2025-11-20
>
> We thank __uay5__ for the positive and detailed review. We continue with our response.
>
> ## [W1] Interpretation of UDA performance
> Thanks for the question. Our results indicate that UDA performance depends not only on the dataset characteristics but also on the model architecture. Table 3 summarizes the best-performing configurations, and while all benchmarked methods appear at least in one setting, Deep CORAL is the most frequent among the top results across datasets and models, despite it being the oldest algorithm in our benchmark (2016).
>
> A notable exception is the *hot rolling* dataset, where Deep CORAL does not appear among the best results. We hypothesize that this is related to its substantially smaller mesh size (~1k nodes) compared to the other datasets (up to ~1M nodes for heatsink). We hypothesize that for higher resolution datasets where the latent spaces naturally have a higher compression rate, Deep CORAL might perform best in most cases since it is arguably the least restricting of all approaches.
>
> Conversely, in low resolution settings where latent features need to compress less, more expressive alignment objectives, such as DARE-GRAM’s inverse-Gram matching or CMD's mathing of higher moments, may help preserve task-relevant structure more effectively than covariance alignment alone, leading to better results on the *hot rolling* dataset.
>
> ## [W2] Steady state
> Thank you for raising this. SIMSHIFT is intentionally built around steady-state problems (Section 3). Unlike many AI4Sci academic benchmarks, our goal is to reflect how simulation is actually used in industry. Most industrial simulation happens inside design-optimization loops, where downstream quantities are derived from steady-state (or statistically averaged) solutions. This holds across thermal systems [1], aerodynamic [2] and automotive design [3], and is reflected in large industry-focused datasets such as DrivAerNet++ [4] and DrivAerML [5] exclusively using steady-state fields.
>
> For some SIMSHIFT datasets, transient modeling is physically inappropriate (e.g., electric motors). For others it adds no value to the design task (e.g., heatsinks). We fully agree that transient dynamics matter in other domains, such as weather and climate. Extending SIMSHIFT to time-dependent PDEs would be valuable future work, but including fair, industrially relevant transient benchmarks is beyond the scope of this release.
>
> ---
>
> [1] P. Majumdar, Design of Thermal Energy Systems, 2021
>
> [2] S. N. Skinner and H. Zare-Behtash, State-of-the-art in aerodynamic shape optimisation methods, 2018
>
> [3] L. Dumas, CFD-based optimization for automotive aerodynamics 2008
>
> [4] Elrefaie M. et al. "DrivAerNet++" NeurIPS, 2024
>
> [5] Ashton, N, et al. "DrivAerML" preprint, 2024
>
> ## [Q1] UDA distances
> Thank you for the question, it was indeed not sufficently not explained in the original manuscript. Below, we summarize how the different algorithms compute the distance metric $d$. Please check Appendix D in the updated paper for the complete formulation and explanation of all variables.
>
> **Deep CORAL [1]**: Aligns the covariance matrices latents via:
>
> $d_{\text{deepcoral}}(\phi(\mathbf{x}),\phi(\mathbf{x'})) = \frac{1}{4k^2} \left\| \mathbf{C} - \mathbf{C'} \right\|_{F}^2.$
>
> **CMD [2]**: Aligns the up to $P$-moments of the latents using:
>
> $d_{\text{cmd}}(\phi(\mathbf{x}), \phi(\mathbf{x}')) = \frac{1}{|b-a|} \left\lVert \boldsymbol{\mu} - \boldsymbol{\mu}' \right\rVert_{2} + \sum_{p=2}^{P} \frac{1}{|b-a|^p} \left\lVert \mathbf{c}_p(\phi(\mathbf{x})) - \mathbf{c}_p(\phi(\mathbf{x}')) \right\rVert_2$
>
> **DANN [3]** takes a different approach. It reduces source–target shift by training a domain classifier, and uses its error to estimate the Proxy A-distance (an upper bound on the $\mathcal{H}$-divergence). Training is performed adversarially, i.e. the domain classifier maximizes accuracy and the encoder tries to fool the classifier, promoting domain invariant representations.
>
> **DARE-GRAM [4]:** Aligns the feature magnitude and the angles of the inverse-Gram subspaces, via:
>
>  $d_{\text{daregram}} = \alpha \cdot d_{\text{scale}}(G, G') + \gamma \cdot d_{\text{angle}}(G, G')$.
>
> For complete definitions of $d_{\text{scale}}(G, G') $ and $d_{\text{angle}}(G, G')$, please check Appendix.
>
> ---
>
> [1] Sun, B. & Saenko, K. (2016). Deep CORAL: Correlation Alignment for Deep Domain Adaptation
>
> [2] Zellinger et al. (2017). Central Moment Discrepancy (CMD) for Domain-Invariant Representation Learning
>
> [3] Ganin et al. (2015). Domain-Adversarial Training of Neural Networks
>
> [4] Ismail Nejjar et al. (2023). DARE-GRAM: Unsupervised Domain Adaptation Regression by Aligning Inverse Gram Matrices
>
> ## [Q2] Figure 1
>
> Thanks for pointing this out. We updated Equation 2 in Section 4.2 and added a passage clarifying the connection between the loss terms and Figure 1. The updated equation includes underbrackets.
>
> ## [Q3] Typos
> Thank you for pointing them out, we fixed them in the updated paper.

---

### Official Review · Reviewer_FxNP · 2025-11-01

**Soundness:** 2
**Presentation:** 2
**Contribution:** 3
**Rating:** 4
**Confidence:** 3

**Summary:**

The paper introduces SIMSHIFT, a benchmark dataset and evaluation suite covering four industrial simulation tasks to study UDA for neural surrogates on unstructured meshes. The framework explicitly conditions neural operators on configuration parameters via a sinusoidal (sin–cos) encoding followed by a shallow MLP, producing an 8-dimensional latent vector used for conditioning; baselines include PointNet, GraphSAGE (FiLM), Transolver (DiT), and UPT selected per task/scale. The study benchmarks three UDA algorithms (Deep CORAL, CMD, DANN) together with unsupervised model selection and reports metrics such as (N)RMSE and custom engineering metrics. All datasets are steady-state by design, and the heatsink meshes are subsampled to one quarter, with experiments showing that UDA+ unsupervised selection usually reduces target error but no single method dominates.

**Strengths:**

1. All architectures share the same conditioning network (sin–cos + shallow MLP, 8-dim latent), making the parametric setup consistent across models.

2. PointNet (global pooling), GraphSAGE with FiLM (local message passing), Transolver with DiT (attention with learned slicing), and UPT (latent field modeling for very large meshes) are adapted to conditional mesh regression and chosen to match scale constraints.

**Weaknesses:**

1. The work extends established DA methods to neural surrogates and benchmarks them, it does not introduce a new neural operator or a new UDA/model-selection algorithm (conditioning, FiLM/DiT choices are standardized, not novel).

2. While the paper reviews neural operator literature, the baseline lineup focuses on PointNet/ GNN/ Transformer/ UPT variants; adding FNO/Geo-FNO/GKN comparisons would better situate results, especially in 3D-large-mesh regimes.

3. The datasets are steady-state only, which may limit claims about transient dynamics and extreme-scale fidelity.

**Questions:**

1. To sharpen the model-side contribution, could you add and ablate conditioning variants (e.g., learned frequency embeddings, FiLM/DiT) and report stability/accuracy sensitivity?

2. Can you provide ablations that incorporate physics-informed constraints (residuals/conservation/BCs) into the loss within the same training–selection pipeline to assess reliability for industrial deployment?

3. Will you include operator baselines (e.g., FNO/Geo-FNO/GKN/GNO)—ideally on the 3D heatsink—to clarify accuracy–efficiency trade-offs against widely used PDE surrogates?

---

> ### Author Response · Authors · 2025-11-20
>
> Thanks for your review and questions, the baseline additions helped strengthen the paper. Our response follows below.
>
> ## [W1] Novelty and Dataset & Benchmarks
> Thank you for the suggestion. We fully agree that our contribution does not introduce a new neural operator or a new UDA/model-selection algorithm. This is intentional and is the reason we submitted to the Datasets & Benchmarks category. The value we provide is laying the foundation of method and algorithm development at the intersection of distribution shifts and neural simulation surrogates.
>
> ## [W2] Neural Operator Baselines
> We understand the reviewer's point and agree that it would be beneficial to include a method that learns in the spectral domain, given their popularity and wide adoption. We include Geometry-Informed Neural Operator (GINO), the most recent FNO variant for large scale irregular grids.
>
> We benchmark GINO for the 3D heatsink dataset, results below. We report target all fields NRMSE (4 seeds, (-) indicates unregularized). See Table 10 in the updated manuscript for the full results.
>
> | UDA|Selection|NRMSE TGT
> |---|---| ---
> |- | - | 0.484 ± 0.023
> |DANN | DEV | 0.486 ± 0.026
> |CMD | SB | 0.485 ± 0.010
> |DARE-GRAM | DEV | 0.393 ± 0.040
> |Deep CORAL | SB | 0.356 ± 0.011
>
> GINO performs worse than UPT/Transolver, possibly due to the relatively coarse latent grid (16x16x16), set to match the other model's param count. Due to the multi-scale irregular mesh, in coarse regions the projection onto the latent regular grid is suboptimal (pooling works with a fixed radius around the latent nodes). We include GINO in Section 4.4. and details Appendix E.
>
> ## [W3] Steady state
> Thank you for raising this. SIMSHIFT is intentionally built around steady-state problems (Section 3). Unlike many AI4Sci academic benchmarks, our goal is to reflect how simulation is actually used in industry. Most industrial simulation happens inside design-optimization loops, where downstream quantities are derived from steady-state (or averaged) solutions. This holds across thermal systems [1], aerodynamics [2, 3], and is reflected in large industry-focused datasets such as DrivAerNet++ [4] and DrivAerML [5] exclusively using steady-state fields.
>
> For some SIMSHIFT datasets, transient modeling is physically inappropriate (e.g., electric motors). For others it adds no value to the design task (e.g., heatsinks). We agree that transient dynamics matter in other domains, such as weather and climate. Extending SIMSHIFT to time-dependent PDEs would be valuable future work, but including fair, industrially relevant transient benchmarks is beyond our scope.
>
> ---
>
> [1] P. Majumdar, Design of Thermal Energy Systems, 2021
>
> [2] S. N. Skinner and H. Zare-Behtash, State-of-the-art in aerodynamic shape optimisation methods, 2018
>
> [3] L. Dumas, CFD-based optimization for automotive aerodynamics 2008
>
> [4] Elrefaie M. et al. "DrivAerNet++" NeurIPS, 2024
>
> [5] Ashton, N, et al. "DrivAerML" preprint, 2024
>
> ## [Q1] Conditioning ablations
> Regarding the different embedding variants and conditioning, we experimented with multiple ones during development. Ultimately, we adopted sin–cos embeddings and FiLM/DiT/concatentation because they are widely used, relatively stable, and not the limiting factor in the UDA setting.
>
> Given the computational cost of the requested ablations during this review, we focused our resources on ablating central aspects like novel operators and UDA methods. We see architectural conditioning ablations as valuable future work that our benchmark allows easily. However we think that they are not central to the contribution of this paper: broad search of models, UDA algorithms and model-selection strategies.
>
>
> ## [Q2] Physics-informed constraints
> Thank you for this suggestion. We agree that incorporating physics-informed constraints is an important future direction and could make neural surrogates more reliable in industrial settings.
>
> In our revised manuscript, we took a step in this direction by adding physics-based metrics for all datasets (see Section F.2.2 and updated tables). They are tailored to the underlying PDEs and modeling assuptions (e.g., BC satisfaction in heatsinks, von Mises consistency in forming, etc.) and therefore allow us to assess the physical "coherence" under domain shifts.
>
> However, even basic PINN (MLP) are often challenging: conflicting gradients [1], ill-conditioned losses [2]. We speculate that optimizing complex neural surrogates on the proposed metrics would prove to be unstable. Moreover, the cost of tuning and ablating each component quickly explodes in the context of SIMSHIFT. We see a physics-inspired domain adaptation algorithm as a very intersting research direction, and add it to future work in the revision (Section 6).
>
> ---
>
> [1] Liu et al., 2021, Conflict-Averse Gradient Descent for Multi-task Learning
>
> [2] Rathore et al., 2024, Challenges in Training PINNs: A Loss Landscape Perspective.
>
> ## [Q3]
> We include GINO. See answer to W2.

---

### Official Review · Reviewer_BN5C · 2025-11-02

**Soundness:** 3
**Presentation:** 2
**Contribution:** 3
**Rating:** 6
**Confidence:** 4

**Summary:**

This paper tackles the problem of unsupervised domain adaptation (UDA) in the scientific machine learning context, for modeling PDE-governed processes. Specifically, the models evaluated in the paper aim to perform UDA under the covariate-shift assumption and employing the Radon-Nikodym derivative for learning distributional transfer from a source distribution to a target distribution where ground-truth solutions are only present in the source distribution. To demonstrate UDA capability, the scientific processes the paper focuses on are self-generated datasets of steady-state processes with applications in metallurgy, manufacturing, machinery and electronics. To evaluate the UDA capability of the proposed methods, the source and target domains are created by splitting the input parameter configurations in a disjoint manner across sorce and target domains. Further, the paper also details performance of state of the art neural surrogates and unsupervised domain adaptation (UDA) techniques on the proposed dataset. Overall, the dataset and the proposed model benchmark evaluations are a good contribution to push research on the critical problem of UDA in the scientific machine learning context.

**Strengths:**

1. The investigated context of unsupervised domain adaptation (UDA) is a critical requirement in the scientific machine learning context because, for a neural surrogate to be useful, it needs to be able to "extrapolate" to unseen parameter configurations. This important problem has (unfortunately) not been investigated in depth as yet by the scientific machine learning community. Hence, this paper is a good start towards proposing a benchmark dataset as well as quantitative evaluations of state-of-the-art surrogates,  model selection techniques to perform the UDA task.

2. The four problems investigated by the paper are (i) diverse (ii) each challenging in their own right (iii) well-motivated for the UDA problem as they each require costly computational simulation for data generation.

**Weaknesses:**

1. The paper requires better organization as there are multiple critical sections where the appendix is referred to, interrupting the flow of the paper and preventing the full understanding of the problem. One such point is the lack of clear demarcation of source and target domains for the four datasets in the main paper (this is done in the appendix in Appendices F1 - F4 where actual parameter ranges are discussed and Appendix C:Table 11 where actual source and target distribution splits are presented).

2. Full cost of the evaluated UDA methods (i.e., surrogate training with UDA + model selection) is unclear from the main paper as presented. For a holistic presentation as a benchmark, this is necessary. Understanding how the training cost scales with (i) number of data points (ii) mesh size, is crucial for independent adaptation of the proposed methods by interested readers to other UDA contexts.

**Questions:**

1. How are the specific parameter ranges for `source configurations` and `target configurations` arrived at for each of the four datasets? Also, how are the "easy", "medium" and "hard" configurations decided? A more pronounced and cohesive description in the main paper of the actual configurations (i.e., parameter ranges), and the classification into "easy", "medium" and hard" would help to clarify the problem design.

2. Are there examples in any of the datasets investigated, where a significant distributional shift occurs in the underlying dynamics? Meaning, even if P_s(Y|X) = P_t(Y'|X'), are there any examples where P_t(Y') is itself out-of-distribution relative to P_s(Y)? Reporting model performance (or identifying any failures) in such cases would also be insightful.

3. Could you please provide numbers for (or pointer to a part of the paper that discusses) total training time of the proposed methods (i.e., surrogate training with UDA, model selection). Specifically, it would be useful to understand how total training time scales with training data size as well as the mesh size (i.e., number of mesh nodes)?

---

> ### Author Response · Authors · 2025-11-20
>
> We are thankful to __BN5C__ for the positive comments and insightful questions. Our response follows below.
>
> ## [W1/Q1] Writing
> Thank you for pointing out these issues regarding clarity and organization. With the additional page being allowed for the revised version, we implemented several improvements to make the paper more self-contained:
>
> **Parameter ranges.**
> We moved the table defining the parameter ranges for all datasets and difficulties to Section 3.5 of the main paper to improve the flow of the paper.
>
> **Difficulty splits.**
> To extend scalar parameter ranges as simplistic proxies for shift difficulty, we add a novel experiment in the main paper: we provide a distance measure in output space (ground truth simulation fields). The maximum transfer error of a model can be bounded (see, e.g., [1, 2] for a summary) by distances such as the $\mathcal{H}$-divergence (estimate is poxy $\mathcal{A}$-distance (PAD) [3, 4]). To strengthen our domain splits analysis, we therefore add PAD values for all settings in Table 2 of the updated manuscript, where we rely on a PointNet mesh source/target classifier for PAD estimation. We show the results for all three difficulty levels and all four datasets below:
>
> | Dataset  |  Easy  | Medium |  Hard  |
> | :------- | :----: | :----: | :----: |
> | Rolling  | 1.063  | 1.159  | 1.210  |
> | Forming  | 0.860  | 0.938  | 1.030  |
> | Motor    | 0.762  | 0.932  | 0.955  |
> | Heatsink | 1.446  | 1.683  | 1.861  |
>
> The new PAD values are consistent with our results of increasing transfer error with increasing domain difficulty on the *hot rolling* task (Figure 6). For all datasets, PAD increases with increasing difficulty.
>
> ---
>
> [1] Bouvier et al. "Robust domain adaptation: Representations, weights and inductive bias.", 2020.
>
> [2] Johansson et al. "Support and invertibility in domain-invariant representations.", 2019.
>
> [3] Zellinger et al. "The balancing principle for parameter choice in distance-regularized domain adaptation.", 2021.
>
> [4] Ben-David et al. "A theory of learning from different domains."  2010.
>
> ## [W2/Q3] Runtimes
> We agree with this, and in Table 13 we added training times (2k epochs) for all models and datasets. To clarify this, we also include samples and average node count to the runtimes table and an explanation of the full UDA pipeline cost in the corresponding section (Appendix F.3) and below:
>
> | Dataset | Samples | Avg. \# Nodes | Model | \# parameters | Avg. training time (h) |
> | :--- | :--- | :--- | :--- | :--- | :--- |
> | Rolling | 4,750 | 576 | PointNet | 0.3M | 0.75 |
> | | | | GraphSAGE | 0.2M | 1.77 |
> | | | | Transolver | 0.57M | 1.77 |
> | Forming | 3,315 | 6,417 | PointNet | 0.3M | 2.35 |
> | | | | GraphSAGE | 0.2M | 10.82 |
> | | | | Transolver | 0.57M | 3.74 |
> | Motor | 3,196 | 9,052 | PointNet | 0.3M | 2.35 |
> | | | | GraphSAGE | 0.2M | 12.14 |
> | | | | Transolver | 0.57M | 3.60 |
> | Heatsink | 460 | 1,385,594 | PointNet | 1.08M | 3.88 |
> | | | | Transolver | 4.07M | 4.94 |
> | | | | UPT | 5.77M | 4.73 |
> | | | | GINO | 2.5M | 5.94 |
>
> All runs were measured on a NVIDIA H200 144GB GPU. This table refers to models trained with Deep CORAL, however different UDA algorithms (or no UDA) do not add measurable computational cost. Calculation for the full pipeline including model selection can be estimated by multiplying the training time by the number of trained models (e.g. $\times 9$ if you sweep over 9 hyperparameters of $\lambda$), for sequential execution. Furthermore the model selection method's runtime after training is negligible compared to the training times.
>
> ## [Q3] Label shifts
> Yes, output-space distribution shifts are present in all datasets, because some parameter-range splits generate target fields who are out-of-distribution, even though the underlying physical mapping (represented by the deterministic numerical simulator) remains the same. This is also reflected by the PAD values that we describe above and add in Table 2 in the updated manuscript.
>
> To make this more concrete, we can give a prominent example where the solution fields in the target are partially outside the statistical range those in the source field. In the *medium* difficulty setting for the *hot rolling* dataset, the range of PEEQ in the source domain is: [0, 0.19], whereas the target domain's range is: [0, 0.28].
>
> We added these clarifications in Section 3.5 to improve the reading experience.

---

### Author Response · Authors · 2025-11-20
**General Answer**

We thank all reviewers for their efforts in providing constructive feedback and questions, enabling us to significantly improve the quality of our work. __Significant additions to the revised version are highlighted in yellow.__

We summarize the praises and concerns shared by reviewers, and elaborate on how we addressed the concerns.

### Major praises
Reviewers (__BN5C__, __uay5__, __hbC7__) recognize SIMSHIFT as a valuable, well-documented, systematic testbed for evaluating and developing high-dimensional UDA regression methods in the context of simulation. The benchmarking itself is thorough and consistent (__FxNP__, __uay5__, __hbC7__). It also addresses the critical and under-studied problem of domain shifts for scientific machine learning surrogates (__BN5C__, __hbC7__). Reviewers also recognize that the four datasets are diverse, challenging, realistic, and industrially relevant, covering a diverse range of physics (__BN5C__, __uay5__, __hbC7__).

### Concerns
- __Evaluation Metrics__: physics-informed evaluation metrics are not included (__FxNP__, __hbC7__). We added a selection of physics conservation and consistency metrics to each dataset.
- __Dataset Scope__: The choice of steady-state only designs (__FxNP__, __uay5__) is intentional and reflects dominant industrial use-cases.
- __Presentation__: Improve organization (__BN5C__, __uay5__). We added further descriptions and moved essential tables (e.g., domain splits) and definitions from the appendix to the main paper and add a section on computational cost. We hope this makes it easier to follow the manuscript better and clarifies the unclear components.
- __Contribution Scope__: The work is a benchmark and does not introduce a new neural operator or a new UDA algorithm (__FxNP__, __hbC7__) but rather enables standardized development and evaluation of high-dimensional regression UDA methods in simulation. We clarify this is an intentional choice for the Datasets & Benchmarks category.

---

### Comment · Area_Chair_zxDw · 2025-11-26
**reminder**

Dear Reviewers,

The authors have now posted their responses to your comments. As the next step in the discussion phase, please take a moment to review their rebuttal and engage with them through the discussion forum. Thank you for your continued effort and thoughtful contributions to this review.

Best,

Your AC

---

### Author Response · Authors · 2025-12-01
**On the OpenReview incident**

Dear Area Chair,

despite the unfortunate OpenReview situation, we fully agree with the Program Chairs' handling of the incident to preserve the integrity of the review process. We understand that this places additional responsibilities on the newly appointed Area Chairs, and we sincerely thank you for your time and effort.

Before the discussion period was frozen, we submitted a full rebuttal addressing all posted concerns. Unfortunately, there was no further engagement with our responses. We would kindly ask you to consider __our detailed rebuttals,__ which directly address all issues raised in the initial reviews as well as __the revised manuscript,__ which includes clarifications and extra experiments (with changes highlighted in yellow for convenience).

We believe the process strengthened our submission nevertheless. Thank you again for stepping in under these unusual circumstances.

Best,

The Authors

---

### Meta-Review · Area_Chair_8WPM · 2026-01-07

**Summary:**

The submission presents SIMSHIFT, a dataset-and-benchmark suite for unsupervised domain adaptation of neural surrogates on four industrial simulation tasks with unstructured meshes, and evaluates multiple surrogate architectures, UDA objectives, and unsupervised model selection methods. Reviewers generally view the benchmark direction as useful, but several concerns push the paper below the acceptance bar: the work is primarily an application/benchmarking of existing DA methods rather than offering new methodological insight; the scope of shifts tested is narrower than implied by the motivation; and the empirical results remain inconclusive, especially regarding which UDA/selection strategies reliably help across tasks. The rebuttal improves the completness of the paper (runtime reporting, difficulty quantification, added baselines/metrics), but does not fully address the core concerns.

**Reviewer Concerns:**

Some key clarity/completeness issues were addressed in the rebuttal. The authors moved parameter-range and split information into the main paper, added a quantitative measure of domain gap/difficulty (PAD), and provided a detailed runtime table including sample counts, mesh sizes, and training times. They also responded to requests for broader baselines by adding an operator-style method (GINO) on the 3D heatsink task, and they expanded evaluation beyond RMSE-style errors by adding physics-based consistency/constraint metrics tailored to each dataset (e.g., boundary-condition violations, constitutive/von Mises consistency). In addition, they included at least one more recent UDA method (DARE-GRAM) and explored a test-time adaptation-style baseline (SSA), which partially addresses concerns that the benchmark focused only on older UDA methods.

Despite the rebuttal, several core issues remain. The paper is largely a benchmarking exercise over existing UDA methods, which many reviewers felt is incremental even for a dataset/benchmark. The types of domain shift studied are still mostly parameter-range splits; more substantive “instrument” shifts (e.g., mesh resolution, discretization, solver changes, or genuinely different boundary conditions) are discussed but not systematically evaluated. Requests for stronger or more modern baselines and deeper architecture-side ablations were only partially addressed, leaving uncertainty about how robust the conclusions are under current best practices. Finally, the results reinforce that no UDA or model-selection method works consistently well across tasks, which limits the benchmark’s takeaways.

**Reviewer Scores:**

Given the rebuttal, the two initially positive-but-borderline reviews would likely remain near their original scores. The added runtime reporting, difficulty quantification, and expanded metrics/baselines address practical concerns, but the remaining issues, such as the scope of shifts and limited methodological insight, would likely prevent a upward revision. The initially below-threshold review would likely remain below threshold: while the authors added newer UDA baselines and physics-based metrics, the reviewer’s core critique about limited modern DA coverage and missing non-parametric shift benchmarking is only partially addressed. The initially rejecting review might increase slightly toward borderline due to the added methods and physics-based evaluation, but would likely remain negative because the paper still centers on older/standard DA techniques, has steady-state-only problems, and does not fully address the shift coverage implied in the motivation. Overall, the score distribution would still lean towards rejection.

---

### Decision · Program_Chairs · 2026-01-26

Reject